# A mosquito salivary protein-driven influx of myeloid cells facilitates flavivirus transmission

Zhaoyang Wang 1,2,3, Kaixiao Nie[4], Yan Liang[1,5], Jichen Niu[1], Xi Yu[1,5], Oujia Zhang[6], Long Liu 7,8, Xiaolu Shi[3], Yibaina Wang[9], Xuechun Feng[2], Yibin Zhu[1,3], Penghua Wang 10 & Gong Cheng 1,2,3,11✉

## Abstract

**Mosquitoes transmit many disease-relevant flaviviruses. Efficient viral transmission to mammalian hosts requires mosquito salivary factors. However, the specific salivary components facilitating viral transmission and their mechanisms of action remain largely unknown. Here, we show that a female mosquito salivary gland-specific protein, here named *A. aegypti* Neutrophil Recruitment Protein (*Aa*NRP), facilitates the transmission of Zika and dengue viruses. *Aa*NRP promotes a rapid influx of neutrophils, followed by virus-susceptible myeloid cells toward mosquito bite sites, which facilitates establishment of local infection and systemic dissemination. Mechanistically, *Aa*NRP engages TLR1 and TLR4 of skin-resident macrophages and activates MyD88-dependent NF-κB signaling to induce the expression of neutrophil chemoattractants. Inhibition of MyD88-NF-κB signaling with the dietary phytochemical resveratrol reduces *Aa*NRP-mediated enhancement of flavivirus transmission by mosquitoes. These findings exemplify how salivary components can aid viral transmission, and suggest a potential prophylactic target.**

**Keywords** Mosquito salivary protein; Neutrophil; Flavivirus; Toll-like receptor; NF-κB
**Subject Categories** Immunology; Microbiology, Virology & Host Pathogen Interaction

## Introduction

Mosquitoes are hematophagous insects that carry and transmit many human viruses in nature (Bartholomay et al, 2010; Ferguson, 2018; Gul et al, 2022). Mosquito-borne viruses, such as dengue (DENV), Zika (ZIKV) and West Nile (WNV) viruses, act as etiological agents of severe human diseases, such as hemorrhagic fever, biphasic fever, encephalitis, and meningitis. Mosquito-borne viruses cause hundreds of millions of infections and many deaths annually (Guzman and Harris, 2015; Katzelnick et al, 2017; Weaver and Lecuit, 2015). As a common trait of hematophagous insects, blood feeding allows viruses to efficiently cycle between hosts and mosquitoes. Mosquitoes acquire blood and infectious viral particles from infected hosts with viremia. Thus, the viruses replicate in the midgut epithelial cells and spread systematically through the hemocoel to the salivary glands, making the infected mosquitoes prepared to transmit the viruses through their subsequent bites (Pingen et al, 2017; Yu et al, 2019). During blood feeding, an infected mosquito probes the host epidermis and dermis for blood vessels and subsequently releases saliva as well as a significant amount of infectious viral particles (Lefteri et al, 2022; Pingen et al, 2016; Pingen et al, 2017). Numerous salivary proteins modulate a variety of host responses, including coagulation, platelet aggregation, thrombin activation, and vasodilation, among other pathways. In addition, there are a number of factors in mosquito saliva that have been identified as able of regulating host susceptibility to arbovirus infection, thus aggravating disease severity and host mortality (Jin et al, 2018; Lefteri et al, 2022; Pingen et al, 2016; Styer et al, 2011; Sun et al, 2020). Nonetheless, the particular salivary proteins responsible for this enhancement remain to be identified.

Mosquito saliva changes the microenvironment of cutaneous immunity, thereby enabling arboviruses to effectively establish their early infection in hosts (Guerrero et al, 2022; Lefteri et al, 2022; Pingen et al, 2016; Schmid et al, 2016; Surasombatpattana et al, 2012). Accumulating evidence indicates that mosquito bites result in an influx of inflammatory neutrophils that coordinate a localized innate immune response to recruit numerous virus-permissive myeloid cells toward the bite sites (Guerrero et al, 2022; Hastings et al, 2019; Lefteri et al, 2022; Pingen et al, 2016). These monocyte-lineage immune cells, such as cutaneous immature dendritic cell (DC) subsets (Schaeffer et al, 2015; Schmid and Harris, 2014; Wu et al, 2000), macrophages (Castanha et al, 2020; Schaeffer et al, 2015; Schmid and Harris, 2014) and monocytes (Chen et al, 1999; Halstead et al, 1977; Kou et al, 2008; Neves-Souza et al, 2005; Pingen et al, 2016; Schmid and Harris, 2014), show high

[1]New Cornerstone Science Laboratory, Tsinghua University-Peking University Joint Center for Life Sciences, School of Basic Medical Sciences, Tsinghua University, Beijing 100084, China. [2]Institute of Infectious Diseases, Shenzhen Bay Laboratory, Shenzhen 518000, China. [3]Institute of Pathogenic Organisms, Shenzhen Center for Disease Control and Prevention, Shenzhen 518055, China. [4]Department of Pathogen Biology, School of Basic Medicine, Shandong First Medical University & Shandong Academy of Medical Sciences, Jinan 250000, China. [5]School of Life Sciences, Tsinghua University, Beijing 100084, China. [6]Department of Pathogen Biology, School of Basic Medical Sciences, Capital Medical University, Beijing 100086, China. [7]Institute of Virology, Hubei University of Medicine, Shiyan 442000, China. [8]Department of Infectious Diseases, Renmin Hospital, School of Basic Medical Sciences, Hubei University of Medicine, Shiyan 442000, China. [9]China National Center for Food Safety Risk Assessment, Beijing 100022, China. [10]Department of Immunology, School of Medicine, University of Connecticut Health Center, Farmington, CT 06030, USA. [11]Southwest United Graduate School, Kunming 650092, China. ✉E-mail: gongcheng@mail.tsinghua.edu.cn

permissiveness to arbovirus infection. Thus, neutrophil-driven inflammation leads to an increase in myeloid cells at the mosquito bite site, and these myeloid cells serve as new cellular targets to retain viruses at the inoculation site (Guerrero et al, 2022; Lefteri et al, 2022; Pingen et al, 2016). Subsequently, the viruses disseminate into the blood circulation after primary replication in the host skin (Bryden et al, 2020; Guerrero et al, 2022; Lefteri et al, 2022; Pingen et al, 2016). Notably, neutrophil depletion and therapeutic blockade of cutaneous inflammation abrogated the bite-mediated promotion of arbovirus infection at both the skin inoculation site and lymphoid tissues (Hastings et al, 2019; Pingen et al, 2016), indicating that a neutrophil-driven influx of myeloid cells is essential to promote cutaneous infection. In this study, we individually expressed and purified A. aegypti salivary proteins in Drosophila S2 cells and then found that a saliva-specific protein activated neutrophil-driven leukocyte influx to facilitate Zika (ZIKV) and dengue (DENV) virus transmission through the Toll-like receptor (TLR)-MyD88 signaling pathway. Dietary supplementation with resveratrol, an anti-inflammatory phytochemical, reduced the saliva-mediated influx of cutaneous neutrophils, thus suppressing ZIKV transmission. This study offers a mechanistic insight into saliva protein-aided transmission of mosquito-borne flaviviruses and provides a potential prophylactic target to interrupt flavivirus transmission by mosquitoes.

## Results

### AaNRP is a female mosquito-specific salivary protein causing an influx of neutrophils and subsequent recruitment of myeloid cells toward mosquito bite sites

During arbovirus transmission by mosquitoes, numerous salivary proteins are intradermally inoculated into host skin with infectious viral particles (Kumar et al, 2021). An essential process triggered by mosquito saliva is the infiltration of inflammatory neutrophils, which coordinates the recruitment of cutaneous monocyte-lineage immune cells toward the biting site (Hastings et al, 2019; Lefteri et al, 2022; Pingen et al, 2016). We therefore aimed to identify the salivary protein(s) triggering the influx of cutaneous neutrophils during mosquito biting. In a recent study, we identified 71 proteins from the saliva of female A. aegypti mosquitoes by SDS–PAGE and mass spectrometry (Sun et al, 2020). Forty-two salivary proteins with a mass spectrometric score greater than 25 were selected (Appendix Table S1), and twenty-nine of them were successfully expressed and purified in Drosophila S2 cells (Fig. 1A,B). We next assessed the role of these salivary proteins in triggering neutrophil influx into the host skin. Thus, 100 ng of purified salivary proteins were individually inoculated into a mouse footpad in an intradermal manner. Subsequently, the skin of the inoculation site was collected at 4 h post inoculation (hpi) for flow cytometry analysis. Intradermal inoculation of salivary proteins encoded by the AAEL006424, AAEL003182, and AAEL007394 genes resulted in substantial infiltration of CD45$^+$CD11b$^+$Ly6G$^+$ neutrophils toward the inoculation site (Figs. 1C and EV1A), with the AAEL007394-encoded protein showing the most robust recruitment of neutrophils. Thus, we targeted AAEL007394 and designated its encoded protein A. aegypti neutrophil recruitment protein (AaNRP) throughout this investigation. In terms of temporal kinetics,

AaNRP-activated neutrophil infiltration occurred as early as 4 hpi and through 24 hpi with a peak at 12 hpi (Fig. 1D). We exploited a heat-inactivated AaNRP (hiAaNRP) and a noninflammatory control (NIC) salivary protein encoded by AAEL009524 as negative and unrelated controls (Fig. 1C), and an equivalent amount of mosquito saliva as a positive control, respectively. The skin injected with hiAaNRP or the NIC protein did not show any changes in abundance of neutrophils over a time course (Figs. 1D and EV1B). We further validated these flow cytometry results by an in situ immunohistochemistry assay (IHC) with a mouse Ly6G antibody (Fig. EV1C,D) and hematoxylin-eosin (HE) staining (Fig. EV1E). Intriguingly, AaNRP was specifically expressed in the salivary glands of female A. aegypti (Fig. EV1F) rather than in those of male A. aegypti (Fig. EV1G). Moreover, the expression of AaNRP was induced following blood feeding (Fig. EV1H), suggesting an additional role of AaNRP after bloodmeal.

Previous studies indicated that an inflammatory neutrophil influx can coordinate a localized innate immune response to recruit numerous flavivirus-permissive myeloid cells toward mosquito bite sites, therefore facilitating cutaneous flavivirus infection (Hastings et al, 2019; Lefteri et al, 2022; Pingen et al, 2016). Thus, we assessed the recruitment of myeloid cells toward the AaNRP-inoculated footpads. Notably, monocytes, macrophages, and immature dendritic cells (DCs) were highly aggregated at the sites inoculated with AaNRP or mosquito saliva at both 6 hpi (Fig. 1E) and 24 hpi (Fig. EV2A). Nonetheless, inoculation of hiAaNRP and a NIC protein did not result in the aggregation of myeloid cells (Fig. 1E). We next asked if AaNRP-mediated myeloid infiltration was dependent on neutrophils. To this end, we systemically removed neutrophils by intraperitoneal injection of a neutrophil-depleting monoclonal antibody (1A8 mAb) (Pingen et al, 2016) before administrating AaNRP. Inoculation of the 1A8 mAb significantly reduced the neutrophil populations in the blood circulation (Fig. 1F) and the AaNRP-mediated influx of neutrophils in the footpads (Fig. 1G). Intriguingly, 1A8 mAb effectively blocked the recruitment of monocytes, macrophages, and immature DCs toward the AaNRP-inoculated mouse footpads (Fig. 1H), suggesting that AaNRP-mediated influx of neutrophils is a prerequisite for recruitment of flavivirus-permissive myeloid cells.

We next assessed the role of AaNRP in neutrophil infiltration and subsequent recruitment of myeloid cells in a mosquito biting model. The AaNRP gene was silenced by intrathoracic injection of double-stranded RNA (dsRNA) in A. aegypti. Mosquitoes inoculated with GFP dsRNA served as a negative control. AaNRP was efficiently silenced at both the mRNA level detected by qPCR (Fig. 1I) and the protein level assessed by an immunofluorescence assay (IFA) (Fig. 1J) in the A. aegypti salivary glands. Three days post-gene silencing, the dsRNA-treated mosquitoes were allowed to bite the hind footpads of 129 Sv/Ev mice (each hind footpad was bitten by five mosquitoes). Subsequently, the skin of the bite sites was isolated at 4 h post biting (hpb) for a flow cytometry assay. Compared to that of mock controls, the footpad skin bitten by GFP dsRNA-inoculated mosquitoes showed a robust influx of neutrophils at the bite sites, while silencing of AaNRP gene nearly offset the mosquito bite-mediated promotion of neutrophil infiltration (Fig. 1K). The bite-mediated recruitment of monocyte-lineage immune cells was impaired by AaNRP silencing at both 6 hpb (Fig. 1L) and 24 hpb (Fig. EV2B), which was measured by a flow cytometry assay. In addition, the infiltration of cutaneous

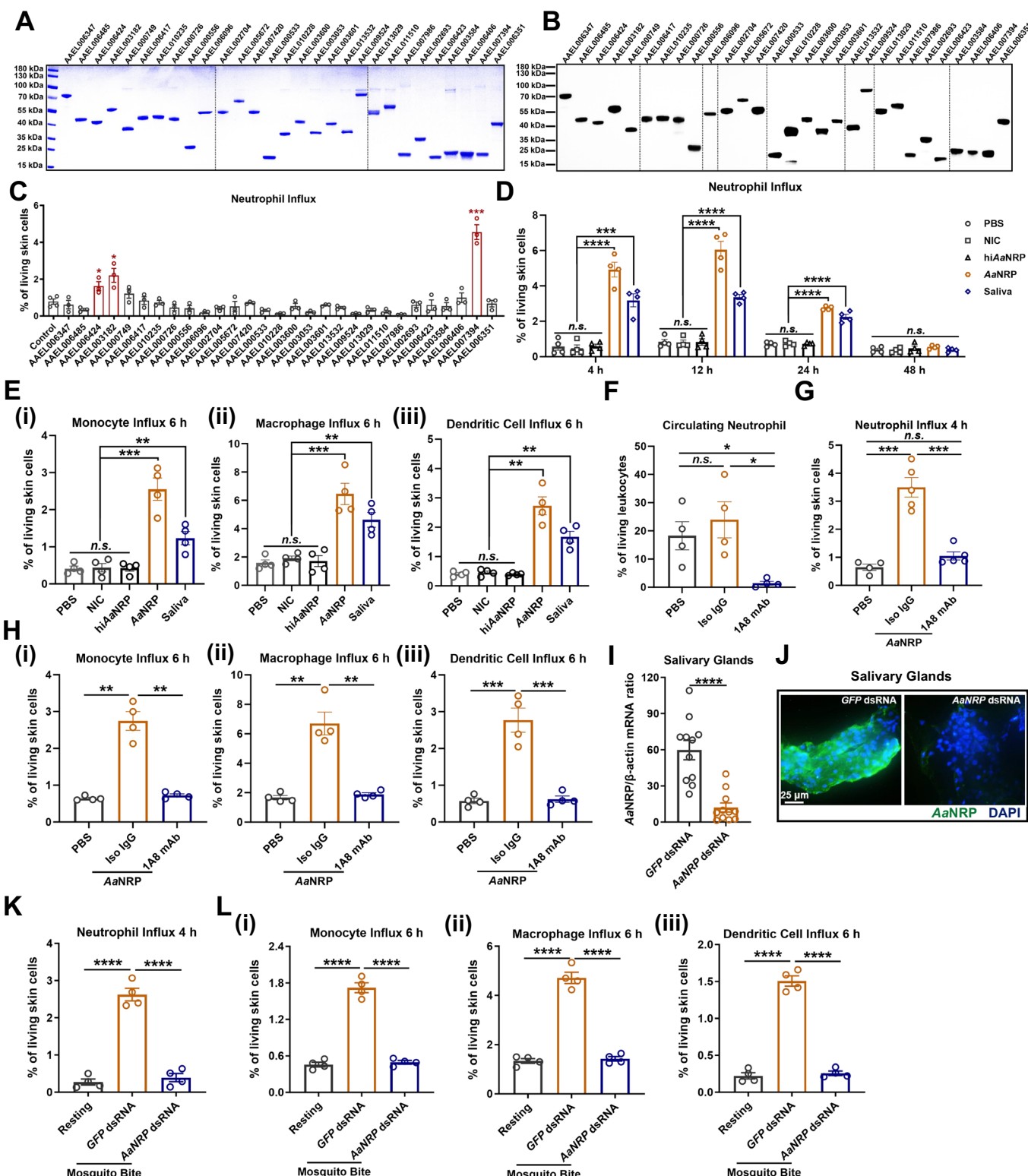

neutrophils at 4 hpb (Fig. EV2C,D) and the recruitment of myeloid cells at 24 hpb (Fig. EV2E–H) were validated by in situ IHC. Altogether, we found that *Aa*NRP acts as a female mosquito-specific salivary protein to trigger neutrophil infiltration and subsequent recruitment of myeloid cells to mosquito-bitten skin.

## *Aa*NRP stimulates skin resident macrophages to release chemoattractants for neutrophils

The abovementioned results demonstrated that salivary *Aa*NRP triggered an influx of inflammatory neutrophils at the bite site. We

**Figure 1.  Identification of *A. aegypti* salivary proteins inducing an influx of neutrophils and monocytic lineage cells into murine skin tissues.**

(A, B) Expression and purification of 29 recombinant *A. aegypti* salivary proteins in *Drosophila* S2 cells. (A) Purity verification of these recombinant salivary proteins by SDS–PAGE and Coomassie blue staining. (B) Validation of the recombinant salivary proteins by immunoblotting assay. (C) Flow cytometric analysis of neutrophils (CD45$^+$CD11b$^+$Ly6G$^+$) in mouse footpad skin at 4 h post inoculation (hpi) with *Aa*NRP. 129 Sv/Ev mice were intradermally inoculated with 100 ng of the recombinant salivary proteins, with PBS as the negative control. At 4 hpi, neutrophil influx in inoculated skin was analyzed by flow cytometry to determine the percentage of neutrophils in all living skin cells. (D) Temporal kinetics of skin neutrophil influx. 129 Sv/Ev mice were intradermally inoculated with 100 ng *Aa*NRP, equivalent amount of mosquito saliva as a positive control, or PBS, heat-inactivated *Aa*NRP (hi*Aa*NRP) and a noninflammatory control (NIC) salivary protein encoded by *AAEL009524* as negative controls. At 4, 12, 24, and 48 hpi, mouse footpads were sampled to measure the percentage of neutrophils in all living skin cells by flow cytometry. (E) Percentages of monocytes (CD45$^+$CD11b$^+$Ly6C$^+$Ly6G$^-$, i), macrophages (CD45$^+$CD11b$^+$F4/80$^+$, ii), and dendritic cells (DCs, CD45$^+$I-A/I-E$^+$CD11c$^+$, iii) in all living skin cells of the murine footpad skin at 6 hpi with *Aa*NRP. (F) The percentage of circulating neutrophils (CD11b$^+$CXCR2$^+$) in all leukocytes in murine peripheral blood. 129 Sv/Ev mice were intraperitoneally injected with 1 mg 1A8 antibody or 1 mg rat IgG2a as the isotype control at 4 days and 1 day before 100 ng *Aa*NRP inoculation in the footpad. Peripheral blood was sampled immediately before *Aa*NRP inoculation. (G, H) The percentage of neutrophils at 4 hpi (G) and the percentages of monocytes (H, i), macrophages (H, ii), and DCs (H, iii) at 6 hpi in all living skin cells. 129 Sv/Ev mice were intraperitoneally injected with 1 mg 1A8 antibody or 1 mg rat IgG2a as the isotype control at 4 days and 1 day before 100 ng *Aa*NRP inoculation in footpads. At 4 and 6 hpi, mouse footpads were collected for flow cytometry. (I, J) RNA (I) and protein (J) expression levels of *Aa*NRP in mosquito salivary glands. *A. aegypti* mosquitoes were intrathoracically injected with 1 μg/300 nL *Aa*NRP dsRNA or *GFP* dsRNA as the negative control. Three days later, the mosquito salivary glands were collected for qPCR detection of *Aa*NRP mRNA and immunofluorescence staining of *Aa*NRP protein, scale bar 25 μm. (K, L) The impact of *Aa*NRP silencing on the mosquito bite-induced neutrophil influx at 4 h post mosquito bite (hpb) (K) and the influx of monocytes (L, i), macrophages (L, ii), and dendritic cells (L, iii) toward the mouse footpad skin at 6 hpb. *A. aegypti* mosquitoes were intrathoracically injected with either 1 μg/300 nL *Aa*NRP dsRNA or *GFP* dsRNA as a negative control. Three days later, the mosquitoes were allowed to bite the hind footpads of 129 Sv/Ev mice. Each hind footpad was bitten by five mosquitoes, and unbitten mice were used as resting controls. (C–I, K, L) Data are expressed as the mean ± SEM, and each dot represents the data from an individual animal. The one-way ANOVA and multiple *t* tests were used for statistical analyses. All experiments were reproduced at least twice. *$p < 0.05$, **$p < 0.01$, ***$p < 0.001$, ****$p < 0.0001$, n.s. not significant. Source data are available online for this figure.

next assessed whether *Aa*NRP directly activated neutrophil migration by a transwell assay. Human and mouse neutrophils were isolated from healthy individuals and then seeded in a hanging cell culture insert above the transmigration membrane. The culture media containing a serial concentration of purified *Aa*NRP were added to the basolateral side of each well. The transmigration of neutrophils toward the lower chamber was recorded by a cell counter (Fig. EV3A). As a positive control, C-X-C motif chemokine ligand 1 (CXCL1) significantly triggered the neutrophil transmigration (Fig. EV3B). However, the number of neutrophils transmigrating to the bottom chamber with *Aa*NRP was not different from that of controls (Fig. EV3B), suggesting that *Aa*NRP cannot directly recruit neutrophils.

We next investigated the underlying mechanism of the mosquito bite-mediated promotion of neutrophil infiltration by a single-cell RNA sequencing (scRNA-Seq). The skin of mouse footpads at 4 hpb or without mosquito bites was isolated for enzymatic digestion. The total skin CD45$^+$ immunocytes were sorted by flowcytometry and subjected to 10× genomics scRNA-Seq. We collected transcriptomic data from 10,928 cells in the 0 h group and 10,511 cells in the 4 h group after quality control. Through Seurat bioinformatic analysis, the immune cells were clustered by Louvain community detection-based modularity optimization (Vallejo et al, 2022). Eight immune cell clusters were delineated, including 7508 mast cells (3482 at 0 h and 4026 at 4 h), 6997 macrophages (4813 at 0 h and 2184 at 4 h), 2620 T cells (743 at 0 h and 1877 at 4 h), 1513 dendritic cells (766 at 0 h and 747 at 4 h), 907 Langerhans cells (537 at 0 h and 370 at 4 h), 729 natural killer (NK) cells (246 at 0 h and 483 at 4 h), 671 neutrophils (163 at 0 h and 508 at 4 h), and 265 eosinophils (93 at 0 h and 172 at 4 h) (Fig. 2A). Intriguingly, the mosquito bites significantly increased the percentage of CD45$^+$ cells in the living skin cells assessed by flow cytometry (Fig. 2B). The top 35 differentially expressed genes in each type of immune cell were selected for analysis (Appendix Fig. S1A–D and Appendix Table S2). Notably, pivotal chemokines that function in recruiting and activating neutrophils toward inflammatory sites, including CXCL1 (Kolaczkowska and Kubes, 2013), CXCL2 (Kolaczkowska

and Kubes, 2013) and CXCL3 (Pingen et al, 2016; Wuyts et al, 1999), were mainly expressed and specifically induced in cutaneous macrophages at 4 hpb compared to the macrophages from the control skin (Fig. 2C; Appendix Fig. S1A–D). The expression of these chemokines in the other individual immune populations was not affected (Appendix Fig. S1A–D). The upregulation of these 3 CXCL chemokine genes in cutaneous macrophages were validated by qPCR at 4 hpb (Fig. 2D). Nonetheless, scRNA-Seq was used to detect the regulation of gene transcripts post mosquito bites. The preformed chemicals stored in granules may be released from activated mast cells, thus leading to the recruitment of neutrophils (Demeure et al, 2005). We therefore assessed the activation of cutaneous mast cells at 4 hpb. The percentage of activated mast cells, defined as CD63$^+$CD203c$^+$ (Ebo et al, 2021), showed no significant difference between resting skin and mosquito bitten skin (Fig. EV3C), indicating that the mosquito bite does not induce the activation of cutaneous mast cells at the early time point after biting. Subsequently, we removed the cutaneous macrophages by intraperitoneal injection of a macrophage depleting antibody (Albacker et al, 2013; Wang et al, 2019a, 2018). The effectiveness of skin macrophage depletion was assessed by flow cytometry (Fig. EV3D) and in situ IHC staining (Fig. EV3E). Depletion of cutaneous macrophages significantly offset the recruitment of neutrophils to the mosquito bite site (Fig. 2E,F). Overall, these data indicate that cutaneous resident macrophages are activated by mosquito saliva to release neutrophil chemoattractants.

Because *Aa*NRP was found to be a mosquito salivary protein that activates cutaneous infiltration of neutrophils, we next assessed whether *Aa*NRP did so by inducing the expression of CXCL1/2/3 in skin resident macrophages. The abundances of these CXCL proteins were robustly enhanced by the intradermal inoculation of purified *Aa*NRP protein in mouse footpads, measured by an ELISA assay (Fig. 2G). We next determined whether *Aa*NRP was a salivary component upregulating the expression of neutrophil chemoattractants by skin resident macrophages. To this end, we silenced the *AaNRP* gene by dsRNA in *A. aegypti* mosquitoes and employed *GFP* dsRNA as a negative control. Three days

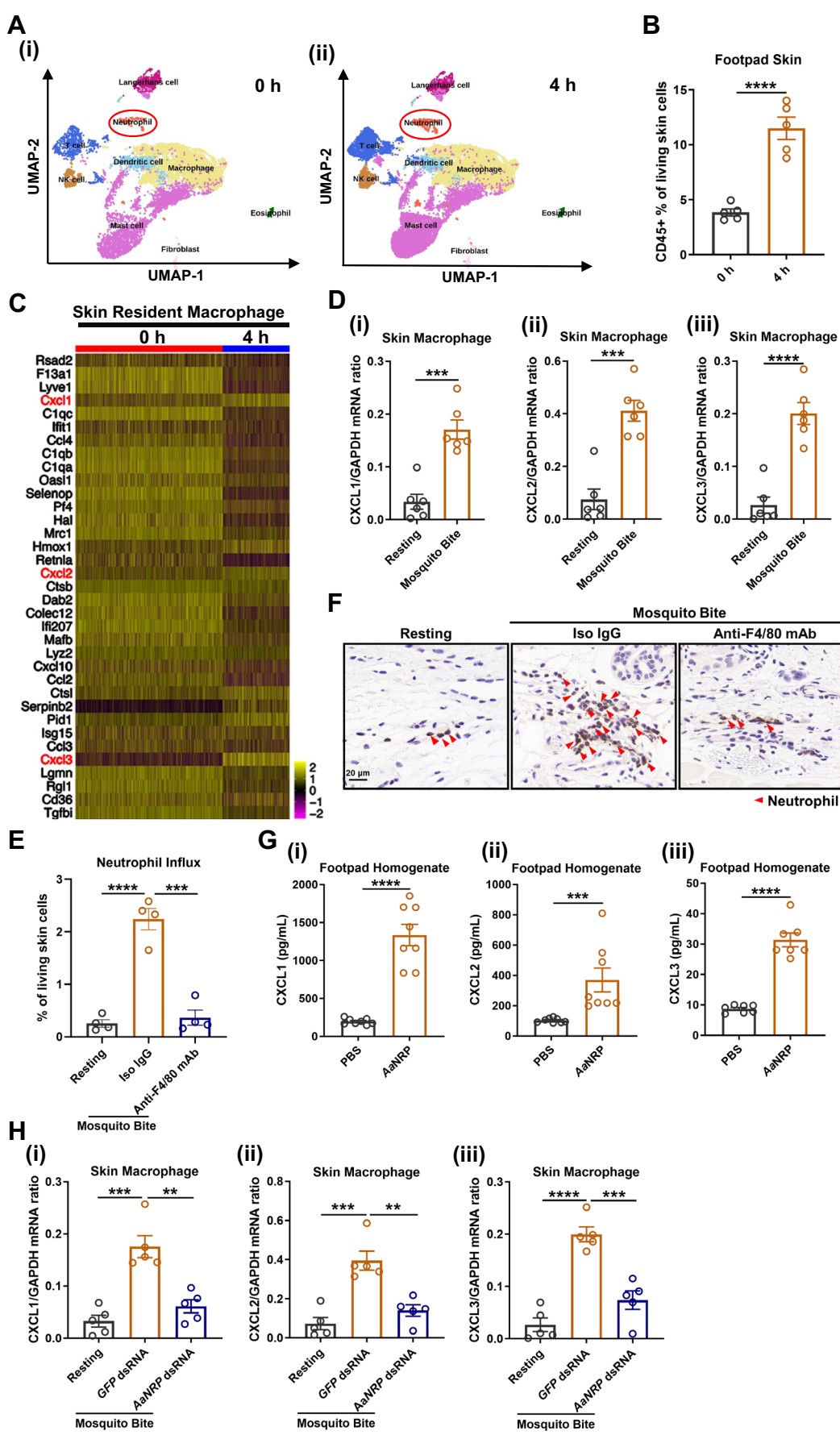

**Figure 2. Skin resident macrophages mediate *Aa*NRP-elicited neutrophil influx.**

(A) Clustering and annotation of skin immunocytes in both the 0 h group (i) and the 4 h group (ii). The mice in the 4 h group were bitten by mosquitoes in the hind footpads (each footpad was bitten by five mosquitoes), while the mice in the 0 h group were left unbitten. Eight mice were used in each group. Mouse footpads from each group were pooled together to represent the average status of the group. CD45$^+$ immunocytes were flow-sorted for 10× genomics single-cell RNA sequencing (scRNA-Seq). The transcriptomic data from 10,928 cells in the 0 h group and 10,511 cells in the 4 h group were successfully recorded after quality control. Through Seurat bioinformatic analysis, immune cells were clustered by Louvain community detection-based modularity optimization (Vallejo et al, 2022). Eight immunocyte clusters were delineated in each group, including 7508 mast cells (3482 in the 0 h group and 4026 in the 4 h group), 6997 macrophages (4813 in the 0 h group and 2184 in the 4 h group), 2620 T cells (743 in the 0 h group and 1877 in the 4 h group), 1513 dendritic cells (766 in the 0 h group and 747 in the 4 h group), 907 Langerhans cells (537 in the 0 h group and 370 in the 4 h group), 729 Natural Killer (NK) cells (246 in the 0 h group and 483 in the 4 h group), 671 neutrophils (163 in the 0 h group and 508 in the 4 h group), and 265 eosinophils (93 in the 0 h group and 172 in the 4 h group). The original scRNA-Seq data is available at NCBI-GEO with the accession number GSE232756. (B) Percentages of CD45$^+$ myeloid cells within all living skin cells of murine footpad skin. (C) Heatmap of the top 35 differentially expressed genes (DEGs) in skin macrophages in both the 0 h group and 4 h group. This heatmap is extracted from Appendix Fig. S1 with a higher resolution and the yellow color scales higher expression level of a gene. CXCL1/2/3 (marked in red font) were highly expressed by the skin macrophages and were significantly induced by mosquito bite. (D) RNA expression levels of CXCL1 (i), CXCL2 (ii), and CXCL3 (iii) in skin in situ macrophages at 4 hpb. Mosquito-bitten mice were bitten in the hind footpads (one footpad was bitten by five mosquitoes), while resting mice were left unbitten. The two hind footpads of each mouse were collected at 4 hpb and pooled together to represent an individual. The skin macrophages (CD45$^+$CD11b$^+$F4/80$^+$) in footpads were flow-sorted for qPCR quantification of CXCL1/2/3. (E, F) Flow cytometric analysis (E) and immunohistochemical (IHC) staining (F) of neutrophils in murine footpad skin at 4 hpb. 129 Sv/Ev mice were intraperitoneally injected with 1 mg of anti-F4/80 antibody (with 1 mg of isotype rat IgG2b serving as the isotype control) at 2 days and 1 day prior to mosquito biting to deplete skin resident macrophages. Subsequently, the antibody-injected mice were bitten by mosquitoes in the hind footpads (one footpad was bitten by five mosquitoes), while the resting mice were left unbitten. The two hind footpads of each mouse were collected at 4 hpb and pooled together to represent an individual for flow cytometric analysis or immunohistochemical staining of skin infiltrated neutrophils (indicated by red arrows) using a neutrophil-specific Ly6G antibody, scale bar 20 µm. Two footpad sections were obtained from each mouse, and three random scopes were sampled from each section, the most representative image was selected to delegate the mouse. (G) The contents of CXCL1 (i), CXCL2 (ii) and CXCL3 (iii) in footpad homogenates. 129 Sv/Ev mice were intradermally inoculated with 100 ng *Aa*NRP or 20 µL PBS as the control in the hind footpads. At 4 hpi, mouse footpads were collected and homogenized in 500 µL of 1× cell lysis buffer. The supernatants were recovered and subjected to enzyme-linked immunosorbent assay (ELISA) to measure the contents of CXCL1/2/3. (H) RNA expression levels of CXCL1 (i), CXCL2 (ii), and CXCL3 (iii) in skin in situ macrophages at 4 hpb. The mosquitoes were intrathoracically injected with *Aa*NRP dsRNA or *GFP* dsRNA as the control. Three days later, the dsRNA-treated mosquitoes were allowed to bite the hind footpads of 129 Sv/Ev mice. At 4 hpb, skin in situ macrophages were isolated from both the mosquito-bitten and resting footpad skin using flow cytometric sorting for the qPCR detection of CXCL1/2/3. (B, D, E, G, H) The data are expressed as the mean ± SEM, and each dot represents data from an individual animal. The unpaired *t* test (B, D, G) and one-way ANOVA and multiple *t* tests (E, H) were used for statistical analyses. All experiments were repeated at least twice. \*\**p* < 0.01, \*\*\**p* < 0.001, \*\*\*\**p* < 0.0001. Source data are available online for this figure.

post-dsRNA treatment, the mosquitoes were allowed to bite the footpads of 129 Sv/Ev mice. Subsequently, cutaneous macrophages were flow-sorted from bitten skin at 4 hpb for detection of CXCL1/2/3 mRNA expression. The mRNA levels of these CXCLs were significantly higher in the macrophages of *GFP*-dsRNA mosquito-bitten footpads than in resting footpads (Fig. 2H). Strikingly, the mosquito bites without *Aa*NRP were unable to promote the expression of these CXCLs in cutaneous macrophages (Fig. 2H). Altogether, these results demonstrated that *Aa*NRP is the key salivary component activating the expression of neutrophil chemoattractants in skin resident macrophages, thus leading to neutrophil infiltration into the mosquito bite sites.

## *Aa*NRP induces expression of these chemoattractants via the MyD88-NF-κB axis in macrophages

MyD88-mediated NF-κB signaling is known to regulate the transcription of CXCL1 (Blackwell et al, 1994; Kemp et al, 2021), CXCL2 (King et al, 2000) and CXCL3 (Lawrence, 2009). We next investigated whether *Aa*NRP induces CXCL expression via the MyD88-NF-κB axis using the mouse macrophage cell line RAW264.7. In agreement with the above in vivo results, CXCL1/2/3 expression was induced by *Aa*NRP in RAW264.7 cells (Fig. 3A). We therefore assessed whether *Aa*NRP activates NF-κB signaling by performing a dual luciferase reporter assay with a NF-κB response element-driven firefly luciferase and an actin promoter-driven *Renilla* luciferase as an internal reference. *Aa*NRP stimulated NF-κB-driven firefly luciferase expression in a dose-dependent manner (Fig. 3B), indicating the activation of NF-κB signaling by *Aa*NRP. To assess the potential activation of the NF-κB signaling mediated by LPS contamination in the purified

*Aa*NRP protein, we assessed the amount of LPS in the *Aa*NRP preparation and the PBS solution used as the negative control throughout our study. The endotoxin concentration in the *Aa*NRP preparation showed no difference with the PBS solution (Appendix Fig. S2A). To further address the concern of LPS contamination, we next used polymyxin B, a reagent effectively neutralizing LPS, to assess the potential presence of LPS in *Aa*NRP preparation. Purified *Aa*NRP and LPS were incubated with sterile water (control) or with polymyxin B at 4 °C for 24 h. Then, the sterile water or polymyxin B-treated *Aa*NRP and LPS were added to murine RAW264.7 macrophages for incubation, respectively. Four hours later, the cells were collected for qPCR detection of CXCL1/2/3 expression. The results showed that both *Aa*NRP and LPS significantly induced the expression of CXCL1/2/3 (Appendix Fig. S2B). Nonetheless, polymyxin B preincubation had no influence on *Aa*NRP-induced expression of CXCL1/2/3, but completely abolished LPS-induced CXCL1/2/3 expression (Appendix Fig. S2B). These data further validated the specific induction of these chemokines by *Aa*NRP.

NF-κB signaling can be induced by either canonical or alternative pathways. In the canonical pathway, the MyD88 signaling axis leads to phosphorylation and degradation of IκB-α and then phosphorylation and activation of RelA (p65)/p50, which regulates the expression of many proinflammatory genes (Hayden and Ghosh, 2008). In contrast, the alternative NF-κB pathway is MyD88-independent and activated by LT-β, CD40L, BAFF and RANKL, which results in activation of the RelB/p52 complex (Lawrence, 2009). Indeed, *Aa*NRP strengthened the phosphorylation and degradation of IκB-α while activating p65 phosphorylation in a dose-dependent manner (Fig. 3C). Nonetheless, phosphorylation of RelB was not regulated by *Aa*NRP (Fig. EV3F), suggesting that *Aa*NRP specifically stimulated the canonical NF-κB signaling

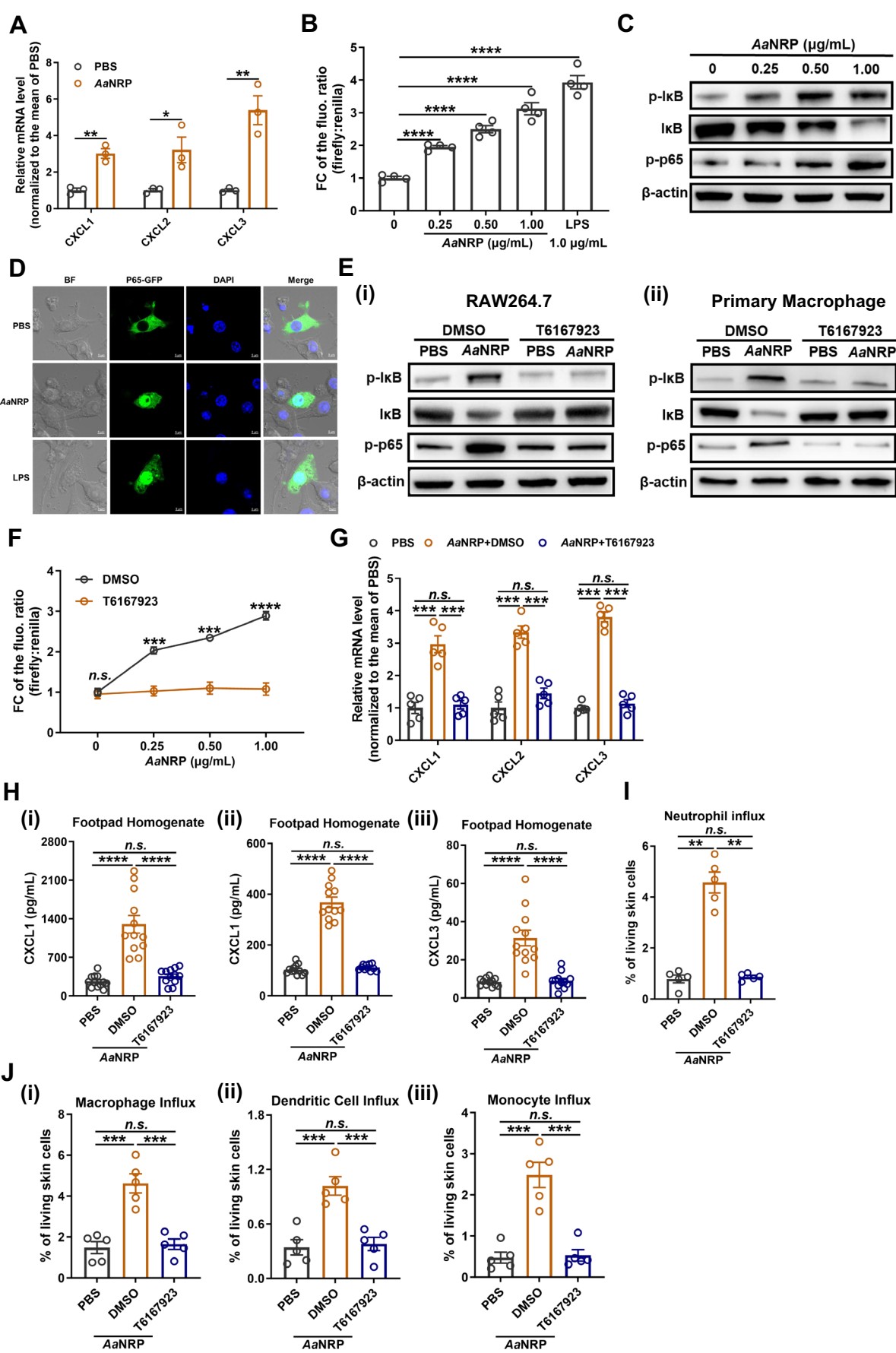

◀ **Figure 3.  *Aa*NRP induces neutrophil recruiting chemokines from macrophages by activating MyD88-mediated NF-κB signaling.**

(A) Induction of CXCL1/2/3 in RAW264.7 cells by *Aa*NRP. RAW264.7 cells were incubated with 1 μg/mL *Aa*NRP for 4 h. Subsequently, total RNA was extracted, and the RNA expression levels of CXCL1/2/3 were quantified by qPCR. The relative RNA abundance of CXCL1, CXCL2, and CXCL3 was normalized to the mean of the PBS group ($2^{-\Delta\Delta Ct}$). (B) Dual luciferase reporter assay demonstrating the activation of NF-κB signaling by *Aa*NRP. NF-κB and *Renilla* reporter plasmids (19:1) were cotransfected into RAW264.7 cells. Twelve hours later, RAW264.7 cells were treated with 0, 0.25, 0.50, or 1.00 μg/mL *Aa*NRP and 1.00 μg/mL LPS (positive control) for 4 h. Dual luciferase activity was detected on a Thermo Varioskan Flash platform. The relative NF-κB luciferase activity was calculated by normalizing the activity of firefly luciferase against that of *Renilla*. The fold change (FC) of the relative NF-κB luciferase activity for each group was calculated by normalization to the mean of the control group, which was treated with 0 μg/mL *Aa*NRP. (C) Dose-dependent activation of the MyD88/NF-κB signaling pathway by *Aa*NRP. RAW264.7 cells were treated with 0, 0.25, 0.50, or 1.00 μg/mL *Aa*NRP for 4 h. Subsequently, the cells were collected and subjected to an immunoblotting assay to measure the protein abundance of some crucial signaling molecules in the MyD88/NF-κB signaling pathway. (D) NF-κB p65 nuclear translocation assay. The NF-κB p65-EGFP plasmid was transfected into RAW264.7 cells, and 12 h later, RAW264.7 cells were treated with 1.0 μg/mL *Aa*NRP and 1.0 μg/mL LPS as a positive control. At 4 h posttreatment, the RAW264.7 cells were fixed with 4% PFA. The translocation of NF-κB p65-EGFP was observed under a Zeiss LSM880 confocal microscope. (E) Immunoblotting assay demonstrating the inhibition of the *Aa*NRP-activated NF-κB signaling pathway by T6167923. RAW264.7 cells (i) and murine peritoneal macrophages (ii) were preincubated with 4 μg/mL T6167923 or an equal volume of DMSO as a negative control for 30 min. Then, the cells were further treated with PBS or 1.00 μg/mL *Aa*NRP in the presence of T6167923 (4 μg/mL) or DMSO for 4 h. Finally, the cells were collected and subjected to an immunoblotting assay to measure the protein abundance of some crucial signaling molecules in the MyD88/NF-κB signaling pathway. (F) Dual luciferase assay demonstrating the inhibition of *Aa*NRP-activated NF-κB signaling by T6167923. RAW264.7 cells were cotransfected with NF-κB and *Renilla* luciferase reporter plasmids (19:1). At 12 h posttransfection, the cells were preincubated with DMSO or 4 μg/mL T6167923 for 30 min, followed by treatment with 0, 0.25, 0.50, or 1.00 μg/mL *Aa*NRP in the presence of T6167923 (4 μg/mL) or DMSO for 4 h. Dual luciferase activity was detected, and the fold change (FC) of the relative NF-κB luciferase activity for each group was calculated by normalization to the mean of the control group, which was treated with 0 μg/mL *Aa*NRP. (G) Inhibition of *Aa*NRP-mediated CXCL1/2/3 induction by T6167923 in RAW264.7 cells. RAW264.7 cells were preincubated with DMSO or 4 μg/mL T6167923 for 30 min. Afterward, the cells were treated with PBS or 1.0 μg/mL *Aa*NRP in the presence of T6167923 (4 μg/mL) or DMSO for 4 h. Finally, total RNA was extracted to quantify the mRNA levels of CXCL1/2/3 by qPCR. The relative RNA abundance was normalized to the mean of the PBS group ($2^{-\Delta\Delta Ct}$). (H) Inhibition of *Aa*NRP-mediated CXCL1/2/3 induction by T6167923 in murine footpad skin. 129 Sv/Ev mice were intraperitoneally injected with 0.5 mg T6167923 or DMSO as the control. One hour later, the mice with T6167923 or DMSO injection were intradermally injected with 100 ng *Aa*NRP, while the control mice were intradermally injected with PBS in the hind footpads. Four hours later, the mouse footpads were collected and homogenized in 500 μL of 1× cell lysis buffer. The supernatants were used for ELISA to measure the contents of CXCL1/2/3. (I, J) Inhibition of the *Aa*NRP-induced influx of neutrophils (I), macrophages (J, i), dendritic cells (J, ii), and monocytes (J, iii) by T6167923. 129 Sv/Ev mice were intraperitoneally injected with 0.5 mg of T6167923 or DMSO (control). One hour later, the mice were intradermally inoculated with PBS or 100 ng *Aa*NRP in the hind footpads. The mouse footpads were collected at 4 hpi for flow cytometric analysis of neutrophil influx and at 24 hpi for macrophages, dendritic cells, and monocytes. (A, B, G) Data are expressed as the mean ± SEM, and each dot represents an independent replicate. The one-way ANOVA and multiple *t* tests were used for statistical analyses. (F) Data are expressed as the mean ± SEM, with $n = 4$ for each group and each concentration. The two-way ANOVA and multiple *t* tests were used for statistical analyses. (H–J) Data are expressed as the mean ± SEM, and each dot represents a mouse. The one-way ANOVA and multiple *t* tests were used for statistical analyses. All experiments were reproduced at least twice. *$p < 0.05$, **$p < 0.01$, ***$p < 0.001$, ****$p < 0.0001$, n.s. not significant. Source data are available online for this figure.

pathway. Furthermore, in an in vivo situation, *Aa*NRP significantly activated the canonical NF-κB signaling axis in mouse footpad skin (Fig. EV3G). In RAW264.7 cells transfected with a plasmid expressing a p65-GFP fusion protein, *Aa*NRP activated obvious translocation of p65-GFP into the cell nucleus, as visualized in Fig. 3D. A small molecule MyD88 signaling inhibitor, T6167923 (Olson et al, 2015), fully abolished the *Aa*NRP-mediated activation of NF-κB signaling, as assessed by an immunoblotting assay (Fig. 3E, i) and a luciferase reporter assay (Fig. 3F). Consistently, T6167923 also offset the *Aa*NRP-mediated induction of CXCL1/2/3 in RAW264.7 cells (Fig. 3G). In addition, we also verified the activation of MyD88-NF-κB signaling by *Aa*NRP in primary murine peritoneal macrophages (Fig. 3E, ii). We further investigated whether blockade of the MyD88 signaling axis may prevent the infiltration of neutrophils and the subsequent recruitment of myeloid cells in vivo. The mouse footpads were inoculated with 100 ng *Aa*NRP with or without T6167923 pre-administration. Treatment with T6167923 largely impaired the production of CXCL1/2/3 induced by *Aa*NRP at 4 hpi (Fig. 3H) and reduced the influx of neutrophils induced by *Aa*NRP at 4 hpi (Fig. 3I) and the subsequent recruitment of myeloid cells at 24 hpi (Fig. 3J). Altogether, these results indicated that *Aa*NRP activates MyD88-dependent canonical NF-κB signaling, which drives the expression of neutrophil chemoattractants and the subsequent recruitment of myeloid cells.

Cell surface Toll-like receptors (TLRs) recognize extracellular factors, such as pathogen- and danger-associated molecular patterns (PAMPs and DAMPs), and activate the MyD88-NF-κB

signaling axis (Takeda et al, 2003). We therefore assessed whether *Aa*NRP directly interacts with the extracellular domain (ECD) of MyD88-dependent TLRs on the cell membrane. We expressed Myc-tagged ECDs of cell surface TLRs (mouse TLR1/2/4/5/6/10 and human TLR1/2/4/5/6/11/12) in HEK293T cells and coimmunoprecipitated them with V5-tagged *Aa*NRP using an anti-Myc antibody. Notably, *Aa*NRP was strongly coimmunoprecipitated with the ECDs of both murine and human TLR1 and TLR4 but not with TLR2/5/6/10/11/12 (Fig. 4A). The binding of *Aa*NRP with murine and human TLR1 and TLR4 was further corroborated by a flow cytometric assay (Fig. 4B,C). The anti-*Aa*NRP antibody efficiently labeled *Aa*NRP-transfected 293T cells with only a few background labeling of the vector-transfected cells, suggesting the specificity of this antibody (Appendix Fig. S3A,B). These results suggest that TLR1 and TLR4 are the sensors of *Aa*NRP. Indeed, pharmacological inhibition of TLR1 and TLR4 by a mixture of TLR1/4 antagonists (Anti-TLR-MIX), including CU-CPT22 against TLR-1 (Cheng et al, 2012) and TLR4-IN-C34 against TLR-4 (Adegoke et al, 2019; Zhang et al, 2022), completely abolished the *Aa*NRP-activated NF-κB signaling in RAW264.7 cells (Fig. 4D) and *Aa*NRP-stimulated CXCL1/2/3 expression in these cells (Fig. 4E). Moreover, knockdown of TLR1 and TLR4 by small interfering RNAs (siRNAs) (Fig. 4F) consistently aborted the *Aa*NRP-activated NF-κB signaling (Fig. 4G) and the subsequent production of CXCL1/2/3 (Fig. 4H) in RAW264.7 cells. Overall, these results indicate that *Aa*NRP induces expression of these chemoattractants in skin resident macrophages by activating the TLR1/4-MyD88-NF-κB signaling axis. This results in an influx of

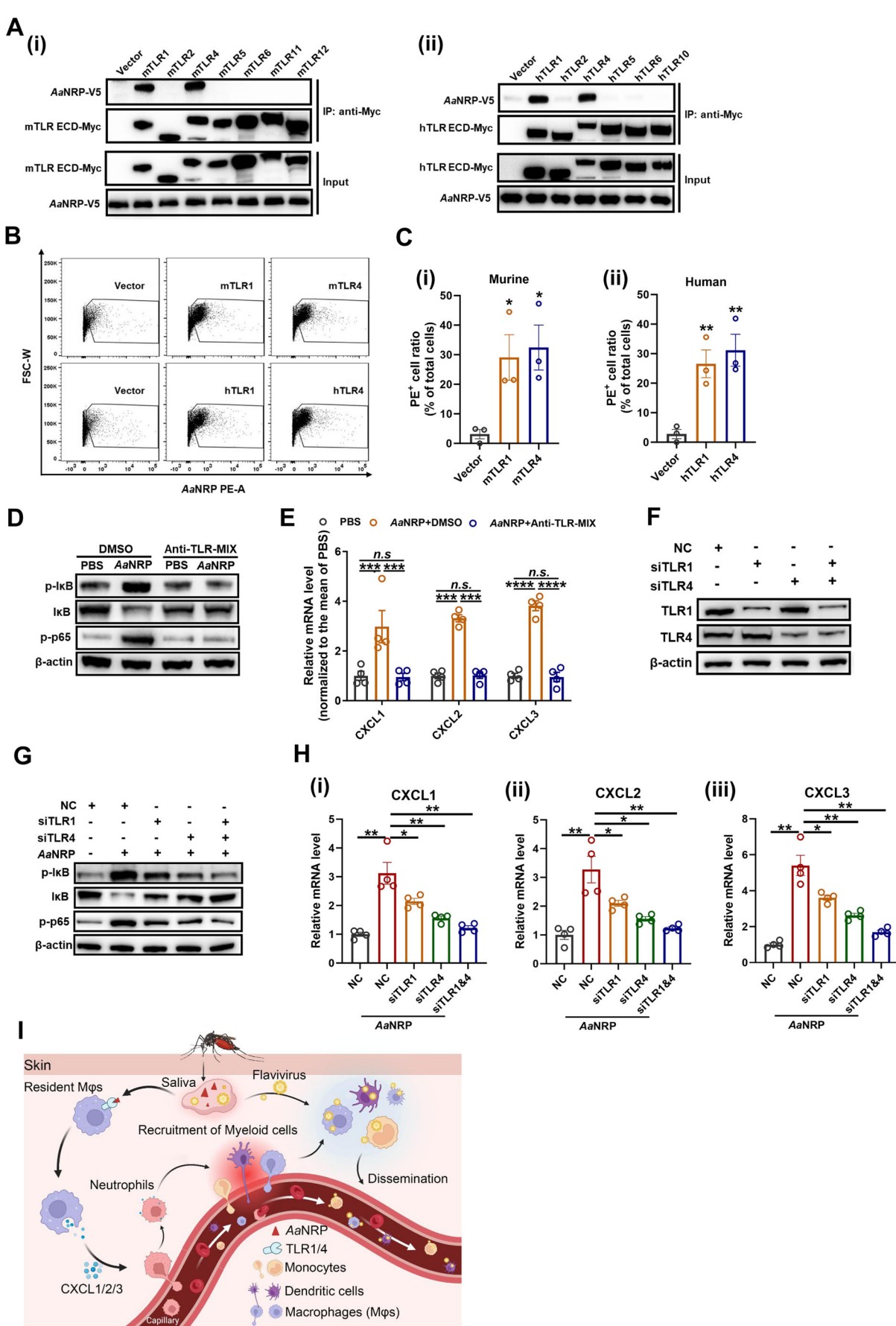

**Figure 4.   *Aa*NRP elicits CXCL1/2/3 through activating TLR1/4-MyD88-NF-κB signaling.**

(A) Coimmunoprecipitation assay showing the interaction of *Aa*NRP with murine (i) and human (ii) plasma membrane TLRs. HEK293T cells were transfected with murine Toll-like receptor 1 (mTLR1), mTLR2, mTLR4, mTLR5, mTLR6, mTLR11, and mTLR12 extracellular domains (ECDs) or human TLR1 (hTLR1), hTLR2, hTLR4, hTLR5, hTLR6, and hTLR10 ECDs. Forty-eight hours later, the transfected cells were lysed and incubated with 80 ng of purified *Aa*NRP at 4 °C for 2 h, followed by incubation with 10 μL of Myc tag antibody against TLR ECDs for another 2 h. Finally, the samples were subjected to an immunoblotting assay to detect TLRs and *Aa*NRP coimmunoprecipitated with TLRs. (B, C) The flow cytometric validation of the binding of *Aa*NRP to human and murine TLR1 and 4. HEK293T cells were transfected with human and mouse TLR1 and 4 or vector plasmids. After 48 h, the cells were incubated with *Aa*NRP (1 μg/$10^6$ cells) at 4 °C for 2 h. Then, the cells were incubated with anti-*Aa*NRP polyclonal antibody and PE-labeled secondary antibody at 4 °C for 1 h and 30 min, respectively. The binding of *Aa*NRP to TLRs was determined by detecting the proportion of PE$^+$ cells using flow cytometry. (B) Dot-plots showing the cells that were engaged with *Aa*NRP, and these cells were gated as PE$^+$ cells. (C) Statistic analysis of the ratio of PE$^+$ cells with murine (i), human (ii) TLRs or vector plasmid transfection. The neat PE$^+$ cell ratio of each sample was obtained by deducting its value from the mean PE$^+$ ratio of the vector controls. (D, E) The influence of TLR1 and TLR4 dual inhibition by Anti-TLR-MIX on *Aa*NRP-mediated MyD88/NF-κB signaling (D) and CXCL1/2/3 induction (E). RAW264.7 cells were preincubated with DMSO or 10 μM Anti-TLR-MIX (CU-CPT22 and TLR4-IN-C34, each 10 μM) for 30 min, followed by treatment with PBS or 1 μg/mL *Aa*NRP in the presence of Anti-TLR-MIX or DMSO for 4 h. The cells were then collected and subjected to immunoblotting assay to measure the protein abundance of some crucial signaling molecules in the MyD88/NF-κB signaling pathway (D) and to qPCR assay to measure the relative RNA abundance of CXCL1/2/3, which was normalized to the mean of the PBS group (E). (F) Silencing of TLR1 and TLR4 in RAW264.7 cells by small interfering RNAs (siRNAs). RAW264.7 cells were transfected with negative control (NC) siRNA or TLR1/4 siRNA. Forty-eight hours later, the cells were collected for measuring the expression of TLR1/4 by an immunoblotting assay. (G, H) Influence of TLR1/4 silencing on the *Aa*NRP-activated NF-κB signaling (G) and the subsequent induction of CXCL1/2/3 (H). RAW264.7 cells were transfected with negative control siRNA or TLR1/4 siRNA. Forty-eight hours later, these cells were further incubated with *Aa*NRP (1 μg/mL) for 4 h. Then, the cells were collected for measuring the expression of some crucial signal molecules in MyD88/NF-κB signaling pathway by an immunoblotting assay and for measuring the expression of CXCL1/2/3 by qPCR (The relative RNA abundance was normalized to the mean of the PBS-NC group). (I) A schematic graph delineating the mechanism underlying the recruitment of myeloid cells by *Aa*NRP. (C, E, H) Data are expressed as the mean ± SEM, and each dot represents an independent replicate. The one-way ANOVA and multiple *t* tests were used for statistical analyses. All experiments were reproduced at least twice. *$p < 0.05$, **$p < 0.01$, ***$p < 0.001$, ****$p < 0.0001$, n.s. not significant. Source data are available online for this figure.

neutrophils, which amplifies a localized inflammatory response to subsequently recruit flavivirus-permissive myeloid cells to mosquito bites (Fig. 4I).

## The *Aa*NRP-mediated influx of myeloid cells promotes flaviviral transmission by mosquitoes

We next assessed the role of *Aa*NRP in flaviviral infection in a type I interferon receptor-deficient (*ifnar*$^{-/-}$) 129 (A129) mouse model (Ma et al, 2016; Yu et al, 2021). We intraperitoneally inoculated 0.5 mg of the MyD88 signaling inhibitor T6167923 or DMSO into A129 mice. One hour later, we intradermally inoculated 100 ng of purified *Aa*NRP with 250 PFU of ZIKV into the footpads of animals treated with T6167923 or DMSO. The mice infected by ZIKV together with PBS served as the negative controls (Fig. 5A). Cutaneous myeloid cells are major targets of flavivirus infection (Castanha et al, 2020; Chen et al, 1999; Halstead et al, 1977; Kou et al, 2008; Neves-Souza et al, 2005; Pingen et al, 2016; Schaeffer et al, 2015; Wu et al, 2000). The proportions of cutaneous myeloid cells, such as macrophages, immature DCs and monocytes, increased in the *Aa*NRP-treated footpads at both 6 hpi (Fig. 5B) and 24 hpi (Fig. EV4A), as did the ratios of ZIKV-positive myeloid cells (Figs. 5C and EV4B). However, the MyD88 signaling inhibitor T6167923 completely abolished the effect of *Aa*NRP on myeloid cell recruitment (Figs. 5B and EV4A) and the ratio of ZIKV-positive myeloid cells in mouse footpads (Figs. 5C and EV4B). Inoculation of T6167923 without *Aa*NRP did not influence myeloid cell influx (Fig. 5B) or ZIKV infection of these cells (Fig. 5C). Nonetheless, the ZIKV-positive myeloid cells may not lead to release of infectious viruses and could also represented phagocytised viral debris. An in vitro culture experiment showed that the primary CD11b$^+$ leukocytes from the footpad skin coinoculated with ZIKV and *Aa*NRP were able to produce infectious progenies by a focus forming assay (Fig. 5D). Consistently, *Aa*NRP inoculation augmented ZIKV viremia (Fig. 5E), viral loads in the footpads (Fig. 5F), lymph nodes (Fig. 5G), and spleens (Fig. 5H) and accelerated animal death (Fig. 5I). These *Aa*NRP enhanced

effects were completely reversed by T6167923 (Fig. 5E–I). Inoculation of T6167923 without *Aa*NRP did not influence ZIKV infection (Fig. 5E–I). These phenomena were recapitulated with dengue virus-2 (DENV-2) infection of A129 mice (Fig. EV4C–H), indicating that *Aa*NRP-mediated myeloid cell recruitment promotes flavivirus infection and pathogenesis in hosts.

Next, we assessed the role of *Aa*NRP in flaviviral transmission using a "mosquito-A129 mouse" viral transmission model (Fig. 6A). The *Aa*NRP gene in *A. aegypti* was silenced by *Aa*NRP dsRNA, and mosquitoes inoculated with *GFP* dsRNA served as a negative control. The *Aa*NRP gene was efficiently knocked down in the *A. aegypti* salivary glands (Fig. 6B). Three days after administration of dsRNA, the mosquitoes were intrathoracically infected with ZIKV. At 8 days after infection, these mosquitoes were allowed to bite A129 mice. Silencing the *Aa*NRP gene did not influence the mosquito's probing time, efficiency of blood engorgement, and capacity for salivation (Appendix Fig. S4A–C). Saliva from individual mosquitoes was collected by forced salivation (Miller et al, 2021) at 8 days after infection. Silencing of the *Aa*NRP gene did not influence either ZIKV loads in the mosquito salivary glands (Fig. 6C) or the amount of infectious ZIKV particles in the mosquito saliva (Fig. 6D). Notably, the animals bitten by the *Aa*NRP-silenced mosquitoes had much lower viremia (Fig. 6E) and viral loads in various tissues (Fig. 6F–H), thus resulting in longer survival (Fig. 6I) than the animals bitten by control mosquitoes (*GFP* dsRNA), indicating that *Aa*NRP plays a key role in flavivirus transmission by mosquitoes.

## Dietary supplementation with an anti-inflammatory phytochemical prevents *Aa*NRP-promoted ZIKV transmission by mosquitoes

Accumulating evidence indicates that some phytochemicals, such as resveratrol (trans-3,5,4-trihydroxystilbene), exhibit potent anti-inflammatory effects by attenuating the MyD88-NF-κB signaling axis (Rahimifard et al, 2017; Ren et al, 2013; Wang et al, 2019b). Supplementation with resveratrol can prevent LPS-induced TLR4-

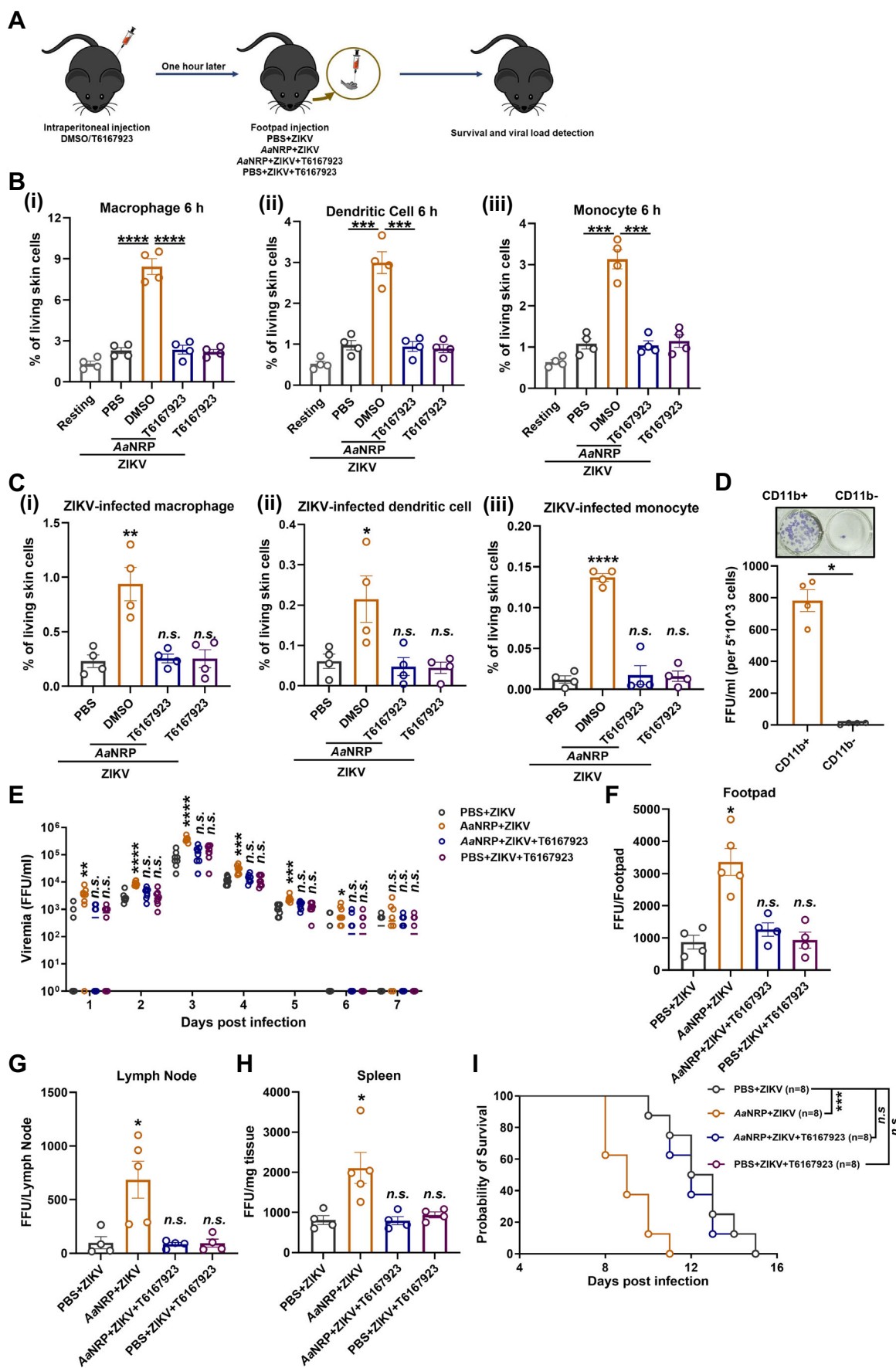

**Figure 5.** ***Aa*NRP-mediated myeloid cell influx promotes ZIKV infection.**

(A) Schematic delineation of the animal study design. Six-week-old female type I interferon receptor-deficient (*ifnar*$^{-/-}$) 129 (A129) mice were intraperitoneally injected with 0.5 mg T6167923 or DMSO (solvent control). One hour later, the mice were intradermally inoculated with 100 ng *Aa*NRP plus 250 PFU ZIKV (Group: *Aa*NRP + ZIKV and Group: *Aa*NRP + ZIKV + T6167923) or PBS plus 250 PFU ZIKV (Group: PBS + ZIKV and Group: PBS + ZIKV + T6167923) in their footpads. (B) Flow cytometric analysis of macrophages (CD45$^+$CD11b$^+$F4/80$^+$, i), dendritic cells (CD45$^+$I-A/I-E$^+$CD11c$^+$, ii), and monocytes (CD45$^+$CD11b$^+$Ly6C$^+$Ly6G$^-$, iii) in murine footpad skin at 6 hpi. (C) Flow cytometric analysis of ZIKV-infected macrophages (CD45$^+$CD11b$^+$F4/80$^+$ZIKV-E$^+$, i), dendritic cells (CD45$^+$I-A/I-E$^+$CD11c$^+$ZIKV-E$^+$, ii), and monocytes (CD45$^+$CD11b$^+$Ly6C$^+$Ly6G$^-$ZIKV-E$^+$, iii) in murine footpad skin at 24 hpi. (D) Determination of the infectious progeny viruses released by in situ CD11b$^+$ myeloid cells and CD11b$^-$ none-myeloid cells in murine footpad skin coinoculated with ZIKV and *Aa*NRP at 16 hpi. CD11b$^+$ myeloid cells and CD11b$^-$ non-myeloid cells from murine skin injected with *Aa*NRP + ZIKV were isolated by flow cytometry at 16 hpi. The CD11b$^+$ and CD11b$^-$ cells were in vitro cultured in equal number of cells (5 × 10$^3$ per well). Six hours later, the supernatants were collected and the viral titers in the supernatants were determined by focus forming unit (FFU) assay. (E) ZIKV loads in murine peripheral blood plasma. Murine tail blood plasma was taken from Day 1 through Day 7 postinfection for viral load detection by FFU assay. The ZIKV loads were shown as FFU/ml blood plasma. (F–H) ZIKV loads in murine footpad (F), lymph node (G), and spleen tissue (H) at 48 hpi. ZIKV loads were measured by FFU assay and shown as FFU/footpad, FFU/lymph node, or FFU/mg spleen tissue. (I) Survival curves of ZIKV-infected mice in Group: PBS + ZIKV (*n* = 8), Group: *Aa*NRP + ZIKV (*n* = 8), Group: *Aa*NRP + ZIKV + T6167923 (*n* = 8), and Group: PBS + ZIKV + T6167923 (*n* = 8). (B–D and F–H) Data are expressed as the mean ± SEM, and each dot represents an individual mouse. The one-way ANOVA and multiple t tests were used for statistical analyses. (E) Data are expressed as the median and each dot represents an individual mouse. The two-way ANOVA and Mann–Whitney tests were used for statistical analyses. (I) The log-rank test was used to compare the survival status. All experiments were reproduced at least twice. *$p < 0.05$, **$p < 0.01$, ***$p < 0.001$, ****$p < 0.0001$, n.s. not significant. Source data are available online for this figure.

MyD88-NF-κB axis activation and subsequent leukocyte migration under both in vitro and in vivo conditions (Wang et al, 2020a). We therefore assessed whether dietary supplementation with resveratrol might prevent ZIKV infection by suppressing *Aa*NRP-mediated cutaneous neutrophil infiltration and subsequent recruitment of myeloid cells. A serial concentration of resveratrol was supplemented daily to A129 mice for 14 days. Dietary administration of resveratrol did not regulate the frequency of myeloid cells in resting skin or blood (Appendix Fig. S5A,B). Oral treatment with resveratrol did not modulate the expression of most of the antiviral immune genes in murine footpad skin, lymph nodes or spleen (Appendix Table S3). Subsequently, 100 ng of purified *Aa*NRP together with 250 PFU of ZIKV was intradermally inoculated into the footpads of A129 mice with or without resveratrol supplementation (Fig. 7A). The mice coinoculated with ZIKV and PBS served as a mock control, while the mice coinoculated with ZIKV and LPS in footpad skin served as a positive control for specific activation of TLR4/MyD88/NF-κB signaling. When compared to PBS, both *Aa*NRP and LPS enhanced Iκ-B phosphorylation and degradation and promoted p65 phosphorylation, while resveratrol offset these effects of *Aa*NRP and LPS in a dose-dependent manner in the footpads of A129 mice (Fig. 7B; Appendix Fig. S6), indicating that supplementation with this anti-inflammatory phytochemical prevented the *Aa*NRP-mediated activation of the MyD88-NF-κB signaling axis. Furthermore, when compared to PBS, *Aa*NRP promoted CXCL1/2/3 expression (Fig. 7C) and infiltration of inflammatory neutrophils (Fig. 7D) and flavivirus-permissive myeloid cells (Fig. 7E) toward the inoculation sites of the footpads, enhanced viremia (Fig. 7F) and viral loads in various tissues (Fig. EV5A–C), and accelerated the death of ZIKV-infected A129 mice (Fig. 7G). However, all these *Aa*NRP effects were reversed by resveratrol (Fig. 7C–G and Fig. EV5A–C). In the "mosquito-A129 mouse" ZIKV transmission model, A129 mice with or without pre-supplementation with resveratrol were allowed to be bitten by ZIKV-infected *A. aegypti* (Fig. 7H). After being bitten by ZIKV-infected mosquitoes, the animals with dietary resveratrol supplementation had much lower viremia (Fig. 7I) and viral loads in various tissues (Fig. EV5D–F). The animals survived longer (Fig. 7J) than the animals without resveratrol supplementation. Overall, these results indicate that dietary supplementation with anti-inflammatory phytochemical prevents the disease sequelae of flavivirus infection by mosquito bites.

## Discussion

Intradermal myeloid cells, such as dendritic cell subsets, monocytes and macrophages, are regarded highly permissive to the initial replication of human flaviviruses (Castanha et al, 2020; Chen et al, 1999; Halstead et al, 1977; Kou et al, 2008; Neves-Souza et al, 2005; Pingen et al, 2016; Schaeffer et al, 2015; Schmid and Harris, 2014; Wu et al, 2000). Intriguingly, mosquito bites can trigger an influx of inflammatory neutrophils to facilitate infection by recruiting myeloid cells as cellular targets for infection (Guerrero et al, 2022; Lefteri et al, 2022; Pingen et al, 2016). In this process, mosquitoes are not merely involuntary carriers of viruses, instead, with numerous salivary proteins they proactively aid viral replication and dissemination (Manning et al, 2018; Pingen et al, 2017). For example, humanized mice infected with DENV via mosquito biting showed much higher sustained DENV viremia and exacerbated disease than those infected via injection (Cox et al, 2012). *A. aegypti* mosquito probing immediately prior to intradermal inoculation of DENV enhanced DENV viremia in mice (McCracken et al, 2014). Compared with mice inoculated with West Nile virus (WNV) alone, mice inoculated with WNV mixed with mosquito salivary gland extract developed higher viremia, more severe disease and higher mortality (Styer et al, 2011). These studies suggest that mosquito salivary components facilitate viral infection. Indeed, an *A. aegypti* salivary 34-kDa protein suppresses interferon signaling to enhance DENV replication in human keratinocytes (Surasombatpattana et al, 2014). The *A. aegypti* saliva-specific protein venom allergen-1 (*Aa*VA-1) promotes DENV and ZIKV transmission by activating autophagy in host skin myeloid cells (Sun et al, 2020). A serine protease called CLIPA3 in *A. aegypti* saliva facilitates DENV infection by proteolyzing extracellular matrix proteins, which benefits viral attachment and cell migration (Conway et al, 2014). Herein, we identified a female *A. aegypti*-specific salivary protein, *Aa*NRP, as a potent immune modulator and facilitator of viral dissemination. *Aa*NRP stimulates a rapid influx of neutrophils followed by DENV/ZIKV-permissive myeloid cells into mosquito bites. Consistent with our results, a recent study showed that therapeutic depletion of neutrophils suppressed the mosquito saliva-mediated enhancement of virus infection (Pingen et al, 2016). *A. aegypti* bacteria-responsive protein 1 (*Ag*BR1) induces early neutrophil-mediated inflammation in the skin of bitten mice, thus facilitating ZIKV transmission (Uraki et al, 2019). However, *Aa*NRP does not directly recruit neutrophils; rather, it activates skin resident

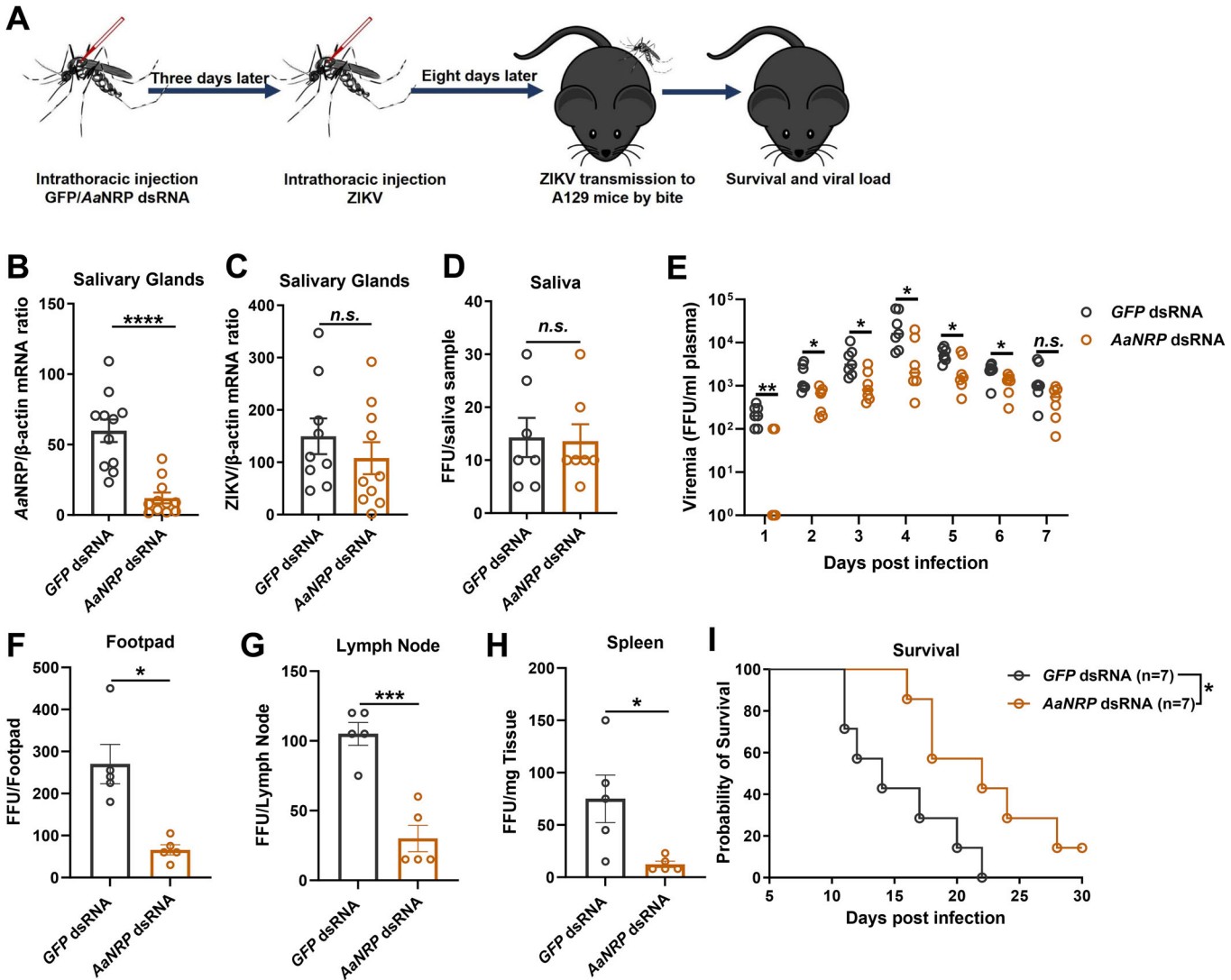

**Figure 6. Silencing mosquito salivary AaNRP inhibits mosquito-borne ZIKV transmission.**

(A) Schematic of the study design. *A. aegypti* mosquitoes were intrathoracically injected with 1 µg/300 nL *AaNRP* dsRNA or *GFP* dsRNA as negative controls. Three days later, the mosquitoes were intrathoracically injected with ZIKV (10 PFU) and maintained for eight days to allow virus infection of the salivary glands. Finally, six-week-old A129 mice were bitten by ZIKV-infected mosquitoes (each mouse was bitten by three mosquitoes). (B, C) RNA expression levels of *AaNRP* (B) and ZIKV envelope protein (C) in the salivary glands of *AaNRP* dsRNA- and *GFP* dsRNA-treated mosquitoes at 8 days post viral inoculation (dpi). (D) Viral load in the saliva of *AaNRP* dsRNA- and *GFP* dsRNA-treated mosquitoes at 8 dpi. The saliva was collected by forced salivation assay and the viral loads were determined by FFU assay. (E) ZIKV loads in murine peripheral blood plasma. Murine tail blood plasma was taken from Day 1 through Day 7 postinfection for viral load detection by FFU assay. The ZIKV loads were shown as FFU/ml blood plasma. (F–H) ZIKV loads in murine footpad (F), lymph node (G), and spleen tissue (H) at 48 hpi. ZIKV loads were measured by FFU assay and shown as FFU/footpad, FFU/lymph node, or FFU/mg spleen tissue. (I) Survival curves of the mice ($n = 7$) bitten by ZIKV-infected *AaNRP*-silenced mosquitoes and the mice ($n = 7$) bitten by ZIKV-infected *GFP* control mosquitoes. (B–D, F–H) Data are expressed as the mean ± SEM, and each dot represents an individual animal. The unpaired t test was used for the statistical analysis. (E) Data are expressed as the median and each dot represents an individual mouse. The two-way ANOVA and Mann–Whitney tests were used for statistical analyses. (I) The log-rank test was used to compare the survival status. All experiments were reproduced at least twice. *$p < 0.05$, **$p < 0.01$, ***$p < 0.001$, ****$p < 0.0001$, n.s. not significant. Source data are available online for this figure.

macrophages to release chemoattractants for neutrophils via the TLR1/4-MyD88-NF-κB axis. In contrast to the above salivary factors known to facilitate viral transmission by mosquitoes, *Aa*NRP directly interacts with the extracellular ligand binding domains of TLR1 and TLR4 and activates its downstream signaling, suggesting that *Aa*NRP acts as a ligand of both TLR1 and TLR4. Nonetheless, previous study indicated that chemokine expression alone is not sufficient to mediate saliva-induced enhancement of arbovirus infection (Lefteri et al, 2022),

and emphasized that induction of chemokines is one of the essential steps in the saliva enhancement of virus infection. Overall, uncovering the roles of mosquito salivary components in arboviral transmission could provide new potential therapeutic targets.

Phytochemicals are potent therapeutics for the treatment of inflammatory diseases. Polyphenols, such as resveratrol, can suppress NF-κB signaling in multiple ways (Rahimifard et al, 2017). Resveratrol, a polyphenol found in red grapes and in several

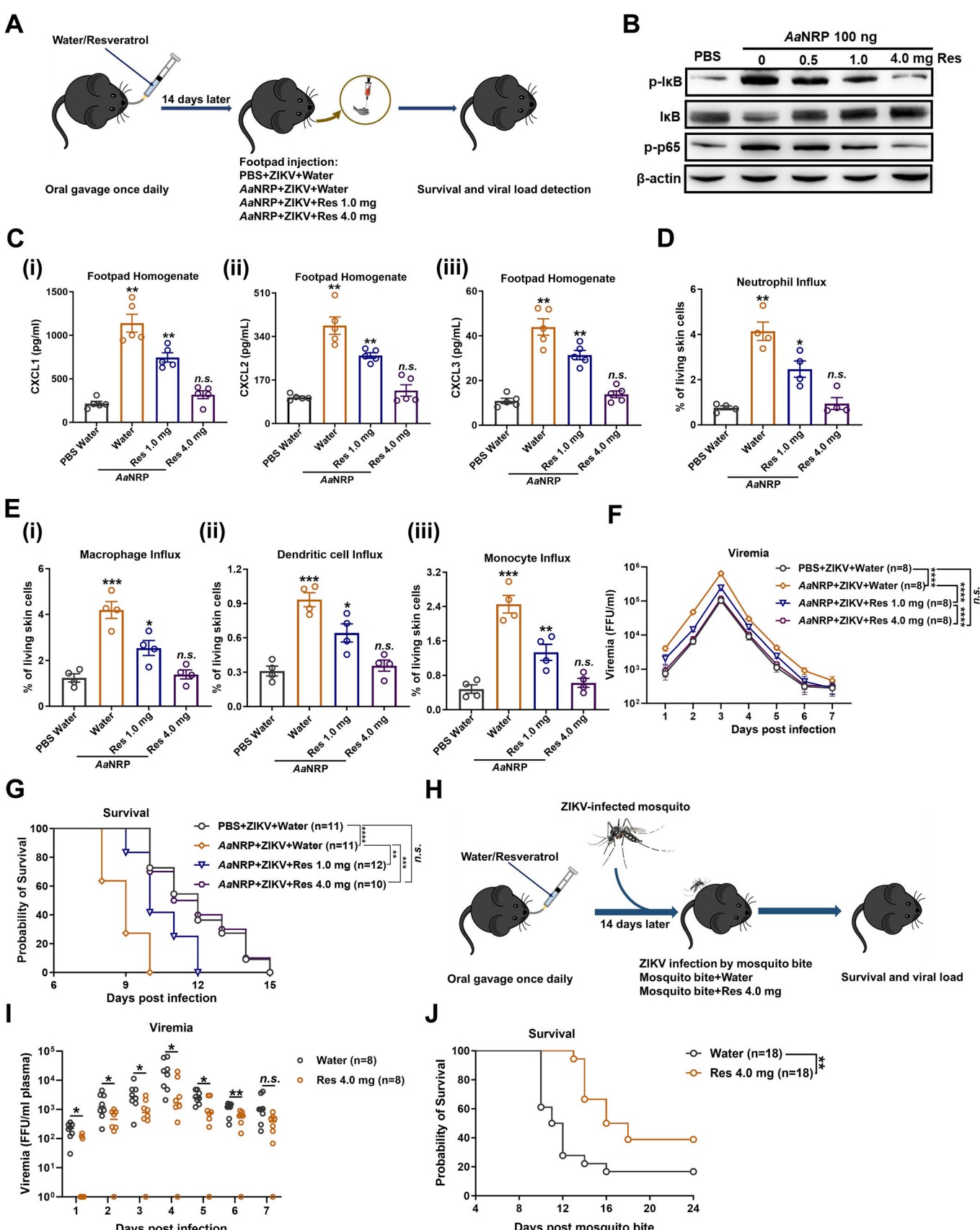

**Figure 7.  Dietary supplementation with resveratrol attenuates *Aa*NRP-enhanced ZIKV infection.**

(A) Schematic of the study design for the resveratrol supplementation experiment. Six-week-old A129 mice were orally administered water, 1.0 mg or 4.0 mg resveratrol (Res) once daily for 14 days. Afterward, A129 mice were intradermally injected with ZIKV (250 PFU) + PBS or ZIKV (250 PFU) + *Aa*NRP (100 ng) immediately after the last dose of resveratrol. (B) Immunoblotting assay showing the attenuating effect of resveratrol on *Aa*NRP-mediated activation of MyD88/NF-κB signaling in mouse footpad skin. Six-week-old A129 mice were orally administered water or 0.5 mg, 1.0 mg, or 4.0 mg resveratrol (Res) once daily for 14 days. Afterward, the resveratrol-administered mice were intradermally injected with 100 ng *Aa*NRP immediately after the last dose of resveratrol, while half of the water-administered mice were injected with PBS as the negative control, and the remaining half were injected with 100 ng *Aa*NRP as the positive control. Four hours later, the mouse footpad skin was collected for an immunoblotting assay to detect the protein abundance of crucial signaling molecules in the MyD88/NF-κB signaling pathway. (C) Determination of CXCL1 (i), CXCL2 (ii), and CXCL3 (iii) contents in mouse footpad abrasive fluid. Six-week-old A129 mice were orally administered water, 1.0 mg or 4.0 mg resveratrol (Res) once daily for 14 days. Afterward, the resveratrol-administered mice were injected intradermally with 100 ng *Aa*NRP + 250 PFU ZIKV immediately after the last dose of resveratrol, while half of the water-administered mice were injected with PBS + 250 PFU ZIKV as the negative control, and the remaining half were injected with 100 ng *Aa*NRP + 250 PFU ZIKV as the positive control. Four hours later, the mouse footpad skin was collected and homogenized in 500 μL of 1× cell lysis buffer. The supernatants were quantified for contents of CXCL1/2/3 by ELISA. (D, E) Flow cytometric analysis of neutrophils at 4 hpi (D), macrophages (E, i), dendritic cells (E, ii), and monocytes (E, iii) at 24 hpi in the mouse footpad skin. (F) ZIKV loads in murine peripheral blood plasma. Murine tail blood plasma was taken from Day 1 through Day 7 postinfection for viral load detection by FFU assay. The ZIKV loads were shown as FFU/ml blood plasma. (G) Survival curves of ZIKV-infected mice. Six-week-old A129 mice were orally administered water, 1.0 mg or 4.0 mg resveratrol (Res) once daily for 14 days. Afterward, the resveratrol-administered mice were intradermally injected with 100 ng *Aa*NRP + 250 PFU ZIKV immediately after the last dose of resveratrol, while half of the water-administered mice were intradermally injected with PBS + 250 PFU ZIKV as the negative control, and the remaining half were intradermally injected with 100 ng *Aa*NRP + 250 PFU ZIKV as the positive control. (H) Study design of the "mosquito-A129 mice" transmission-blocking effect of resveratrol. Six-week-old A129 mice were orally administered 4.0 mg resveratrol or an equivalent volume of water as the control once daily for 14 days. Afterward, each A129 mouse was bitten by three ZIKV-infected mosquitoes immediately after the last dose of resveratrol. (I) ZIKV loads in murine peripheral blood plasma. Murine tail blood plasma was taken from Day 1 through Day 7 postinfection for viral load detection by FFU assay. The ZIKV loads were shown as FFU/ml blood plasma. (J) Survival curves of mice bitten by ZIKV-infected mosquitoes receiving oral gavage of water ($n = 18$) and those receiving oral gavage of 4.0 mg resveratrol ($n = 18$). (C–F) Data are expressed as the mean ± SEM and each dot represents an individual mouse. The one-way ANOVA and multiple $t$ tests (C–E) or two-way ANOVA (F) were used for statistical analyses. (G, J) The log-rank test was used to compare survival status. (I) Data are expressed as the median and each dot represents an individual mouse. Two-way ANOVA and multiple $t$ tests were used for the statistical analyses. All experiments were reproduced at least twice. $*p < 0.05$, $**p < 0.01$, $***p < 0.001$, $****p < 0.0001$, n.s. not significant. Source data are available online for this figure.

other plant sources, has demonstrated chemopreventative effects and potent anti-inflammatory properties (Baur et al, 2006; Rahimifard et al, 2017; Wang et al, 2020a). Resveratrol was shown to inhibit the protein expression of TLR4, MyD88, and p-NF-κB while increasing the abundance of p-IκB in animal models of acute pharyngitis (Zhou et al, 2018). The therapeutic effects of resveratrol have been shown in several diseases, such as inflammatory bowel disease (Shi et al, 2017) and neuroinflammation (Huang et al, 2021), by downregulating the enzymes and cytokines belonging to NF-κB signaling (Rahimifard et al, 2017). In addition, resveratrol showed protective effects against LPS-induced acute lung injury, such as infiltration of inflammatory immune cells and alveolar structure damage of lungs in mice, by interrupting the TLR-MyD88-NF-κB signaling axis (Zhang et al, 2014). In this study, we demonstrated that supplementation with resveratrol interrupted the *Aa*NRP-mediated activation of MyD88-dependent NF-κB signaling in mouse skin. Dietary supplementation with this phytochemical blocked ZIKV infection by suppressing *Aa*NRP-mediated cutaneous neutrophil infiltration and subsequent recruitment of myeloid cells, thus preventing the disease sequelae of flavivirus infection by mosquito bites.

Some previous studies have suggested that resveratrol can increase DNA damage in pathogenic bacteria (Lee and Lee, 2017) and tumor cells (Qian et al, 2022). Although both of these effects are beneficial for host animals and resveratrol-increased DNA damage is usually observed in tumor cells rather than in normal cells (Hadi et al, 2010), resveratrol still has the potential to induce DNA damage in some types of normal cells under specific conditions (Hadi et al, 2010). Nonetheless, there are also many investigations demonstrating that resveratrol can prominently reduce DNA damage and other oxidative injuries induced by various physical, chemical or biological factors in multiple tissues (such as skin, liver and brain) (Afaq et al, 2003; Jin et al, 2023; Pal and Sarkar, 2014; Zemheri-Navruz et al, 2023) and different cell

types (such as macrophages, lymphocytes and epithelial cells) (Aydin et al, 2013; Dobrzynska and Gajowik, 2022; Moon et al, 2006; Quincozes-Santos et al, 2007; Zhang et al, 2020). Resveratrol, a polyphenolic compound classified as a nutraceutical, demonstrates a broad spectrum of biological activities in human subjects. The efficacy, safety, and pharmacokinetic properties of resveratrol have been extensively evaluated across more than 244 clinical trials, with an additional 27 trials presently underway (Singh et al, 2019). This compound is purported to enhance therapeutic outcomes for a wide array of human diseases, offering potential benefits in medical treatment (Singh et al, 2019). Furthermore, resveratrol has been deemed safe for consumption in dosages up to 5 grams per day, whether administered as a standalone agent or in conjunction with other therapies (Breuss et al, 2019; Singh et al, 2019). Nonetheless, we should be aware of the potential side effects of resveratrol. Long-term epidemiological surveillance is needed to assess the safety and prophylactic benefits to prevent flavivirus transmission.

There are several technical limitations in functional investigation of mosquito saliva. First, we exploited 100 ng of purified salivary proteins inoculated into a mouse footpad to assess the role of these proteins in triggering neutrophil influx. Indeed, the dosage for studying mosquito saliva proteins is varied in different studies (Jin et al, 2018; Martin-Martin et al, 2022; Uraki et al, 2019). Schmid et al reported a total protein amount of approximately 370–600 ng in the salivary gland extraction per individual mosquito (Schmid et al, 2016). Intriguingly, a mosquito bite might secrete ~0.7 protein amount of SGE (Wasserman et al, 2004), which equates to 250–420 ng of salivary proteins. Given the relatively high prevalence of selected components in mosquito saliva, we decided to inoculate 100 ng of a single mosquito saliva protein into one mouse footpad for our investigation. Second, the volume of recombinant protein injected into murine skin was 20 μl in our experimental setting. Indeed, the saliva volume expectorated by a mosquito is hard to quantify due to very small amounts and

individual differences. By using a microtube forced salivation assay, Sanchez-Vargas et al reported that a single *Aedes* mosquito salivates an average of $6.82 \pm 2.88$ nL saliva (Sanchez-Vargas et al, 2019). The injected volume in our experimental setting was much greater than this natural volume, which is a technical limitation of our study. Third, we exploited an inoculation of antibody against F4/80 to deplete skin macrophages. Accumulating evidence indicates that skin resident macrophages are derived from yolk sac progenitors and highly express F4/80 (Davies et al, 2013; Wynn et al, 2013). Nonetheless, the inoculation of F4/80 antibody may also deplete other skin immune cells e.g. Langerhans cells, since F4/80 is also expressed by some Langerhans cells (Davies et al, 2013; Kashem et al, 2017). Indeed, the scRNA-Seq data indicate that Langerhans cells show a relatively low prevalence in murine skin compared to that of skin macrophages, which is consistent with previous reports (Kashem et al, 2017). In addition, our the scRNA-Seq data showed that neutrophil chemoattractants such as CXCL1/2/3 were not induced in Langerhans cells after a mosquito bite. Thus, these results suggest that Langerhans cells may not be essential in *Aa*NRP-mediated neutrophil influx, even if they might be nonspecifically targeted by the F4/80 mAb.

Overall, this study mechanistically extends our understanding of the molecular basis of mosquito bites in flavivirus transmission, thus providing an avenue by which the development of therapeutics for preventing mosquito-borne viral diseases can be pursued in the future.

# Methods

## Ethics statement

The human blood used for peripheral neutrophil isolation was donated and collected at Tsinghua University Hospital by healthy volunteers who had given informed consent. The ethics committee at Tsinghua University approved the collection of human blood samples.

## Mice, mosquitoes, viruses, and cells

Six-week-old 129 Sv/Ev mice (wild type) were obtained from Beijing Vital River Corporation, while A129 mice (with loss of type I interferon receptor, *ifnar*$^{-/-}$) were provided by the Institute Pasteur of Shanghai, Chinese Academy of Sciences. The mice were bred and maintained in a specific pathogen-free animal facility at Tsinghua University, which is accredited by the Association for Assessment and Accreditation of Laboratory Animal Care International (AAALAC). All animal experiments were approved by the Institutional Animal Care and Use Committee of Tsinghua University and conducted in accordance with their guidelines (approval number: 22-CG3). Six-week-old female mice were first balanced by body weight and then randomly allocated to different groups for treatments. *A. aegypti* mosquitoes (Rockefeller strain) were reared at 28 °C and 80% humidity in a specially designed incubator (Model 818, Thermo Fisher) following standard rearing procedures (Liu et al, 2016). Zika Puerto Rico strain PRVABC59 (KU501215) and DENV-2 (New Guinea C strain, AF038403.1) were used. The viruses were passaged in *A. albopictus* C6/36 cells. The viral titers were determined by a plaque forming assay (Liu

et al, 2017). The *Drosophila melanogaster* S2 cell line was maintained in Schneider's medium supplemented with 10% heat-inactivated fetal bovine serum (Cat# 16000-044, Gibco) and 1% antibiotic-antimycotic (Cat# 15240-062, Invitrogen) at 28 °C. Vero and 293T cells were maintained in Dulbecco's modified Eagle's medium (Cat# 11965-092, Gibco) supplemented with 10% heat-inactivated fetal bovine serum at 37 °C. The murine macrophage cell line RAW264.7 was maintained in RPMI 1640 medium (Cat# 22400089, Gibco) supplemented with 10% heat-inactivated fetal bovine serum at 37 °C.

## Recombinant *A. aegypti* salivary protein purification

The *A. aegypti* salivary protein-coding genes with a score greater than 25 were cloned from the mass spectrometry analysis of the protein components of *A. aegypti* saliva. The cDNA derived from the mosquito salivary glands was used to amplify the genes, which were then inserted into the pMT/Bip/V5-HisA vector (Cat# V4130-20, Invitrogen) to construct the salivary protein plasmids. *Drosophila melanogaster* S2 cells were cotransfected with the salivary protein plasmids and a hygromycin-resistant plasmid to establish stable cell lines that expressed the recombinant mosquito salivary proteins. The recombinant proteins were C-terminally labeled with a V5 tag and a 6× His tag and purified using a cobalt column (Cat# 635515, Clontech). The purified recombinant salivary proteins were validated by immunoblot analysis of the V5 tag and checked for purity with SDS−PAGE.

## Screening of neutrophil-recruiting salivary proteins by flow cytometry assay

To screen for neutrophil-recruiting salivary proteins, purified proteins were dissolved in PBS and intradermally injected into the hind footpads of 6-week-old female 129 Sv/Ev mice. Each footpad was injected with a single salivary protein (100 ng in 20 µL PBS), while control mice were injected with 20 µL PBS. After 4 h, the mice were euthanized and perfused with ice-cold PBS to remove circulating blood. Footpads were dissected, pooled, and enzymatically digested with collagenase I and IV (1 mg/mL each, Cat# 17100017 and 17104019, Gibco) at 37 °C for 2 h. The resulting digestive juice was filtered with a 70 µm cell strainer (Cat# 352350, BD Falcon), and the cells were rinsed with PBS followed by red blood cell lysis (Cat# TNB-4300, TONBO). The resulting footpad skin cells were blocked with an anti-mouse CD16/CD32 (2.4G2) antibody (Cat# 70-0161-U100, TONBO) and stained with a panel of antibodies including FITC anti-mouse CD45 (30-F11) (Cat# 35-0451-U100, TONBO), violet-Fluor™ 450 anti-human/mouse CD11b (M1/70) (Cat# 75-0112-U100, TONBO), PE-Cyanine7 anti-mouse F4/80 Antigen (BM8.1) (Cat# 60-4801-U100, TONBO), redFluor™ 710 anti-mouse Ly-6G (1A8) (Cat# 80-1276-U100, TONBO), PerCP-Cyanine5.5 anti-mouse Ly-6C (HK1.4) (Cat# 65-5932-U100, TONBO), Brilliant Violet 605™ anti-mouse I-A/I-E (Cat# 107639, BioLegend), and APC-anti-mouse CD11c (N418) (Cat# 20-0114-U100, TONBO). Ghost Dye™ UV 450 (Cat# 13-0868-T100, TONBO) was used to distinguish living cells from dead cells. We followed a previously reported gating strategy (Liu et al, 2020) to sort and analyze skin neutrophils (CD45$^+$CD11b$^+$Ly6G$^+$), monocytes (CD45$^+$CD11b$^+$Ly6C$^+$Ly6G$^-$), macrophages (CD45$^+$CD11b$^+$F4/80$^+$), and dendritic cells (CD45$^+$I-A/I-E$^+$CD11c$^+$).

## Detection of ZIKV-infected myeloid cells by flow cytometry

The footpad skin of mice was enzymatically digested to obtain a single-cell suspension. The cells were then treated with CD16/CD32 blocking agent and stained with CD45, CD11b, F4/80, Ly6G, Ly6C, I-A/I-E, CD11c, and Ghost Dye™ UV 450 as mentioned above. Following surface staining, the cells were fixed using 4% paraformaldehyde (PFA), permeabilized with 0.3% Triton, and blocked with 5% bovine serum albumin (BSA). Subsequently, the cells were stained with murine 4G2 antibody (Cat# GTX57154, Genetex) targeting ZIKV envelope protein (ZIKV-E), followed by staining with a PE-labeled rat anti-mouse IgG secondary antibody (Cat# 12-4015-82, Invitrogen). Cells infected with ZIKV were identified by gating for PE$^+$ expression.

## Detection of mast cell activation by flow cytometry

The footpad skin of mice was enzymatically digested to obtain a single-cell suspension. The cells were then treated with CD16/CD32 blocking agent and stained with CD45, CD63 (Cat# 143905, Biolegend), CD203c (Cat# 324605, Biolegend), and Ghost Dye™ UV 450 as mentioned above. Mast cells that were activated were identified by gating for CD45$^+$CD63$^+$CD203c$^+$ expression (Ebo et al, 2021).

## Verification of the specificity of anti-AaNRP antibody by flow cytometry

HEK293T cells were transfected with either a vector or *A. aegypti* AaNRP plasmid. After forty-eight hours, the transfected cells were fixed using 4% PFA, permeabilized with 0.3% Triton, blocked with 5% BSA, and then incubated with or without murine anti-AaNRP antibody in 5% BSA. Subsequently, the cells were incubated with AF488-labeled anti-mouse IgG secondary antibody (Cat# ab150113, abcam) in 5% BSA. The cells bound by the antibody were identified by gating for AF488$^+$ expression.

## Immunohistochemical (IHC) and immunofluorescent (IF) staining

The tissue sections were initially treated with a dewaxing and hydration process, followed by a heat-mediated antigen retrieval technique. Subsequently, a 0.3% Triton (Cat# X100-5ML, Sigma Aldrich) membrane permeabilization step was conducted for 5 min, and the sections were then blocked with 1% bovine serum albumin (Cat# HY-D0842, MCE) for 1 h. Specific primary antibodies (1:250 to 1:300) were applied to the tissue sections, and they were incubated at 4 °C overnight, followed by HRP-conjugated second-ary antibody (Cat# A0208 and Cat# A0192, Beyotime) incubation at room temperature for 50 min. Positive staining visualization was achieved using an IHC kit (Cat# SNP-9001, Histostain), and the primary antibodies employed in the IHC staining included anti-Ly6G (Cat# 87048, CST), anti-F4/80 (Cat# ab6640, Abcam), and anti-CD11c (Cat# DF7585, Affinity). The mosquito salivary glands were dissected and fixed in ice-cold 4% PFA for 10 min before being smeared on a detachment-proof slide. The salivary glands were subjected to a 0.3% Triton membrane permeabilization step for 5 min and then blocked with 1% bovine serum albumin for 1 h.

The glands were incubated with mouse anti-AaNRP polyclonal IgG antibodies (1:300) at 4 °C overnight and finally incubated with Alexa Fluor 488-labeled donkey anti-mouse IgG antibody (1:500; Cat# A21202, Thermo) at room temperature for 30 min. The cell nuclei were visualized using 4',6-diamidino-2-phenylindole (DAPI) (Cat# D9542, Sigma Aldrich).

## Hematoxylin-eosin staining

To prepare mouse footpads for sectioning and paraffin embedding, they were immersed in 4% paraformaldehyde (PFA). The footpads were then cut into sagittal sections, mounted on 5 μm slides and stained with hematoxylin and eosin (H&E) using a standard protocol. Briefly, the slides were dewaxed in xylene, rehydrated in a decreasing concentration gradient of ethanol, and rinsed in deionized water. Hematoxylin and eosin were then used to stain the footpad sections. After staining, the sections were dehydrated in an increasing concentration gradient of ethanol, transparentized with xylene, and sealed with neutral resin.

## In vivo depletion of neutrophils and macrophages

For neutrophil depletion, 129 Sv/Ev mice were intraperitoneally injected with 1 mg of anti-Ly6G neutrophil depleting antibody (Cat# BE0075-1, BioxCell) or an irrelevant rat IgG2a antibody (Cat# BE0089, BioxCell) as an isotype control 4 days and 1 day before AaNRP administration. The depletion of neutrophils was confirmed by flow cytometry analysis of circulating neutrophils (CD11b$^+$CXCR2$^+$). For macrophage depletion, 129 Sv/Ev mice were intraperitoneally injected with 1 mg of anti-F4/80 macrophage depleting antibody (CI: A3-1) (Cat# BE0206, BioxCell) or an irrelevant rat IgG2b antibody (LTF-2) (Cat# BE0090, BioxCell) as an isotype control 2 days and 1 day before mosquito biting. The depletion of skin macrophages was confirmed by flow cytometry analysis (CD11b$^+$CD163$^+$) and IHC staining of the skin macrophage-specific marker CD163 (Duangkhae et al, 2018) in footpad skin.

## RNA extraction, reverse transcription, and qPCR quantification

Peripheral blood, draining lymph node, spleen, and footpad tissues were sampled, and total RNA was extracted using the AxyPrepTM Multisource Total RNA Miniprep Kit (Cat# AP-MN-MS-RNA-250, Axygen). Reverse transcription of the total RNA into total cDNA was performed using the iScript cDNA synthesis kit (Cat# 170-8890, Bio-Rad). Quantification of RNA abundances of specific genes was accomplished using an iTaq Universal SYBR Green Supermix kit (Cat# 1725121, Bio-Rad) on the Bio-Rad CFX-96 Touch Real-Time Detection System.

The qPCR primers used are shown in Appendix Table S4. Gene expression was normalized to glyceraldehyde-3-phosphate dehy-drogenase (*GAPDH*, Gene ID: 14433) in murine systems and to β-actin (*AAEL011197*) in mosquitoes using the $2^{-\Delta Ct}$ methodology.

## Single cell RNA sequencing (scRNA-Seq) and analysis

In the study, ten *A. aegypti* mosquitoes were used to bite the two hind footpads of each of eight mice in two groups (0 h and 4 h).

Footpads from each group were pooled together after collection. Control mice were not bitten, and the footpads were digested at 37 °C for 1 h using a mixed digestive solution containing type I and type II collagenases (1 mg/mL each). After dead cells and red blood cells were removed, flow cytometry was performed to sort CD45+ immunocytes for single-cell sequencing. Sequencing was carried out by CapitalBio Technology, Beijing, using a single-cell 3' Library and Gel Bead Kit V3.1 (10× Genomics, 1000121) and Chromium Single Cell G Chip Kit (10× Genomics, 1000120). The cell suspension (300–600 living cells/mL determined by Count Star) was loaded onto the Chromium single-cell controller (10× Genomics) to generate single-cell gel beads in the emulsion according to the manufacturer's protocol. Briefly, single cells were suspended in PBS containing 0.04% BSA. Approximately 6000 cells were added to each channel, and the target number of cells to be recovered was approximately 3000 cells. The captured cells were lysed, and the released RNA was barcoded through reverse transcription in individual gel-bead-in-emulsions (GEMs). Reverse transcription was performed on a S1000TM Touch Thermal Cycler (Bio-Rad) with steps of 53 °C for 45 min followed by 85 °C for 5 min and a holding step at 4 °C. The cDNA was generated and then amplified, and quality was assessed using an Agilent 4200. Single-cell RNA-seq libraries were constructed using Single Cell 3' Library and Gel Bead Kit V3.1 and sequenced using an Illumina NovaSeq6000 sequencer with a sequencing depth of at least 100,000 reads per cell with a paired-end 150 bp (PE150) reading strategy. CellRanger software obtained from the 10× Genomics website (https://support.10xgenomics.com/single-cell-gene-expression/software/downloads/latest) was used for alignment, filtering, barcoding, and UMI counting to generate feature barcodes and matrices and determine clusters. Dimensionality reduction was performed using PCA, and the first ten principal components were used to generate clusters by the K-means algorithm and graph-based algorithm. Another clustering method used was Seurat 3.0 (R package). Cells with a gene number less than 200, a gene number ranked in the top 1%, or a mitochondrial gene ratio more than 25% were regarded as abnormal and were filtered out. Dimensionality reduction was performed using PCA, and visualization was realized by TSNE and UMAP. Cell types were annotated by singleR (https://bioconductor.org/packages/devel/bioc/html/SingleR.html) using reference transcriptomic datasets of pure cell types to infer the cell origin of each single cell independently. The original scRNA-Seq data is available at NCBI-GEO with accession number GSE232756.

## Coimmunoprecipitation (Co-IP) and immunoblotting assay

HEK293T cells were transfected with pcDNA3.1(+) plasmids expressing N-terminal Flag-tagged and C-terminal Myc-tagged recombinant murine Toll-like receptor 1 (mTLR1), mTLR2, mTLR4, mTLR5, mTLR6, mTLR11, and mTLR12 extracellular domains or human plasma membrane TLR1 (hTLR1), hTLR2, hTLR4, hTLR5, hTLR6, and hTLR10 extracellular domains using Lipofectamine 2000 (Cat# 11668019, Invitrogen). After 48 h, Co-IP was performed using a Pierce Classic IP Kit (Cat# 26146, Thermo Scientific). To perform Co-IP, the cells were lysed on ice with the IP lysis buffer containing a protease inhibitor cocktail (Cat# 4693132001, Roche) and incubated with 80 ng of purified *Aa*NRP

protein at 4 °C for 4 h. This was followed by incubation with 10 μL of magnetic beads linked with anti-Myc tag antibody (Cat# P2118-2ml, Beyotime) at 4 °C for 2 h. After five washes with the IP lysis buffer, the samples were heated at 100 °C for 5 min in an SDS-PAGE loading buffer (Cat# P0015F, Beyotime) and analyzed by immunoblotting assay with a V5 tag HPR conjugated antibody (Cat# R961-25, Invitrogen) and a Myc tag HRP conjugated antibody (Cat# R951-25, Invitrogen). For immunoblotting analysis, the samples were separated by 12% SDS-PAGE and then transferred to polyvinylidene fluoride (PVDF) membranes (18 V for 55 min). The blotted PVDF membranes were blocked with 5% skim milk and incubated with primary antibodies at 4 °C overnight. The next day, the PVDF membranes were washed and incubated with HRP-conjugated secondary antibodies at room temperature for 1 h. Finally, the protein bands were visualized with a chemiluminescent reagent (WBKLS0100, Millipore). The following primary antibodies were used for the immunoblotting analysis: IκB alpha rabbit monoclonal antibody (Cat# AF1282, Beyotime), phospho-IκB alpha (Ser32) rabbit monoclonal antibody (Cat# AF1870, Beyotime), phospho-NF-κB p65 (Ser536) rabbit polyclonal antibody (Cat# AF5881, Beyotime), phospho-RelB (Ser573) rabbit polyclonal antibody (Cat# AF3380, Affinity Biosciences), and β-actin antibody (Cat# AF7018, Affinity Biosciences). The secondary antibodies were anti-rabbit IgG (H + L chain) pAb-HRP (Cat#458, MBL) and anti-mouse IgG (H + L chain) pAb-HRP (Cat#330, MBL).

## Detecting the binding of *Aa*NRP with TLRs by flow cytometry

HEK293T cells were transfected with either human and murine TLR1 and TLR4 plasmids or vector plasmids. After 48 h, the cells were collected and incubated with *Aa*NRP (1 μg/10⁶ cells) at 4 °C for 2 h. Next, the cells were rinsed and incubated with anti-*Aa*NRP polyclonal antibody (purified IgG from the sera of *Aa*NRP-immunized mice) for 1 h, followed by a PE-labeled secondary antibody (Cat# 12-4015-82, Invitrogen) for 30 min at 4 °C. The binding of *Aa*NRP to TLRs was evaluated by detecting the proportion of PE+ cells using flow cytometry. To calculate the neat PE+ cell ratio of each sample, the mean PE+ ratio of vector controls was deducted from the values obtained.

## NF-κB luciferase reporter assay and NF-κB nuclear translocation assay

To perform the NF-κB luciferase reporter assay, an NF-κB luciferase reporter plasmid and a *Renilla* luciferase reporter plasmid (in a 19:1 ratio) were cotransfected into RAW264.7 cells using Lipofectamine 3000 reagent (Cat# L3000015, Invitrogen). Twelve hours post transfection, RAW264.7 cells were treated with different concentrations of *Aa*NRP (0, 0.25, 0.50, or 1.00 μg/mL) and 1.00 μg/mL LPS (Cat# HY-D1056, MCE) as a positive control for 4 h. Dual luciferase activity was detected on a Thermo Varioskan Flash platform with a dual luciferase reporter detection kit (Cat# E1910, Promega). The relative luciferase activity was calculated by normalizing the activity of NF-κB luciferase against the activity of *Renilla* luciferase. The NF-κB p65 nuclear translocation assay was conducted by following previously reported methods (Bartfeld et al, 2009; Chen et al, 2012). In brief,

RAW264.7 cells were transfected with the NF-κB p65-EGFP plasmid. Twelve hours post transfection, RAW264.7 cells were treated with 1.0 μg/mL *Aa*NRP and 1.0 μg/mL LPS as a positive control. Four hours posttreatment, the RAW264.7 cells were fixed with 4% PFA. The translocation of NF-κB p65-EGFP was observed under a Zeiss LSM880 confocal microscope.

## Antagonist-mediated inhibition of MyD88 signaling

To inhibit MyD88 signaling in vitro, RAW264.7 cells and murine peritoneal macrophages (Wang et al, 2020b) were preincubated with 4 μg/mL T6167923, a MyD88-specific inhibitor (Olson et al, 2015), for 30 min. After T6167923 pretreatment, the cells were further treated with 4 μg/mL T6167923 together with other specific treatments for the indicated time. For in vivo MyD88 signaling inhibition, mice were intraperitoneally injected with 0.5 mg of T6167923 1 h before other treatments.

## Inhibition of TLR1 and TLR4 by Anti-TLR-MIX

To achieve dual inhibition of TLR1 and TLR4, RAW264.7 cells were preincubated with Anti-TLR-MIX (CU-CPT22 and TLR4-IN-C34, 10 μM each) dissolved in DMEM supplemented with 2% FBS for 30 min. Afterward, the cells were further treated with 10 μM Anti-TLR-MIX together with other specific treatments for the indicated times.

## Gene silencing and viral infection in mosquitoes

To silence the expression of the mosquito salivary protein *Aa*NRP, female mosquitoes were briefly anesthetized on ice before being transferred to a cold platform (Cat# 1431, BioQuip) for intrathoracic injection of 1 μg/300 nL double-stranded RNA (dsRNA) targeting the *Aa*NRP gene. The injected mosquitoes were then allowed to recover under standard rearing conditions before their salivary glands were dissected for gene silencing efficiency evaluation via qPCR and immunofluorescence staining of *Aa*NRP. To study viral infection in mosquitoes, the mosquitoes were anesthetized on a cold platform and inoculated with 10 PFU of ZIKV via microinjection into the mosquito thoraxes. The virus-inoculated mosquitoes were maintained in a rearing climate chamber at 28 °C for 8 days before bite-mediated viral transmission experiments were conducted.

## Mouse infection and tissue collection

For intradermal viral infection, six-week-old female A129 mice were intradermally injected with either 250 PFU ZIKV dissolved in PBS or a mixture of 250 PFU ZIKV and 100 ng *Aa*NRP dissolved in PBS, and three-week-old female A129 mice were intradermally injected with either 3000 PFU DENV2 dissolved in PBS or a mixture of 3000 PFU DENV2 and 100 ng *Aa*NRP dissolved in PBS; all solutions were intradermally injected in the left hind footpad using a very fine tipped syringe. For mosquito-borne viral infection, each 6-week-old female A129 mouse was bitten by three ZIKV-infected mosquitoes under anesthesia induced by Avertin (Cat# HY-B1372, MCE) for 30 min. After infection, the mice were monitored daily for body weight and survival status, and tail blood was collected from Day 1 through Day 7 to detect viremia. At 48

hpi, the mice were euthanized and perfused, and their footpads, draining lymph nodes, and spleen tissues were collected for further analysis.

## Focus forming assay

The focus forming assay (FFA) previously reported by Brien et al (Brien et al, 2019) was employed to quantify the viral load of infectious agents in blood plasma, animal tissues, and cell supernatants. Murine tissues were homogenized in DMEM supplemented with 2% FBS at a ratio of 300 μL per footpad skin, lymph node, or 1 mg spleen tissue. Subsequently, the tissue homogenate was retrieved through centrifugation at $10,000 \times g$ for 10 min at 4 °C. To facilitate the FFA, the tissue homogenates, blood plasma, and cell supernatants were appropriately diluted using DMEM supplemented with 2% FBS.

## Enzyme-linked immunosorbent assay (ELISA)

The footpads of the mice were collected, weighed, and then ground using 1× Cell Lysis Buffer (Cat# 9803, CST). The resulting supernatants were obtained by centrifugation at $12,000 \times g$ for 10 min. To quantify the concentrations of mouse CXCL1, CXCL2, and CXCL3 in the supernatants, three ELISA kits were utilized (ELM-KC-1 for CXCL1 by Raybiotech, ELM-MIP2-1 for CXCL2 by Raybiotech, and KIT50070 for CXCL3 by SinoBio), following the instructions provided by the manufacturers.

## Resveratrol administration

To investigate the effect of resveratrol on viral infection and MyD88-NF-κB signaling, different concentrations of resveratrol (2.5, 5, and 20 mg/mL) were prepared in sterile water and mixed well. Female 6-week-old A129 mice were orally administered 200 μL of 2.5 mg/mL (0.5 mg), 5 mg/mL (1.0 mg), or 20 mg/mL (4.0 mg) resveratrol once daily for 14 days before and 7 days after viral infection. The resveratrol used in this study was obtained from Macklin (Cat# R817263).

## Isolation of circulating neutrophils from peripheral blood

Human and murine peripheral blood samples were collected to isolate circulating neutrophils. Neutrophils were isolated using a density gradient centrifugation-based neutrophil isolation kit (Cat# P9040 and P9201, Solarbio). In brief, the blood samples were gently pipetted on the neutrophil separating solution and centrifuged at $800 \times g$ for 20 min to separate the cells into different layers. The neutrophils were then carefully collected, and contaminated erythrocytes were removed using a red blood cell lysis buffer to acquire pure neutrophils. Flow cytometry was used to confirm the purity of neutrophils (CD45[+]CD11b[+]Ly6G[+]). The isolated neutrophils were immediately subjected to the transwell assay.

## Neutrophil transmigration assay

To perform the transwell assay, we used a 5.0 μm pore transwell insert (Cat# 3421, Corning). A total of $2 \times 10^5$ peripheral neutrophils were suspended in 250 μL of RPMI 1640 medium

containing 2% FBS and added to the upper chamber. The lower chamber was filled with 600 μL of RPMI 1640 medium supplemented with 2% FBS and containing 1.0, 2.0, or 4.0 μg/mL *Aa*NRP or 0.5 μg/mL of murine or human CXCL1 (Cat# 50150-MNCE and 10877-HNCE, SinoBio) as the positive controls. After 3 h of incubation at 37 °C, the number of neutrophils that had transmigrated to the lower chamber or adhered to the basolateral side of the insert was collected and counted using a cell counter (Celldrop FL/BF, Denovix).

## Statistical analysis

The investigators were not blinded to the allocation during the experiments or to the outcome assessment. No statistical methods were used to predetermine the sample size. Quantitative data that conformed to normal distribution and had equal variance are presented as the mean ± SEM. Data that failed to conform to a normal distribution or were not equal in variance are presented as medians. GraphPad Prism 10.0.0 software was used for statistical analyses. The unpaired *t* test (parametric) or Mann–Whitney test (nonparametric) was used to compare quantitative data between two groups. One-way analysis of variance (ANOVA), two-way ANOVA (test for murine viremia) and post-ANOVA multiple *t* tests were used to compare data with three or more groups. The log-rank test was used to compare survival curves. Statistical significance was set at $p < 0.05$.

## Data availability

The scRNA-seq data are available on the Gene Expression Omnibus database under the accession GSE232756. The RNA-seq data are available on the Sequence Read Archive database under the accession PRJNA1045313. Other data that supports the findings of this manuscript can be found in sourced data or obtained from the corresponding author upon request.

## Peer review information

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

## Acknowledgements

We thank the core facilities of the Center for Life Sciences and Center of Biomedical Analysis for technical assistance (Tsinghua University). This study was supported by grants from the National Key Research and Development Plan of China (2021YFC2300200, 2021YFC2302405, 2022YFC2303200, and 2022YFC2303400), the National Natural Science Foundation of China (32188101, 31825001, 81961160737, and 82102389), the Research Fund-Vanke School of Public Health, Tsinghua University (2023JC002), Shenzhen Science and Technology Project (JSGG20191129144225464), Shenzhen San-Ming Project for Prevention and Research on Vector-borne Diseases (SZSM202211023), Innovation Team Project of Yunnan Science and Technology Department (202105AE160020), Science and Technology Project of Southwest United Graduate School of Yunnan (202302AO370010), the Young Elite Scientists Sponsorship Program (2021QNRC001). The New Cornerstone Science Foundation through the New Cornerstone Investigator Program, and the Xplorer Prize from Tencent Foundation.

## Author contributions

**Zhaoyang Wang**: Conceptualization; Data curation; Formal analysis; Investigation; Visualization; Methodology; Writing—review and editing. **Kaixiao Nie**: Resources; Visualization. **Yan Liang**: Validation; Investigation. **Jichen Niu**: Resources. **Xi Yu**: Methodology. **Oujia Zhang**: Validation; Investigation. **Long**

**Liu**: Resources. **Xiaolu Shi**: Resources. **Yibaina Wang**: Resources. **Xuechun Feng**: Resources. **Yibin Zhu**: Visualization; Methodology. **Penghua Wang**: Validation; Methodology; Writing—review and editing. **Gong Cheng**: Conceptualization; Data curation; Formal analysis; Supervision; Funding acquisition; Visualization; Methodology; Writing—original draft; Project administration.

## Disclosure and competing interests statement

The authors declare no competing interests.

# Expanded View Figures

**Figure EV1.** *AaNRP*-mediated neutrophil influx and the specific expression of *AaNRP* in *A. aegypti* mosquitoes, related to Fig. 1. ▶

(**A**) Dot-plot graphs showing the flow cytometric analysis of neutrophils (CD45$^+$CD11b$^+$Ly6G$^+$) in mouse footpad skin at 4 h post inoculation (hpi, related to Fig. 1C). 129 Sv/Ev mice were intradermally inoculated with 100 ng *AaNRP* or PBS in the hind footpads as negative controls. (**B**) Dot-plot graphs showing the flow cytometric analysis of neutrophils in mouse footpad skin at 4 hpi, 12 hpi, 24 hpi and 48 hpi (related to Fig. 1D). 129 Sv/Ev mice were intradermally inoculated with 100 ng *AaNRP*, equivalent amount of mosquito saliva as a positive control, or PBS, heat-inactivated *AaNRP* (hi*Aa*NRP) and a noninflammatory control (NIC) salivary protein encoded by *AAEL009524* as negative controls. (**C**) IHC staining of the specific neutrophil marker Ly6G in footpad sections at 4 hpi and 12 hpi. 129 Sv/Ev mice were intradermally inoculated with 100 ng *AaNRP* or PBS in the hind footpads as negative controls. Red arrows indicate neutrophils, scale bar 20 μm. (**D**) Quantification of the number of Ly6G$^+$ cells per mm$^2$ of footpad skin sections at 4 hpi and 12 hpi. (**E**) Hematoxylin-eosin (HE) staining of neutrophils in footpad skin at 4 hpi and 12 hpi. Red arrows indicate neutrophils, and regions circled with dashed red lines denote intravascular lumens, cale bar 20 μm. (**F–H**) RNA expression level of *AaNRP* in different mosquito tissues (**F**), in salivary glands with female specificity (**G**) and in salivary glands at 24 h post blood meal (**H**). The RNA level of *AaNRP* was detected by qPCR and expressed as the ratio of β-actin with 2$^{-\Delta Ct}$ normalization. (**D, F–H**) Data are expressed as the mean ± SEM, and each dot represents an individual mouse (**D**) or mosquito (**F–H**). The two-way ANOVA and multiple *t* tests (**D**), one-way ANOVA (**F**), and unpaired *t* test (**G, H**) were used for the statistical analyses. All experiments were reproduced at least twice. \*\**p* < 0.01, \*\*\**p* < 0.001, \*\*\*\**p* < 0.0001.

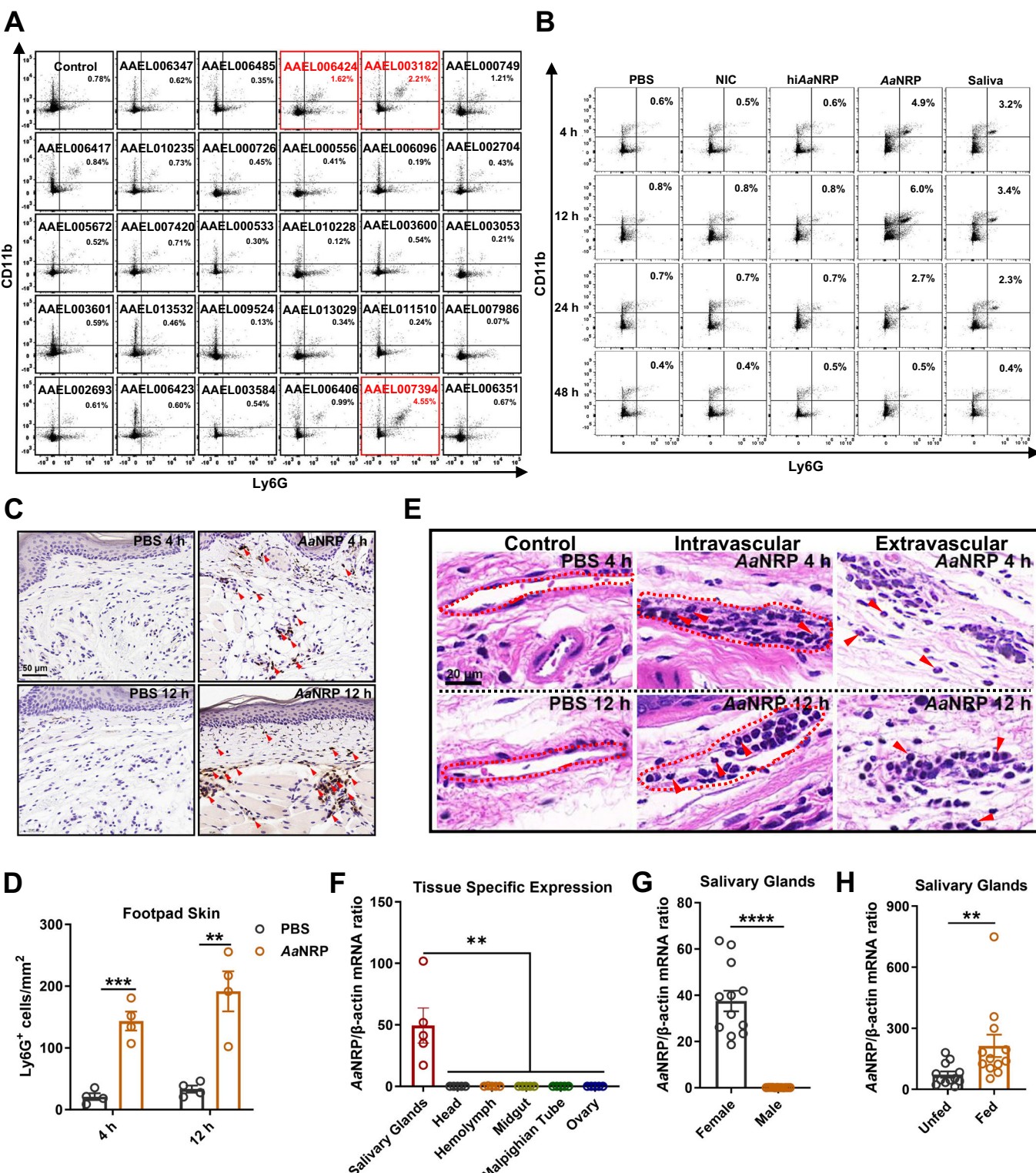

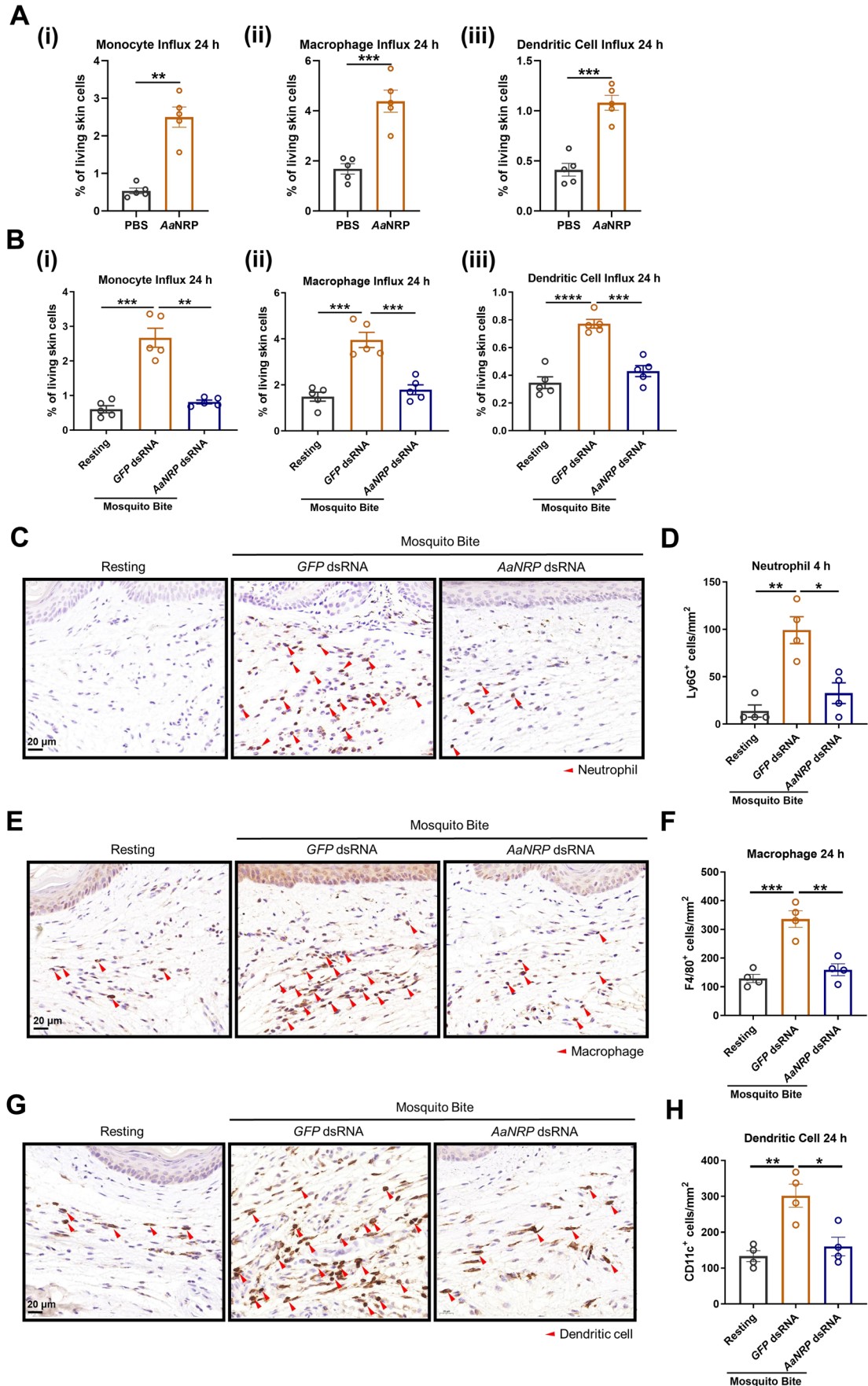

◀ **Figure EV2. *Aa*NRP induces myeloid cell influx, related to Fig. 1.**

(**A**) *Aa*NRP-induced influx of monocytes (i), macrophages (ii), and dendritic cells (DCs, iii) toward murine footpad skin at 24 hpi. 129 Sv/Ev mice were intradermally inoculated with 100 ng *Aa*NRP or PBS in the hind footpads as negative controls. At 24 hpi, murine footpads were sampled to measure the percentages of myeloid cells in all living skin cells by flow cytometry. (**B**) Influence of *AaNRP* silencing on mosquito bite-induced influx of monocytes (i), macrophages (ii), and DCs (iii) at 24 h post bite (hpb). *A. aegypti* mosquitoes were intrathoracically injected with 1 μg/300 nL *AaNRP* dsRNA or *GFP* dsRNA (negative control). Three days later, mosquitoes were allowed to bite the hind footpads of 129 Sv/Ev mice (each hind footpad was bitten by five mosquitoes) and the unbitten mice served as the resting controls. At 24 hpb, murine footpads were sampled to measure the percentages of myeloid cells in all living skin cells by flow cytometry. (**C–H**) Influence of *AaNRP* silencing on mosquito bite-induced myeloid cell influx assessed by immunohistochemical (IHC) staining. *A. aegypti* mosquitoes were intrathoracically injected with 1 μg/300 nL *AaNRP* dsRNA or *GFP* dsRNA (negative control). Three days later, mosquitoes were allowed to bite the hind footpads of 129 Sv/Ev mice (each hind footpad was bitten by five mosquitoes) and the unbitten mice served as the resting control. At 4 hpb and 24 hpb, murine footpads were dissected for IHC staining of neutrophils at 4 hpb (**C**) and of macrophages (**E**) and DCs (**G**) at 24 hpb. (**C**) IHC staining of the neutrophil-specific marker Ly6G indicated by red arrows, scale bar 20 μm. (**D**) Quantification of the counts of neutrophils per mm$^2$ of footpad section. Four random scopes were sampled from each mouse, and the mean of the four scopes was used to represent the mouse. (**E**) IHC staining of the macrophage-specific marker F4/80 indicated by red arrows, scale bar 20 μm. (**F**) Quantification of the counts of macrophages per mm$^2$ of footpad section. (**G**) IHC staining of the DC-specific marker CD11c indicated by red arrows, scale bar 20 μm. (**H**) Quantification of the counts of DCs per mm$^2$ of footpad section. (**A**, **B**, **D**, **F**, **H**) Data are expressed as the mean ± SEM, and each dot represents an individual mouse. The unpaired *t* test (**A**) and the one-way ANOVA and multiple *t* tests (**B**, **D**, **F**, **H**) were used for statistical analyses. All experiments were reproduced at least twice. *$p < 0.05$, **$p < 0.01$, ***$p < 0.001$, ****$p < 0.0001$.

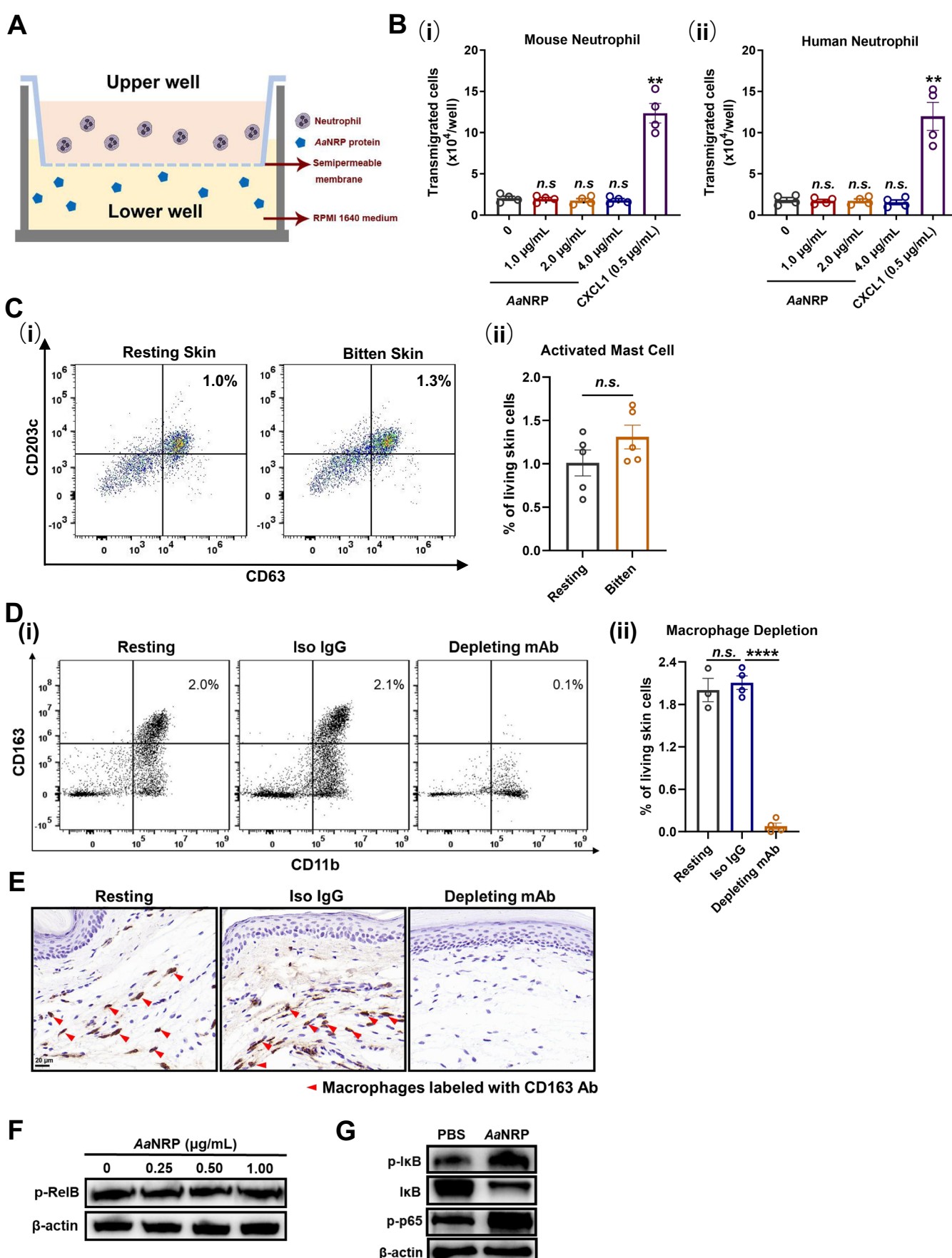

◄  **Figure EV3.   Influence of *Aa*NRP on the chemotaxis, activation, and inflammatory signaling of skin innate immunocytes, related to Figs. 2 and 3.**

(A–C) Transwell assays exploring the direct chemotaxis of neutrophils by *Aa*NRP. (A) Study design for the neutrophil transmigration assay. A transwell with a 5.0 μm pore was used. Neutrophils ($2 \times 10^5$) isolated from human and murine peripheral blood and suspended in 250 μL RPMI 1640 medium supplemented with 2% FBS were added to the upper chamber. RPMI 1640 (600 μL) containing 0, 1.0, 2.0, or 4.0 μg/mL *Aa*NRP or 0.5 μg/mL mouse and human CXCL1 (as positive controls) supplemented with 2% FBS was added to the lower chambers. Three hours later, the neutrophils that transmigrated into the lower chamber or adhered to the basolateral side of the upper chamber were collected and counted using a cell counter (Celldrop, FL/BF, Denovix). (B) Counts of mouse (i) and human (ii) neutrophils that transmigrated to the lower chambers under different concentrations of *Aa*NRP or CXCL1. (C) Influence of *Aa*NRP on skin mast cell activation as assessed by flow cytometry. (i) Dot plots showing the activated mast cells that are CD63$^+$CD203c$^+$. (ii) Percentages of the activated mast cells in resting and mosquito-bitten skin. Mice were bitten by mosquitoes in the hind footpads (one footpad was bitten by five mosquitoes) or left unbitten as the resting controls. Four hours later, the footpad skin was collected and processed for flow cytometric analysis of mast cell activation. (D–E) Verification of resident macrophage depletion in murine footpad skin by flow cytometry (D) and IHC (E). (D, i) Dot plots showing the skin resident macrophages (CD45$^+$CD11b$^+$CD163$^+$). (D, ii) Percentages of skin resident macrophages within all the living skin cells. (E) IHC staining of resident macrophages in murine footpad skin with a specific biomarker CD163 indicated by red arrows, scale bar 20 μm. 129 Sv/Ev mice were intraperitoneally injected with 1 mg anti-F4/80 monoclonal antibody (depleting mAb) or 1 mg rat isotypic IgG2b antibody at 2 and 1 days before sampling. Footpads were collected from untreated (resting), isotype IgG-injected, and anti-F4/80 mAb-injected mice and were then used for flow cytometry and IHC analysis of skin resident macrophages. (F) Influence of *Aa*NRP on the expression of phosphorylated RelB (p-RelB). RAW264.7 cells were treated with 0, 0.25, 0.50, or 1.00 μg/mL *Aa*NRP in DMEM supplemented with 2% FBS for 4 h. Subsequently, the cells were collected and subjected to an immunoblotting assay to detect the protein abundance of p-RelB. (G) Influence of *Aa*NRP on the canonical NF-κB signaling in murine footpad. 129 Sv/Ev mice were intradermally injected with 100 ng *Aa*NRP or 20 μL PBS in the hind footpads with a very fine-tipped syringe. Four hours later, the footpads were collected for an immunoblotting assay to detect the protein abundance of some crucial signaling molecules in the canonical MyD88-NF-κB signaling axis. (B–D) Data are expressed as the mean ± SEM and each dot represents an individual mouse. The one-way ANOVA and multiple *t* tests (B, D) and unpaired *t* test (C) were used for statistical analyses. All experiments were reproduced at least twice. **$p < 0.01$, ****$p < 0.0001$, n.s. not significant.

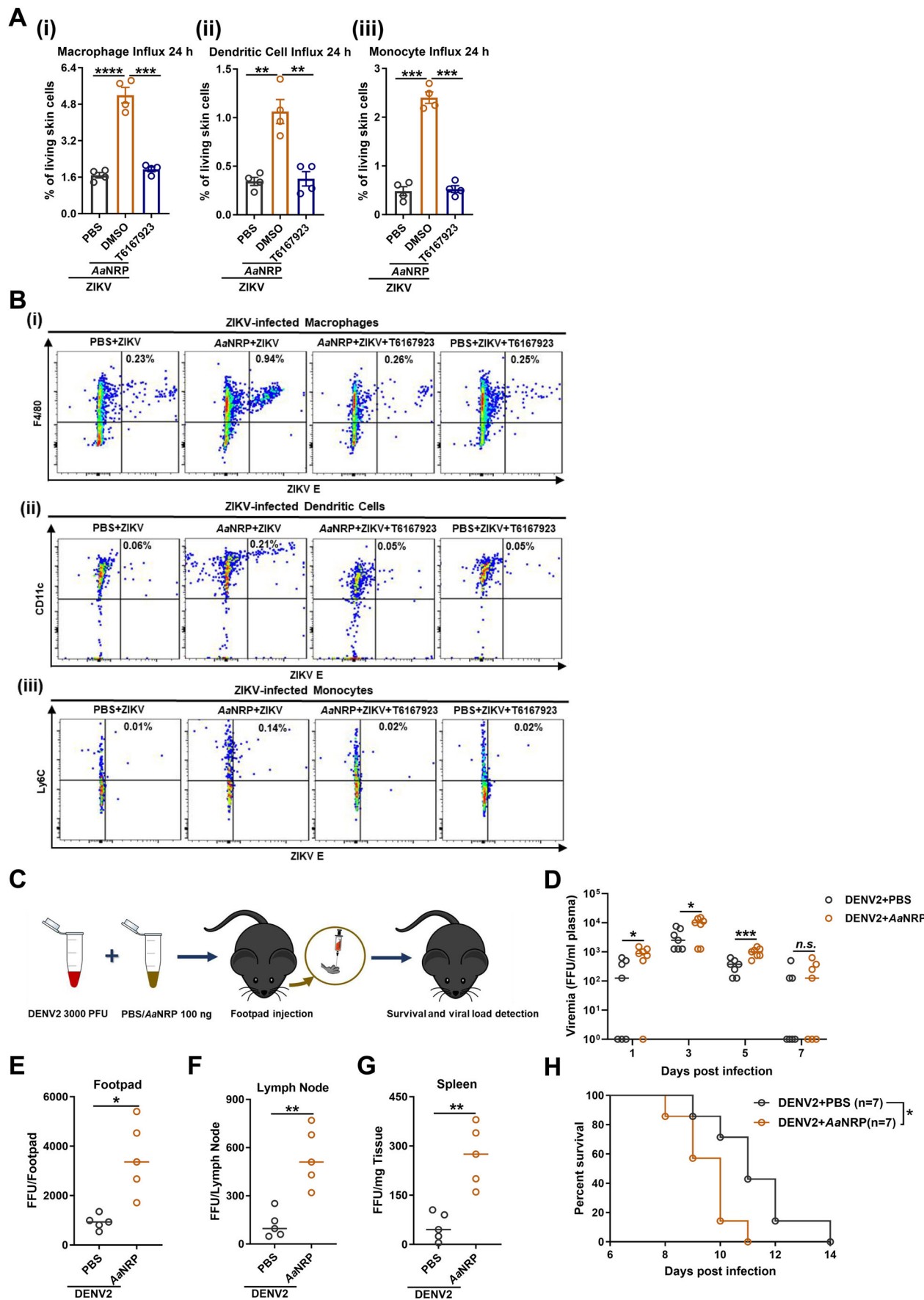

◄   **Figure EV4.   *Aa*NRP enhances flaviviral infection in mice, related to Figs. 5 and 6.**

(A) Influx of macrophages (i), DCs (ii) and monocytes (iii) toward murine footpad skin at 24 hpi. Six-week-old female type I interferon receptor-deficient (*ifnar⁻/⁻*) 129 (A129) mice were intraperitoneally injected with 0.5 mg T6167923 or DMSO (solvent control). One hour later, the mice were intradermally inoculated with 100 ng *Aa*NRP plus 250 PFU ZIKV (Group: *Aa*NRP + ZIKV and Group: *Aa*NRP + ZIKV + T6167923) or PBS plus 250 PFU ZIKV (Group: PBS + ZIKV) in their footpads. At 24 hpi, murine footpad skin was collected for flow cytometric analysis of the percentages of different myeloid cells in all the living skin cells. (B) Representative flow cytometric pseudocolor plots of ZIKV-infected macrophages (CD45⁺CD11b⁺F4/80⁺ZIKV-E⁺, i), DCs (CD45⁺I-A/I-E⁺CD11c⁺ZIKV-E⁺, ii) and monocytes (CD45⁺CD11b⁺Ly6C⁺Ly6G⁻ZIKV-E⁺, iii) in murine footpad skin at 24 hpi. This data is the representative gating of ZIKV-infected myeloid cells in Fig. 5C. (C–H) *Aa*NRP-mediated enhancement of dengue virus 2 (DENV2) infection in A129 mice. (C) Schematic of the study design. Three-week-old female A129 mice were intradermally injected with 3000 PFU of DENV2 plus PBS (DENV2 + PBS) or 3000 PFU of DENV2 plus 100 ng *Aa*NRP (DENV2 + *Aa*NRP) in the hind footpad with a fine-tipped syringe. (D) DENV2 load in murine peripheral blood plasma. Murine tail blood was collected for viral load detection by FFU assay. The DENV2 loads were shown as FFU/ml blood plasma. (E–G) DENV2 load in the murine footpad (E), lymph node (F), and spleen (G) at 48 hpi. DENV2 loads were measured by FFU assay and shown as FFU/footpad, FFU/lymph node, or FFU/mg spleen tissue. (H) Survival curves of the mice inoculated with DENV2 + PBS ($n = 7$) or DENV2 + *Aa*NRP ($n = 7$). (A) Data are expressed as the mean ± SEM and each dot represents an individual mouse. The one-way ANOVA and multiple *t* tests were used for statistical analyses. (D–G) Data are expressed as the median and each dot represents an individual mouse. The two-way ANOVA (D) and Mann–Whitney tests (E–G) were used for statistical analysis. (H) The log-rank test was used to compare the survival curves. All experiments were reproduced at least twice. *$p < 0.05$, **$p < 0.01$, ***$p < 0.001$, ****$p < 0.0001$, n.s. not significant.

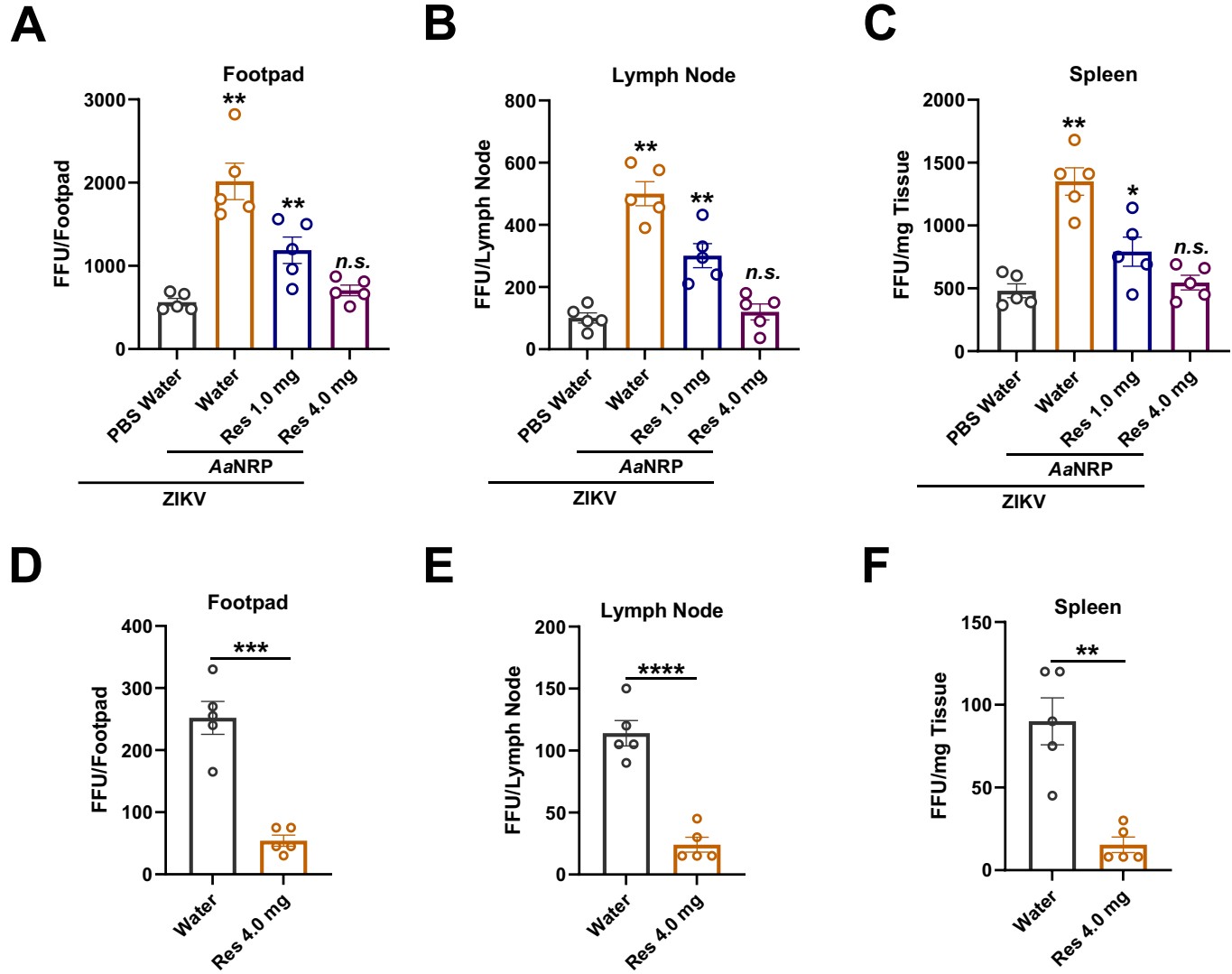

**Figure EV5. Oral gavage of resveratrol relieves *Aa*NRP-promoted ZIKV infection in A129 mice, related to Fig. 7.**

(A–C) ZIKV loads in the footpads (A), lymph nodes (B), and spleens (C) of mice intradermally inoculated with ZIKV at 48 hpi. Six-week-old A129 mice were orally administered water or 1.0 mg or 4.0 mg resveratrol (Res) once daily for 14 days. Afterward, the resveratrol-administered mice were intradermally injected with 100 ng *Aa*NRP + 250 PFU ZIKV immediately after the last dose of resveratrol, while half of the water-administered mice were intradermally injected with PBS + 250 PFU ZIKV as the negative control, and the remaining half were intradermally injected with 100 ng *Aa*NRP + 250 PFU ZIKV as the positive control. At 48 hpi, murine footpads, draining lymph nodes, and spleens were sampled for ZIKV load detection by FFU assay. Viral loads are shown as FFU/footpad, FFU/lymph node, or FFU/mg spleen tissue. (D–F) ZIKV loads in the footpads (D), lymph nodes (E), and spleen (F) of mice bitten by ZIKV-infected mosquitoes at 48 hpi. Six-week-old A129 mice were orally administered 4.0 mg/200 μL resveratrol (Res) or 200 μL sterile water once daily for 14 days. Afterward, each A129 mouse was bitten by three ZIKV-infected mosquitoes immediately after the last dose of resveratrol or water. At 48 hpi, murine footpads, lymph nodes, and spleens were sampled for ZIKV load detection by FFU assay. (A–F) Data are expressed as the mean ± SEM and each dot represents an individual mouse. The one-way ANOVA and multiple *t* tests (A–C) and unpaired *t* test (D–F) were used for statistical analyses. All experiments were reproduced at least twice. *$p < 0.05$, **$p < 0.01$, ***$p < 0.001$, ****$p < 0.0001$, n.s. not significant.

