## [Peer Review File · The EMBO Journal]

A mosquito salivary protein-driven influx of myeloid cells facilitates flavivirus transmission.

Zhaoyang Wang, Kaixiao Nie, Yan Liang, Jichen Niu, Xi Yu, Oujia Zhang, Long Liu, Xiaolu Shi, Yibaina Wang, Xuechun Feng, Yibin Zhu, Penghua Wang, and Gong Cheng

Corresponding author: Gong Cheng (gongcheng@mail.tsinghua.edu.cn)

Review Timeline:

Submission Date:	11th Jul 23
Editorial Decision:	28th Aug 23
Revision Received:	14th Dec 23
Editorial Decision:	26th Jan 24
Revision Received:	29th Jan 24
Accepted:	2nd Feb 24

Editor: Ieva Gailite

Transaction Report:

Dear Dr. Cheng,

Thank you for submitting your manuscript for consideration by the EMBO Journal. I apologise for the protracted assessment process due to delays in referee report submission. We have now received comments from three reviewers, which are included below for your information.

As you will see from the reports, all reviewers find the study of interest. However, they also raise important concerns that would have to be addressed, including exclusion of LPS contamination, further support for the role of neutrophils in myeloid cell recruitment, and an assessment of potential effects on mosquito feeding and saliva deposition, as well as a better quantification of infected cells and the amount of the infectious virus.

Based on the interest expressed in the reports, I invite you to address these issues in a revised version of the manuscript. I think it would be helpful to discuss the revision in more detail via email or phone/videoconferencing - please let me know which option you prefer. I should also add that it is The EMBO Journal policy to allow only a single major round of revision and that it is therefore important to resolve the main concerns at this stage.

We generally allow three months as standard revision time, which can be extended to six months in the case of major revisions. As a matter of policy, competing manuscripts published during this period will not negatively impact on our assessment of the conceptual advance presented by your study. However, please contact me as soon as possible upon publication of any related work to discuss the appropriate course of action. Should you foresee a problem in meeting this deadline, please let us know in advance to discuss an extension.

When preparing your letter of response to the referees' comments, please bear in mind that this will form part of the Review Process File and will therefore be available online to the community. For more details on our Transparent Editorial Process, please visit our website: <https://www.embopress.org/page/journal/14602075/authorguide#transparentprocess>. Please also see the attached instructions for further guidelines on preparation of the revised manuscript.

Please feel free to contact me if you have any further questions regarding the revision. Thank you for the opportunity to consider your work for publication. I look forward to discussing your revision.

With best regards,

leva

leva Gailite, PhD
Senior Scientific Editor
The EMBO Journal
Meyerhofstrasse 1
D-69117 Heidelberg
Tel: +4962218891309
i.gailite@embojournal.org

- a point-by-point response to the referees' comments, with a detailed description of the changes made (as a word file).

- a word file of the manuscript text.
 - individual production quality figure files (one file per figure)
 - a complete author checklist, which you can download from our author guidelines (<https://www.embopress.org/page/journal/14602075/authorguide>).
 - Expanded View files (replacing Supplementary Information)
- Please see out instructions to authors
<https://www.embopress.org/page/journal/14602075/authorguide#expandedview>

We realize that it is difficult to revise to a specific deadline. In the interest of protecting the conceptual advance provided by the work, we recommend a revision within 3 months (26th Nov 2023). Please discuss the revision progress ahead of this time with the editor if you require more time to complete the revisions.

Referee #1:

The ability of mosquito saliva to modulate skin biology and thereby enhance infection of the vertebrate host with mosquito-borne virus is an active area of research. This manuscript reports the identification of a novel factor (AaNRP) in mosquito saliva that modulates host susceptibility to mosquito-borne virus infection. They define the ability of AaNRP to modulate infection with two medically important viruses, ZIKV and DENV and further identify a mechanism, involving the activation of innate immune signalling pathway (TLR4-MyD88) that leads to the expression of neutrophil attracting chemokines by skin macrophages. Recruited leukocytes become infected and this is inferred to account for the ability of mosquito saliva to enhance infection with virus.

Overall, this study is nicely done and provides some important new insights. However, there are some major and minor concerns that should be addressed prior to publication. The role that mosquito saliva has in modulating host susceptibility to virus infection is poorly defined and yet represents a key stage of infection that is common to all viruses transmitted by mosquito to the vertebrate host. As such the findings, if further validated with corrections listed below, are important, novel and of relevance to several key research areas (e.g. virology, immunology, vector biology).

Major concerns

LPS contamination.

It is common for recombinant proteins made in-house to have LPS contamination, even for those made using insect cell lines. The accidental presence of LPS could activate these responses observed, both in vivo (e.g. neutrophil recruitment) and in vitro assays (e.g. MyD88 pathway activation). TLR2/4 ligands, when co-inoculated with arbovirus are known to enhance infection with arbovirus. It is essential that AaNRP preps are tested for their ability to activate LPS specific responses and / or LPS concentration defined. As the proposed mechanism of action is via MyD88, a kit that that doesn't use this system to detect endotoxin should be employed e.g. the Pierce LAL Chromogenic Endotoxin Quantitation Kit.

Even if very low levels of endotoxin are detected, it would then be necessary to show that endotoxin, when spiked into a control recombinant product (e.g. a non-inflammatory mosquito salivary protein other than AaNRP, or scrambled protein) at a similar concentration, does not replicate results observed with AaNRP.

Minor corrections.

Introduction

Lines 71-72. There are a number of factors in mosquito saliva that have been identified as able of modulating host susceptibility to arbovirus infection, as listed later in discussion. I would suggest rephrasing this here to make this clear.

Lines 86-88. Onset of oedema is very rapid and the role that leukocyte influx has on this process is still to be defined. It is like that both neutrophil influx and oedema occur at the same time based on the published studies cited.

Results

Table S1

Line 123 " We next assessed the role of these salivary proteins in triggering neutrophil influx into the host skin. Thus, 100 ng of purified salivary proteins were individually inoculated into a mouse footpad in a subcutaneous manner". It is not clear why 100ng was chosen and whether this reflects the mass inoculated by a probing mosquito. The amount salivated by mosquitoes, e.g. following forced salivation into oil or media should be define.

Figure 1C - a Students T test has been used incorrectly and doesn't account for multiple testing. One would expect a p value of less than 0.05 for every 20 proteins studied, based on chance differences. Since there are more than 20 groups, this is quite likely. A one-way ANOVA or Kruskal Wallis, depending on whether data is normally distributed, should be applied instead. This also applies to most of the statistics used elsewhere, in which a Students T test has been used. All should be corrected.

Figure 1D-L. It is not clear how leukocyte influx, elicited by injection with AaNRP, compares to mosquito biting and/or injection with equivalent amount of mosquito saliva injection. The authors should offer this comparison for key cell types identified.

Figure 1E,F and L. The authors have only assessed monocyte, macrophage and DC influx at 24 hours post injection. This quite a late timepoint, considering enhancement of arbovirus infection by saliva is evident before 6 hours (including in the studies cited in this manuscript e.g. Styer et al). In addition, previously reported infection of myeloid cells occurs within this time frame, long before the 24-hour time point. Virus life cycle is 8 to 10 hours, which means by 24 hours there would have been several rounds of infection and e.g. dissemination of virus to blood. The authors should expand these analyses to include earlier time points. This is key, as entry of virus permissive cells at 24 hours and later may not have significant bearing on defining the outcome to infection, as virus tends to disseminate from skin to blood / systemically within this time frame. This is also the case for later figures e.g. Figure 5, which only shows 24-hour post infection.

Volume of recombinant protein injected was 20ul, this is many orders of magnitude higher than what a mosquito would deposit. This limitation should be discussed.

Figure 1 and 5. Absence of virus vs. uninfected skin comparison in leukocyte entry kinetics experiments. The purpose of this research is to inform on leukocyte migration during transmission of virus from mosquito to vertebrate skin. Virus is an important activator of chemokines and leukocyte influx, and yet is excluded from figure 1, and the comparator to resting skin missing in figure 5. This should be included.

Figure 2.

Validation of macrophage depletion.

Skin macrophages were depleted through use of an antibody to F4/80. Such depletions are difficult to get working at high efficiency for non-circulating tissue resident cells such as these. The complete absence of macrophages in the F4/80 Ab treated group seems surprising, however the antibody used to validate macrophage depletion was the same as that used for depletion. This is not suitable as it could simply indicate epitope masking by antibody rather than true cell depletion. Observed alterations in neutrophil recruitment could instead reflect consequences of antibody infusion associated with this specific Ab clone. Findings should reassessed or macrophage depletion validated through alternative means.

Furthermore, is F4/80 a good marker for skin macrophages? As they will also deplete other cells in the skin e.g. Langerhans cells. Some macrophage population, e.g. those CD11b+ macrophages derived from bone marrow progenitors are typically low for F4/80. This last point should be discussed e.g. <https://www.ncbi.nlm.nih.gov/pmc/articles/PMC4045180/>

Figure 3 and 4 are generally well done, although would be more meaningful if LPS contamination of AaNRP preparation was ruled out. Confirmation of TLR4-MyD88 signalling pathway activation in primary cultures of Macrophages is required, as immortalised RAW cells often generate data that is hard to replicate in terminally differentiated (and thereby relevant) cells. The specificity of the new anti-AaNRP antibody needs to be defined for data in Fig 4B/C to meaningful. As for most of the paper, the wrong statistical test has been used; ANOVA is the correct test if there are more than two groups.

Figure 5

Representative gating of ZIKV positive cells should be shown.

Infected cells do not necessarily lead to release of new infectious virus and could also represented phagocytised viral debris. The ability of skin isolated infected leukocytes to release new infectious virus should be defined.

Figure 6. The authors need to demonstrate that siRNA KO of AaNRP does not have any unintentional effects on mosquito feeding, and thereby transmission, that is unrelated to their mechanisms. For example, they show ZIKV levels are similar in whole salivary gland homogenates, but the PFU/ul of ZIKV in saliva needs to be assessed (e.g. through forced salivation). Probing time and ability to engorge with blood should also be assessed, as differences in probing time and feeding efficiency could impact on volume of saliva deposited and thereby virus titre.

Figure 7. It is not clear how specific resveratrol is on innate immune responses. Ability of this agent to suppress other TLR4 activating ligands should ideally be defined as a positive control. Resveratrol could be modulating numerous innate immune

pathways, and/or myeloid cell frequency in resting skin/blood, independent of AaNRP function. What effect does resveratrol have on anti-viral innate immune pathways e.g. type I IFN and IFN stimulated gene expression? All of these pathways could impact on overall host susceptibility to infection.

Discussion.

The results presented here suggest that AaNRP accounts for almost all virus enhancing ability of mosquito saliva. What does this mean for the role of other mosquito salivary proteins, including the finding that AaVA-1 (Sun et al) was pivotal for this process, previously published by this group?

Furthermore, the discussion as written here suggests that chemokine expression is the limiting step required for saliva-enhancement of virus infection. Some other reports have suggested chemokine expression alone is not sufficient to mediate saliva-induced enhancement of arbovirus infection (e.g Lefteri 2022 that is widely cited here). Chemokine expression is clearly key (as shown here) but is one of a number of steps, and this should be placed into context in the discussion.

Referee #2:

SUMMARY

In this manuscript, Wang et al. report the identification of a mosquito salivary protein (AaNRP) that triggers the infiltration of neutrophils into the skin. This is important as several prior studies indicated that an influx of neutrophils into the skin enhances arbovirus infection and dissemination in the vertebrate host. The authors further show that this protein triggers the production of chemokines in skin macrophages via activation of MyD88 signaling. Remarkably, the authors identify interactions between AaNRP and murine/human TLR1/TLR4 and show that disrupting these interactions abrogates the induction of chemokines. A series of well-designed studies in mice provide additional evidence that AaNRP enhances ZIKV infection through activation of the MyD88 pathway, promotes establishment and dissemination of ZIKV infection after transmission from infected mosquitoes that express AaNRP in the salivary gland, while KD of AaNRP in mosquitoes abrogates these effects, and that limiting the inflammatory response induced by AaNRP alone, co-delivered with ZIKV, or delivered by mosquito bite can diminish effects and improve outcomes. Overall, this is a comprehensive and careful study that uses a variety of approaches that identifies how AaNRP enhances ZIKV infection. The study should be of substantial interest. A few comments to improve the study are outlined below.

MAJOR

1. Fig 1C: This experiment could include additional controls beyond PBS. For example, does heat-treatment of the protein sample prior to injection in mice negate the neutrophil influx?
2. Fig 1F-1H. The 1A8 antibody targets both Ly-6C and Ly-6G and thus depletes multiple cell type including monocytes. This experiment is not sufficient to conclude that an influx of neutrophils is a prerequisite for recruitment of other myeloid cells. The authors should perform more specific cell depletion studies to support this conclusion and evaluate additional cell populations to demonstrate the specificity of the depletion.
3. Fig 4G-H. These findings suggest that TLR4 is an important recognition receptor for AaNRP. Showing that AaNRP-mediated induction of chemokines is resistant to inhibition by polymyxin B would further support the authors conclusions, as the presence of LPS is a concern.
4. The data in Fig 5C could be more informative if the percentage and total number of infected cells within the specific cell type evaluated was reported. That is, among macrophages, did the frequency and total number of infected macrophages increase with AaNRP? Same question for other cell types.
5. For Fig 5D and other measurements of viremia, the authors should also assess the amount of infectious virus present, as the approach used could be influenced by differences in circulating cells between the experimental groups. In addition, infectious virus in the blood is more informative. Similarly, due to the changes in cell composition in tissues (e.g., shown in Fig 5B and 5C), using the ZIKV/GAPDH mRNA ratio to quantify viral tissue burden in murine tissues throughout this study could be misleading.

MINOR

1. Line 323-324. The authors indicate that experiments were performed to assess flavivirus transmission. This is not an accurate description of the experiments displayed in Fig 5.

ADDITIONAL SUGGESTIONS

1. The authors could consider the use of TLR1 and/or TLR4 deficient animals (which can be treated with an IFNAR1 blocking

Ab) to confirm a role for these receptors in AaNRP driven inflammation and enhancement of ZIKV infection in vivo. This would enhance the study and support the author's model regarding mechanism of action.

Referee #3:

This manuscript by Wang and colleagues reports an intriguing investigation into *Aedes aegypti* salivary proteins that augment ZIKV and DENV infection. The authors reasoned from previous findings that neutrophil recruitment into the skin increases the likelihood of infection establishment and dissemination in the human host. They thus systematically screened recombinant salivary proteins of *Ae aegypti* to determine candidates that recruit neutrophils into the skin and identified AaNRP. They showed that AaNRP bound to TLR-1 and -4 stimulation and activated canonical MyD88-NFkB signaling to induce expression of chemokines that recruited neutrophils to the site of the mosquito bite. They then showed that inhibition of MyD88 signaling in mice, and knockdown of AaNRP expression in infected mosquitoes both reduced ZIKV and DENV-2 infection in mouse footpad. Similarly, both interventions reduced viremia levels at almost all time points measured. They finally showed that daily prophylactic supplementation with resveratrol, which is known to inhibit MyD88-NFkB signaling, could also reduce ZIKV viremia levels. They concluded that their findings provide mechanistic insights into the augmentation of flaviviral infection by mosquito a salivary protein and that this finding could be used prophylactically to prevent disease.

The topic of this study is interesting and important, given the paucity of information on this subject. The authors have also done a huge amount of work for this manuscript. There are, however, several areas of concern that need additional attention. These are:

1. Line 125 and Figure 1C. The authors screened the panel of proteins for neutrophil recruitment activity by inoculating these proteins subcutaneously into mouse footpads. However, the mosquito proboscis only probes the dermis in humans and cannot extend into the subcutaneous layer. Could there be differences in how antigen presenting cells resident in the dermis or even keratinocytes respond to these proteins and hence influence downstream neutrophil recruitment? The experimental approach could thus have underestimated the number of salivary proteins that could play a role in augmenting ZIKV and DENV infection.
2. What is the normal function of AaNRP? Obviously, neutrophil recruitment is not required for successful completion of bloodmeal for mosquitoes to lay their eggs.
3. For many of the experiments, AaNRP was compared against PBS as a negative control. PBS is not the ideal control. A more appropriate negative control would be another protein, preferably a protein that is also excreted in *Ae aegypti* saliva. Given the many negative findings in Figure 1C, a negative control could have been chosen from that list of proteins. Such a control would have supported the specificity of AaNRP in mediating the effects shown in the rest of the manuscript.
4. Lines 138-140 and Fig S1G. Increased expression of AaNRP after a bloodmeal does not support the notion proposed in this study, that AaNRP is necessary to augment infection. AaNRP should be measured before and not after a bloodmeal. This could be done by quantifying AaNRP expression from when the mosquito hatches to when it is ready to become gravid through a bloodmeal.
5. Lines 200-201. The approach of using scRNAseq to identify mediators of neutrophil recruitment completely misses a major contributor of acute inflammation - pre-formed mast cell mediators. These pre-formed chemicals are stored in granules that, upon mast cell activation, would be released through degranulation, several of which are known to play roles in neutrophil recruitment. Post-mosquito bite scRNAseq would thus miss many factors that facilitate neutrophil recruitment.
6. Lines 255-256, Figures 2E and F. The data in Figure 2E is incongruent with 2F, the latter showing that anti-F4/80 mAb treatment inhibited neutrophil infiltration compared to isotype IgG control treatment but not to baseline resting levels. As Figure 2E only showed one point per animal, it would be important to know how the data was obtained. How many sections were examined with IHC per animal? How many fields on each slide were viewed and hence sampled to calculate % neutrophil infiltration. Without this information, Figure 2E could be due to biased sampling.
7. Line 230 and 235. There is no evidence that AaNRP is "the primary" salivary protein involved in neutrophil recruitment. Neither is there any evidence that it was the "major" factor in upregulation of neutrophil chemoattractants. The authors did not assess the other salivary proteins beyond the first screen which was done through subcutaneous and not intradermal inoculation. Please rephrase.
8. Lines 237-240. Does silencing of AaNRP have off-target effects on mosquito salivary production/secretion?
9. Section under lines 320-321 and Figure 5. Inclusion of a control constituting T6167923 treatment followed by ZIKV infection without AaNRP would have been helpful. Blocking MyD88 signaling could impact ZIKV and DENV infection and thus confound the data reported in this section.

10. Lines 353-354 and Figure 6C. RT-qPCR alone to measure viral load is not sufficient. There is no information on how AaNRP silencing would affect ZIKV lifecycle and maturation. Besides measuring viral RNA, the amount of infectious ZIKV particle in the mosquito saliva should also be shown.

11. Lines 437-456. Resveratrol has pleiotropic effects, some of which, such as increased risk of DNA damage, are harmful. For this compound to be used prophylactically to reduce the level of viremia and hence risk of severe disease, persons living in endemic countries will have to take this drug over long periods. How would the prophylactic benefits weigh against the risk associated with long-term usage of this compound? A more balanced discussion would be helpful.

12. Lines 458-460. The notion that lower viremia will reduce human-mosquito-human transmission is incorrect (Duong et al, PNAS 2015; 112:14688). Please rephrase.

Dear Editors,

Thank you for the comments on our manuscript (*EMBOJ-2023-114972*) titled "*A mosquito salivary protein-driven influx of myeloid cells facilitates flavivirus transmission*". We have taken all of the comments into consideration and have revised our manuscript accordingly. We believe that our additional experiments, revised analyses, and modifications of the text have substantially improved the manuscript.

We were encouraged by the positive comments from the editor and reviewers, such as "*As you will see from the reports, all reviewers find the study of interest*" (Editor), "*Overall, this study is nicely done and provides some important new insights.....The role that mosquito saliva has in modulating host susceptibility to virus infection is poorly defined and yet represents a key stage of infection that is common to all viruses transmitted by mosquito to the vertebrate host. As such the findings, if further validated with corrections listed below, are important, novel and of relevance to several key research areas (e.g. virology, immunology, vector biology).*" (Reviewer #1), "*Overall, this is a comprehensive and careful study that uses a variety of approaches that identifies how AaNRP enhances ZIKV infection. The study should be of substantial interest.*" (Reviewer #2), "*The topic of this study is interesting and important, given the paucity of information on this subject. The authors have also done a huge amount of work for this manuscript.*" (Reviewer #3). The reviewers also provided important suggestions to improve the manuscript. We have included new experimental data in our manuscript per the reviewers' suggestions. Along with this letter, we provide point-by-point responses to the queries raised by the reviewers.

Responses to Editor's Comments:

Answer: Thank you for these comments. We have taken all these concerns into consideration and have fully solved these concerns by performing enough experiments. We believe solving these concerns has significantly improved our manuscript. The detailed response to each concern is listed below:

- 1) Exclusion of LPS contamination. Please see **#1A Answer**.
- 2) Further support for the role of neutrophils in myeloid cell recruitment. Please see **#1G Answer, #1J Answer, #2B Answer and #3E Answer**.
- 3) Assessment of potential effects on mosquito feeding and saliva deposition. Please see **#1N Answer and #3H Answer**.
- 4) Better quantification of infected cells and the amount of the infectious virus. Please see **#1M Answer, #2D Answer, #2E Answer and #3J Answer**.

Responses to Referee #1:

#1A. *It is common for recombinant proteins made in-house to have LPS contamination, even for those made using insect cell lines. The accidental presence of LPS could activate these responses observed, both in vivo (e.g. neutrophil recruitment) and in vitro assays (e.g. MyD88 pathway activation). TLR2/4 ligands, when co-inoculated with arbovirus are known to enhance infection with arbovirus. It is essential that AaNRP preps are tested for their ability to activate LPS specific responses and / or LPS concentration defined. As the proposed mechanism of action is via MyD88, a kit that that doesn't use this system to detect endotoxin should be employed e.g. the Pierce LAL Chromogenic Endotoxin Quantitation Kit.*

Even if very low levels of endotoxin are detected, it would then be necessary to show that endotoxin, when spiked into a control recombinant product (e.g. a non-inflammatory mosquito salivary protein other than AaNRP, or scrambled protein) at a similar concentration, does not replicate results observed with AaNRP.

#1A Answer: We recognize the reviewer's concern. In this study, the results indicated that AaNRP activates MyD88-dependent canonical NF- κ B signaling, which drives the expression of neutrophil chemoattractants and the subsequent recruitment of myeloid cells. The recombinant AaNRP protein was expressed and purified in *Drosophila* S2 cells. To address the reviewer's concern, we assessed the LPS contamination in the preparation of 1 $\mu\text{g}/\mu\text{l}$ AaNRP and the PBS solution (Gibco, Cat# 10010023) used to dilute AaNRP preps by the Pierce™ Chromogenic Endotoxin Quant Kit (Cat# A39552S). The endotoxin concentration in the preparation of purified AaNRP protein showed no difference with the PBS solution (average level: AaNRP 0.038 EU/ml, PBS 0.037 EU/ml, $p=0.66$) (Appendix Fig S2A). It is generally believed that an endotoxin content below 0.1 EU/ml has no detectable effect on the cultured cells or animals. In this case, the observed effects should be attributed to AaNRP rather than LPS contamination.

To further address the concern of LPS contamination, we next used polymyxin B, a reagent effectively neutralizing LPS, to assess the possibility of LPS-mediated activation of the MyD88-dependent NF- κ B pathway. Purified AaNRP and LPS were incubated with sterile water (solvent control) or with 30 $\mu\text{g}/\text{ml}$ polymyxin B (neutralization) at 4°C for 24 hours. Then, either sterile water or polymyxin B-treated AaNRP and LPS were added to murine RAW264.7 macrophages for incubation, respectively. Four hours later, the cells were collected for qPCR detection of CXCL1/2/3 expression. The results showed that both sterile water-treated AaNRP and LPS significantly induced the expression of CXCL1/2/3 (Appendix Fig S2B). Polymyxin B preincubation had no influence on AaNRP-induced expression of CXCL1/2/3 but completely abolished LPS-induced CXCL1/2/3 expression (Appendix Fig S2B). These data further validated the specific induction of these chemokines by AaNRP and well addressed the concern of LPS contamination in AaNRP preparation. We have added the data to the revised manuscript (Line 278-293, Page 10 and 11).

Moreover, we included PBS and a heat-inactivated AaNRP (hiAaNRP) as negative

controls, and a noninflammatory control (NIC) salivary protein encoded by *AAEL009524* as an unrelated control. Thus, 100 ng of each protein or an equal volume of PBS was injected into mouse footpads. The injected skin was later collected for a flow cytometry analysis. Skin injected with PBS, hiAaNRP or the *AAEL009524*-encoded protein showed similar levels of neutrophils (Fig 1D) or monocyte-lineage myeloid cells (Fig 1E), while AaNRP-injected skin manifested much higher levels of neutrophils (Fig 1D) and monocyte-lineage myeloid cells (Fig 1E), indicating the specific role of AaNRP in inducing myeloid cell influx. Since AaNRP rather than LPS can be heat-inactivated, the result further excludes the contamination of LPS in the AaNRP preparation.

#1B. Lines 71-72. *There are a number of factors in mosquito saliva that have been identified as able of modulating host susceptibility to arbovirus infection, as listed later in discussion. I would suggest rephrasing this here to make this clear.*

#1B Answer: According to the reviewer's suggestion, we have rephrased the role of previously identified mosquito saliva factors in modulating host susceptibility to arbovirus infection (Line 70-74, Page 3).

#1C. Lines 86-88. *Onset of oedema is very rapid and the role that leukocyte influx has on this process is still to be defined. It is like that both neutrophil influx and oedema occur at the same time based on the published studies cited.*

#1C Answer: Thank you for the reviewer's suggestion. We have revised the sentence to "Thus, neutrophil-driven inflammation leads to an increase in myeloid cells at the mosquito bite site, and these myeloid cells serve as new cellular targets to retain viruses at the inoculation site" (Line 89-92, Page 4).

#1D. Table S1-Line 123 " *We next assessed the role of these salivary proteins in triggering neutrophil influx into the host skin. Thus, 100 ng of purified salivary proteins were individually inoculated into a mouse footpad in a subcutaneous manner". It is not clear why 100ng was chosen and whether this reflects the mass inoculated by a probing mosquito. The amount salivated by mosquitoes, e.g. following forced salivation into oil or media should be define.*

#1D Answer: The dosage for studying mosquito saliva proteins is varied in different studies. For instance, Jin et al. utilized 2 μ g of LTRIN per mouse to examine the enhancing effect of mosquito salivary proteins on ZIKV infection (**Jin et al., 2018, Nature immunology, 19: 342-353**). Ryuta Uraki et al. used 10 μ g of AgBR1 per mouse for their investigation (**Uraki et al., 2019, Nature microbiology, 4: 948-955**). Martin-Martin et al. employed 6 μ g of salivary gland extract (SGE) per mouse to study the impact of mosquito salivary proteins on leukocyte recruitment (**Martin-Martin et al., 2022, Cell reports, 39: 110648**).

In consideration of the traits of mosquito saliva, we chose the optimal dosage of inoculated salivary proteins according to previous evidence. Schmid et al. reported a total protein amount of approximately 370-600 ng in the SGE per individual mosquito (**Schmid et al., 2016, PLoS pathogens, 12: e1005676**). Intriguingly, a mosquito bite

might secrete approximately 0.7 protein amount of SGE (**Wasserman et al., 2004, Parasite immunology, 26: 295-306**), which equates to 250-420 ng of salivary proteins. Given the relatively high prevalence of selected components in mosquito saliva (Appendix Table S1), we decided to inoculate 100 ng of a single mosquito saliva protein into one mouse footpad for our investigation.

According to the reviewer's suggestion, we further measured the amount of salivary proteins secreted by a mosquito bite under forced salivation into microscopic immersion oil. The amount of salivary proteins secreted by a mosquito bite is approximately 300 ng (Appendix Fig S4C).

#1E. *Figure 1C - a Students T test has been used incorrectly and doesn't account for multiple testing. One would expect a p value of less than 0.05 for every 20 proteins studied, based on chance differences. Since there are more than 20 groups, this is quite likely. A one-way ANOVA or Kruskal Wallis, depending on whether data is normally distributed, should be applied instead. This also applies to most of the statistics used elsewhere, in which a Students T test has been used. All should be corrected.*

#1E Answer: We recognize the reviewer's concern. We have reanalyzed all the data throughout our study by using appropriate statistical models. The detailed information about statistical analysis has been shown in figure legends and *Materials and Methods* section of the revised manuscript (Line 934-943, Page 34).

#1F. *Figure 1D-L. It is not clear how leukocyte influx, elicited by injection with AaNRP, compares to mosquito biting and/or injection with equivalent amount of mosquito saliva injection. The authors should offer this comparison for key cell types identified.*

#1F Answer: Thank you for the reviewer's suggestion. We have accordingly included the control inoculation with an equivalent amount of mosquito saliva. In terms of temporal kinetics, both AaNRP and mosquito saliva were able to activate neutrophil infiltration, which occurred as early as 4 hpi and through 24 hpi with a peak at 12 hpi (Fig 1D). Thus, we assessed the recruitment of myeloid cells toward the inoculated footpads. Notably, monocyte-lineage myeloid cells, such as monocytes, macrophages, and immature dendritic cells (DCs), were highly aggregated at the sites inoculated with either AaNRP or mosquito saliva at 6 hpi (Fig 1E).

In the experimental design in Fig 1F-1H, we would ask if AaNRP-mediated myeloid infiltration was dependent on neutrophils. Indeed, accumulating evidence indicates that inoculation of mosquito saliva results in an influx of inflammatory neutrophils, thus coordinating a localized innate immune response to recruit numerous virus-permissive myeloid cells toward bite sites (**Hastings et al., 2019, Journal of virology, 93; Pinggen et al., 2016, Immunity, 44: 1455-1469**). Subsequently, we assessed the role of AaNRP in neutrophil infiltration and subsequent recruitment of myeloid cells in a mosquito biting model. In this experimental model, the mosquitoes with or without AaNRP silencing in saliva were allowed to bite the footpads of mice. The bitten skin was collected for assessment of neutrophil infiltration and recruitment of monocyte lineages (Fig 1I-L). We believe that

the logical designs of these experiments have addressed the reviewer's concern.

#1G. *Figure 1E, F and L. The authors have only assessed monocyte, macrophage and DC influx at 24 hours post injection. This quite a late timepoint, considering enhancement of arbovirus infection by saliva is evident before 6 hours (including in the studies cited in this manuscript e.g. Styer et al). In addition, previously reported infection of myeloid cells occurs within this time frame, long before the 24-hour time point. Virus life cycle is 8 to 10 hours, which means by 24 hours there would have been several rounds of infection and e.g. dissemination of virus to blood. The authors should expand these analyses to include earlier time points. This is key, as entry of virus permissive cells at 24 hours and later may not have significant bearing on defining the outcome to infection, as virus tends to disseminate from skin to blood / systemically within this time frame.*

This is also the case for later figures e.g. Figure 5, which only shows 24-hour post infection.

#1G Answer: We recognize the reviewer's concern. We repeated these experiments in Fig 1E, 1H, 1L and Fig 5B and assessed the influx of monocytes, macrophages and DCs at both 6 and 24 hours post injection. Overall, monocytes, macrophages, and immature dendritic cells (DCs) were highly aggregated at the AaNRP inoculation sites at both 6 hpi (Fig 1E) and 24 hpi (Fig EV2A). Inoculation of anti-neutrophil 1A8 mAb effectively blocked the recruitment of monocytes, macrophages, and immature DCs toward the AaNRP-inoculated mouse footpads at 6 hpi (Fig 1H). The bite-mediated recruitment of monocyte-lineage immune cells was impaired by AaNRP silencing at both 6 hpb (Fig 1L) and 24 hpb (Fig EV2B). The proportions of cutaneous myeloid cells increased in the AaNRP-treated footpads at 6 hpi (Fig 5B), as did the ratios of ZIKV-positive myeloid cells at 24 hpi (Fig 5C). The data have been updated in the revised manuscript.

#1H. *Volume of recombinant protein injected was 20ul, this is many orders of magnitude higher than what a mosquito would deposit. This limitation should be discussed.*

#1H Answer: Thank you for the reviewer's suggestion. The saliva volume expectorated by a mosquito is hard to quantify due to very small amounts and individual differences. By using a microtube forced salivation assay, Sanchez-Vargas et al. reported that a single *Aedes* mosquito salivates an average of 6.82 ± 2.88 nL saliva (**Sanchez-Vargas et al., 2019, *Insects*, 10**). Our injected volume is much greater than this natural volume, which is a limitation of our study. We have discussed this limitation in the revised manuscript (Line 541-548, Page 20).

#1I. *Figure 1 and 5. Absence of virus vs. uninfected skin comparison in leukocyte entry kinetics experiments.*

The purpose of this research is to inform on leukocyte migration during transmission of virus from mosquito to vertebrate skin. Virus is an important activator of

chemokines and leukocyte influx, and yet is excluded from figure 1, and the comparator to resting skin missing in figure 5. This should be included.

#1I Answer: As mentioned by the reviewer, both salivary proteins and viruses play critical roles in stimulating the cutaneous influx of chemokines and leukocytes. In this study, the leading aim is to identify the key factor(s) that cause an influx of neutrophils and subsequent recruitment of myeloid cells toward mosquito bite sites. Thus, we did not involve viral factors in the identification of salivary protein(s) triggering the influx of cutaneous neutrophils during mosquito biting (Fig 1). Nonetheless, we validated the role of AaNRP-mediated influx of myeloid cells in flaviviral transmission by mosquitoes (Fig 5, Fig 6 and Fig 7). According to the reviewer's suggestion, the controls of resting murine skin have been added to the revised manuscript (Fig 5B).

#1J. *Skin macrophages were depleted through use of an antibody to F4/80. Such depletions are difficult to get working at high efficiency for non-circulating tissue resident cells such as these. The complete absence of macrophages in the F4/80 Ab treated group seems surprising, however the antibody used to validate macrophage depletion was the same as that used for depletion. This is not suitable as it could simply indicate epitope masking by antibody rather than true cell depletion. Observed alterations in neutrophil recruitment could instead reflect consequences of antibody infusion associated with this specific Ab clone. Findings should be re-assessed or macrophage depletion validated through alternative means. Furthermore, is F4/80 a good marker for skin macrophages? As they will also deplete other cells in the skin e.g. Langerhans cells. Some macrophage population, e.g. those CD11b+ macrophages derived from bone marrow progenitors are typically low for F4/80. This last point should be discussed e.g. <https://www.ncbi.nlm.nih.gov/pmc/articles/PMC4045180/>*

#1J Answer: We recognize the reviewer's concern. Indeed, the mAb F4/80 (BioxCell, Catalog #BE0206) has been widely used to deplete skin macrophages as well as other noncirculating tissue-resident macrophages. The F4/80-based protocol for the removal of skin macrophages has been generally acknowledged for actual effectiveness (**Bedoret et al., 2009, *The Journal of clinical investigation*, 119: 3723-3738; Wang et al., 2019, *Cell stem cell*, 24: 654-669 e656; Wang et al., 2018, *Cancer cell*, 34: 757-774 e757**). Nonetheless, we validated macrophage depletion by an antibody (Invitrogen, Cat#17-1631-82) against another skin macrophage marker CD163 (**Duangkhae et al., 2018, *The Journal of investigative dermatology*, 138: 618-626**) through both flow cytometry and IHC assays. The results indicated that inoculation of the F4/80 mAb effectively removed the cutaneous macrophages in the murine footpad skin (Fig EV3D and E). We have added the data to the revised manuscript (Line 239-241, Page 9).

We recognize the reviewer's concern for off-target depletion of skin Langerhans cells by inoculation with the F4/80 mAb. Although F4/80 is a good surface marker for dermal and hypodermal macrophages, it is also expressed by some Langerhans cells (**Davies et al., 2013, *Nature immunology*, 14: 986-995; Kashem et al., 2017, *Annual review of immunology*, 35: 469-499**). Nonetheless, the scRNA-Seq data

indicate that Langerhans cells show a relatively low prevalence in murine skin compared to that of skin macrophages, which is consistent with previous reports (**Kashem et al., 2017, Annual review of immunology, 35: 469-499**). In addition, the scRNA-Seq data showed that neutrophil chemoattractants such as CXCL1/2/3 were not induced in Langerhans cells after a mosquito bite (Appendix Fig S1). Thus, these results suggest that Langerhans cells may not be important in AaNRP-mediated neutrophil influx, even if they might be nonspecifically targeted by the F4/80 mAb. We have added this discussion to the revised manuscript (Line 549-559, Page 20 and 21).

We recognize the reviewer's concern about some macrophage populations, e.g., those CD11b+ macrophages derived from bone marrow progenitors are typically low for F4/80. Nonetheless, accumulating evidence indicates that skin resident macrophages are derived from yolk sac progenitors and highly express F4/80 (**Davies et al., 2013, Nature immunology, 14: 986-995; Wynn et al., 2013, Nature, 496: 445-455**). According to the reviewer's suggestion, we have added this to the discussion in the revised manuscript (Line 549-559, Page 20 and 21).

#1K. *Figure 3 and 4 are generally well done, although would be more meaningful if LPS contamination of AaNRP preparation was ruled out. Confirmation of TLR4-MyD88 signalling pathway activation in primary cultures of Macrophages is required, as immortalised RAW cells often generate data that is hard to replicate in terminally differentiated (and thereby relevant) cells.*

#1K Answer: Thank you for the comments. We have addressed the concerns about the potential LPS contamination of *the* AaNRP preparation in **Answer #1A**. According to the reviewer's suggestion, we have reproduced the activation of MyD88-NF- κ B signaling by AaNRP in primary murine peritoneal macrophages. AaNRP strengthened the phosphorylation and degradation of I κ B- α while activating p65 phosphorylation (Fig 3E, ii). A small molecule MyD88 signaling inhibitor, T6167923, fully abolished the AaNRP-mediated activation of NF- κ B signaling in primary murine macrophages, as assessed by the immunoblotting assay (Fig 3E, ii). Furthermore, in the *in vivo* condition, AaNRP significantly activated the canonical NF- κ B signaling axis in mouse footpad skin (Fig EV3G). Overall, we concluded that AaNRP was able to activate MyD88-NF- κ B signaling in primary murine macrophages. We have added the results to the revised manuscript (Line 312-314, Page 12).

#1L. *The specificity of the new anti-AaNRP antibody needs to be defined for data in Fig 4B/C to be meaningful. As for most of the paper, the wrong statistical test has been used; ANOVA is the correct test if there are more than two groups.*

#1 L Answer: We validated the specificity of the new anti-AaNRP antibody by using a flow cytometry assay. The results showed that incubation of the vector-transfected 293T cells with AaNRP Ab (primary Ab) + Fluo. Ab (secondary Ab) showed a few background labeling of the cells, while incubation of the AaNRP-transfected 293T cells with AaNRP Ab + Fluo. Ab showed significantly increased labeling of the cells (Appendix Fig S3A and B). These results validate the specificity of the AaNRP Ab and

have been added to the revised manuscript (Line 335-337, Page 12). We have also reanalyzed the data by using one-way ANOVA (Fig 4C).

#1M. *Figure 5- Representative gating of ZIKV positive cells should be shown.*

#1 M Answer: We have shown the representative gating of ZIKV-positive cells in the revised manuscript (Fig EV4B).

#1N. *Infected cells do not necessarily lead to release of new infectious virus and could also represented phagocytised viral debris. The ability of skin isolated infected leukocytes to release new infectious virus should be defined.*

#1N Answer: To address the reviewer's concern in Fig 5D, we validated the ability of skin CD11b⁺ leukocytes isolated from the murine footpad skin coinoculated with ZIKV and AaNRP to release progeny infectious viruses at 6 hours post *in vitro* culture by focus forming assay. First, we isolated CD11b⁺ leukocytes and CD11b⁻ nonleukocytes from the footpad skin coinoculated with ZIKV and AaNRP at 16 hpi by using flow cytometry. CD11b⁺ leukocytes and CD11b⁻ nonleukocytes were cultured *in vitro* in RPMI 1640 medium with 2% FBS at 37 °C. Six hours later, the supernatants were collected, and the viral titers in the supernatants were determined by focus forming assay. The results showed that the virus-infected leukocytes were definitely able to produce infectious progenies, and in terms of an equal number of cells (5*10³), CD11b⁺ leukocytes produced many magnitudes of infectious viruses than CD11b⁻ nonleukocytes (Fig 5D). This result further corroborated the pivotal role of infiltrated myeloid cells in mediating flavivirus infection.

#1O. *Figure 6-The authors need to demonstrate that siRNA KO of AaNRP does not have any unintentional effects on mosquito feeding, and thereby transmission, that is unrelated to their mechanisms. For example, they show ZIKV levels are similar in whole salivary gland homogenates, but the PFU/ul of ZIKV in saliva needs to be assessed (e.g. through forced salivation). Probing time and ability to engorge with blood should also be assessed, as differences in probing time and feeding efficiency could impact on volume of saliva deposited and thereby virus titre.*

#1O Answer: We recognize the reviewer's concern. We therefore assessed whether dsRNA-mediated silencing of *AaNRP* might cause any unintentional effects on mosquito feeding. In this experiment, the *AaNRP* dsRNA was intrathoracically inoculated into female mosquitoes. The mosquitoes inoculated with *GFP* dsRNA served as negative controls. Three days after administration of dsRNA, the mosquitoes were intrathoracically infected with ZIKV. At 8 days after infection, these mosquitoes were allowed to bite A129 mice. Silencing the *AaNRP* gene did not influence the mosquito's probing time, efficiency of blood engorgement, or capacity for salivation (Appendix Fig S4A-C). Moreover, saliva from individual mosquitoes was collected by forced salivation (**Miller et al., 2021, Insects, 12**) at 8 days after infection. Compared to that of controls, dsRNA-mediated knockdown of *AaNRP* did not influence the ZIKV titer in the saliva of infected mosquitoes, which was assessed by a

focus forming assay (Fig 6D). We have added the data to the revised manuscript (Line 390-395, Page 14).

#1P. *Figure 7-It is not clear how specific resveratrol is on innate immune responses. Ability of this agent to suppress other TLR4 activating ligands should ideally be defined as a positive control. Resveratrol could be modulating numerous innate immune pathways, and/or myeloid cell frequency in resting skin/blood, independent of AaNRP function. What effect does resveratrol have on anti-viral innate immune pathways e.g. type I IFN and IFN stimulated gene expression? All of these pathways could impact on overall host susceptibility to infection.*

#1P Answer: We recognize the reviewer's concern. We exploited LPS, a well-known activating ligand of the MyD88-NF-kB signaling pathway, for subcutaneous inoculation in murine footpad skin. Dietary administration of resveratrol attenuated the LPS-mediated activation of MyD88-NF-kB signaling in murine skin (Appendix Fig S6). Consistently, accumulating evidence has shown that resveratrol can efficiently inhibit the activation of TLR-MyD88-NF-kB signaling by various stimulants, including LPS, in both *in vitro* and *in vivo* conditions (**Huang et al., 2021, The Journal of nutritional biochemistry, 88: 108552; Shi et al., 2017, Annals of the New York Academy of Sciences, 1403: 38-47; Zhang et al., 2014, Molecular medicine reports, 10: 101-106; Zhou et al., 2018, Molecular medicine reports, 17: 1269-1274**).

Given that resveratrol may modulate innate immune pathways, we next assessed whether oral administration of resveratrol in our study may influence immune hemostasis in resting skin and blood by using a flow cytometry assay. Dietary administration of resveratrol did not regulate the frequency of myeloid cells in resting skin (Appendix Fig S5A) or blood (Appendix Fig S5B). We next assessed the potential effects of resveratrol on host antiviral innate immune pathways during ZIKV infection. The skin, lymph nodes and spleens from ZIKV-infected mice at 24 hours post infection with or without dietary administration of resveratrol were collected for an RNA-Seq assay. Oral treatment with resveratrol did not modulate the expression of most of the antiviral immune genes in murine footpad skin, lymph node or spleen (Appendix Table S3). These new results have been added to revised manuscript (Line 413-416, Page 15).

#1Q. *Discussion-The results presented here suggest that AaNRP accounts for almost all virus enhancing ability of mosquito saliva. What does this mean for the role of other mosquito salivary proteins, including the finding that AaVA-1 (Sun et al) was pivotal for this process, previously published by this group?*

#1Q Answer: Mosquito saliva has been demonstrated to promote the infection of mosquito-borne viruses by affecting different steps of infection and by various mechanisms (**Pingen et al., 2017, Trends in parasitology, 33: 645-657; Wang et al., 2023, Insect science**). Accumulating evidence indicates that mosquito bites result in an influx of inflammatory neutrophils that coordinate a localized innate immune response to recruit numerous virus-permissive myeloid cells toward bite sites (**Guerrero et al., 2022, Nature communications, 13: 7036; Hastings et al., 2019,**

Journal of virology, 93; Pingen et al., 2016, Immunity, 44: 1455-1469). Notably, neutrophil depletion and therapeutic blockade of cutaneous inflammation abrogated the bite-mediated promotion of arbovirus infection at both the skin inoculation site and lymphoid tissues (**Pingen et al., 2016, Immunity, 44: 1455-1469**). In this study, we identified AaNRP as a mosquito salivary protein that causes a significant influx of neutrophils and subsequent recruitment of myeloid cells toward mosquito bite sites, thus promoting arboviral infection in their target cells.

In addition, previous studies have identified other mosquito salivary components that facilitate arboviral infection. For example, an *A. aegypti* salivary 34-kDa protein suppresses interferon signaling to enhance DENV replication in human keratinocytes (**Surasombatpattana et al., 2014, The Journal of investigative dermatology, 134: 281-284**). The *A. aegypti* saliva-specific protein venom allergen-1 (AaVA-1) promotes DENV and ZIKV transmission by activating autophagy in host skin myeloid cells (**Sun et al., 2020, Nature communications, 11: 260**). A serine protease called CLIPA3 in *A. aegypti* saliva facilitates DENV infection by proteolyzing extracellular matrix proteins, which benefits viral attachment and cell migration (**Conway et al., 2014, Journal of virology, 88: 164-175**).

As mentioned by the reviewer in **Question #1R**, the previous investigation has highlighted that chemokine expression alone is not sufficient to mediate saliva-induced enhancement of arbovirus infection (**Lefteri et al., 2022, Proceedings of the National Academy of Sciences of the United States of America, 119: e2114309119**) and emphasized that induction of chemokines is definitely one of the essential steps in the saliva enhancement of virus infection. Based on our findings and other studies, we concluded that these salivary components may play critical roles in regulating the different steps in the lineage cascade of cutaneous infection by arboviruses. Interruption of each step in the cascade may effectively impair the whole process of cutaneous infection.

#1R. *Discussion-Furthermore, the discussion as written here suggests that chemokine expression is the limiting step required for saliva-enhancement of virus infection. Some other reports have suggested chemokine expression alone is not sufficient to mediate saliva-induced enhancement of arbovirus infection (e.g Lefteri 2022 that is widely cited here). Chemokine expression is clearly key (as shown here) but is one of a number of steps, and this should be placed into context in the discussion.*

#1R Answer: Thank you for the reviewer's suggestion. We have highlighted that chemokine expression alone is not sufficient to mediate saliva-induced enhancement of arbovirus infection (**Lefteri et al., 2022, Proceedings of the National Academy of Sciences of the United States of America, 119: e2114309119**) and emphasized that induction of chemokines is one of the essential steps in the saliva enhancement of virus infection (Line 480-484, Page 18).

Responses to Referee #2:

#2A. *Fig 1C: This experiment could include additional controls beyond PBS. For example, does heat-treatment of the protein sample prior to injection in mice negate the neutrophil influx?*

#2A Answer: Thank you for the reviewer's question. Reviewer 3 raised a similar concern for the negative controls (**Question #3C**). According to the suggestions, we included PBS and a heat-inactivated AaNRP (hiAaNRP) as negative controls, and a salivary protein encoded by *AAEL009524* acted as an unrelated control. Thus, 100 ng of each protein or an equal volume of PBS was injected into mouse footpads. The injected skin was later collected for a flow cytometry analysis. Skin injected with PBS, hiAaNRP or the *AAEL009524*-encoded protein showed similar levels of neutrophils (Fig 1D) or monocyte-lineage myeloid cells (Fig 1E), while AaNRP-injected skin manifested much higher levels of neutrophils (Fig 1D) and monocyte-lineage myeloid cells (Fig 1E), indicating the specific role of AaNRP in inducing myeloid cell influx.

#2B. *Fig 1F-1H. The 1A8 antibody targets both Ly-6C and Ly-6G and thus depletes multiple cells type including monocytes. This experiment is not sufficient to conclude that an influx of neutrophils is a prerequisite for recruitment of other myeloid cells. The authors should perform more specific cell depletion studies to support this conclusion and evaluate additional cell populations to demonstrate the specificity of the depletion.*

#2B Answer: We recognize the reviewer's concern. Nonetheless, the 1A8 monoclonal antibody (BioXCell, Cat# BP0075-1) binds mouse Ly6G specifically. Unlike the RB6-8C5 antibody, the 1A8 antibody specifically detects murine Ly6G, rather than cross-reactivity with Ly6C (<https://bioxcell.com/invivoplus-anti-mouse-ly6g-bp0075-1>). Thus, the 1A8 monoclonal antibody has been widely used to specifically deplete neutrophils in mice (**Coffelt et al., 2015, Nature, 522: 345-348; Finisguerra et al., 2015, Nature, 522: 349-353; Pingen et al., 2016, Immunity, 44: 1455-1469**). To further address this concern, we ectopically expressed murine Ly-6C in HEK293T cells and used a Ly6C-specific antibody as a positive control to test whether the 1A8 antibody could cross-react with Ly6C. The results showed that although Ly6C-transfected cells can be well labeled by Ly6C-specific antibodies, they could not be labeled by the 1A8 antibody, thereby excluding the potential binding of the 1A8 antibody to Ly6C (Fig R1).

Fig R1. 1A8 antibody (Ab) has no cross-reaction with murine Ly6C.

(A) Representative dot plots showing Ab-bound HEK293T cells. (B) Percentage of Ab-bound HEK293T cells. HEK293T cells were transfected with vector or murine Ly6C plasmid. Forty-eight hours later, the vector- or Ly6C-transfected HEK293T cells were stained with 1A8 Ab or Ly6C Ab as a positive control and the Ab-bound cells were detected by flow cytometry. Data are expressed as the mean \pm SEM, and each dot represents an independent replicate. One-way ANOVA and multiple t tests were used for statistical. **** $p < 0.0001$, n.s. not significant.

#2C. Fig 4G-H. These findings suggest that TLR4 is an important recognition receptor for AaNRP. Showing that AaNRP-mediated induction of chemokines is resistant to inhibition by polymyxin B would further support the authors conclusions, as the presence of LPS is a concern.

#2C Answer: We recognize the reviewer's concern. In this study, the results indicated that AaNRP activates MyD88-dependent canonical NF- κ B signaling, which drives the expression of neutrophil chemoattractants and the subsequent recruitment of myeloid cells. The recombinant AaNRP protein was expressed and purified in *Drosophila* S2 cells. To address the reviewer's concern, we first used polymyxin B, a reagent that effectively neutralizes LPS, to assess the possibility of LPS-mediated activation of the MyD88-dependent NF- κ B pathway. Purified AaNRP and LPS were incubated with sterile water (control) or with 30 μ g/ml polymyxin B (neutralization) at 4 $^{\circ}$ C for 24 hours. Then, sterile water or polymyxin B-treated AaNRP and LPS were added to murine RAW264.7 macrophages for incubation. Four hours later, the cells were collected for qPCR detection of CXCL1/2/3 expression. The results showed that both sterile water-treated AaNRP and LPS significantly induced the expression of CXCL1/2/3. Polymyxin B preincubation had no influence on AaNRP-induced expression of CXCL1/2/3 but completely abolished LPS-induced CXCL1/2/3 expression (Appendix Fig S2B). These data further validated the specific induction of these chemokines by AaNRP and well addressed the concern of LPS contamination in AaNRP preparation. We have added the data to the revised manuscript (Line 282-292, Page 11).

To address the reviewer's concern, we assessed the LPS contamination in the preparation of 1 μ g/ μ l AaNRP and the PBS solution (Gibco, Cat# 10010023) used to dilute AaNRP preps by the Pierce LAL Chromogenic Endotoxin Quantitation Kit (Cat# A39552S). The endotoxin concentration in the preparation of purified AaNRP protein

showed no difference with the PBS solution (average level: AaNRP 0.038 EU/ml, PBS 0.037 EU/ml, $p=0.66$) (Appendix Fig S2A). It is generally believed that an endotoxin content below 0.1 EU/ml has no detectable effect on the cultured cells or animals. In this case, the observed effects should be attributed to AaNRP rather than LPS contamination.

#2D. *The data in Fig 5C could be more informative if the percentage and total number of infected cells within the specific cell type evaluated was reported. That is, among macrophages, did the frequency and total number of infected macrophages increase with AaNRP? Same question for other cell types.*

#2D Answer: We appreciate the reviewer's suggestion. We found that AaNRP did not increase the frequency of ZIKV-infected myeloid cells within their specific cell types (Fig R2A-C). Nevertheless, considering that AaNRP induced considerable recruitment of myeloid cells, and the total number of skin cells is relatively stable, the absolute number of ZIKV-infected myeloid cells was surely increased. Therefore, we believe that increases in the proportions of ZIKV-infected myeloid cells within all skin cells should be better to reflect the infection-enhancing effect of AaNRP.

Fig R2. Frequency of ZIKV-infected myeloid cells within their specific cell types.
 (A) The frequency of ZIKV-infected macrophages within all skin macrophages at 24 hours post infection (hpi). (B) The frequency of ZIKV-infected dendritic cells within all skin dendritic cells at 24 hpi. (C) The frequency of ZIKV-infected monocytes within all skin monocytes at 24 hpi. These data are related to Fig 5C. Data are expressed as the mean \pm SEM, and each dot represents an individual mouse. One-way ANOVA was used for statistical analysis. n.s. not significant.

#2E. *For Fig 5D and other measurements of viremia, the authors should also assess the amount of infectious virus present, as the approach used could be influenced by differences in circulating cells between the experimental groups. In addition, infectious virus in the blood is more informative. Similarly, due to the changes in cell composition in tissues (e.g., shown in Fig 5B and 5C), using the ZIKV/GAPDH mRNA ratio to quantify viral tissue burden in murine tissues throughout this study could be misleading.*

#2E Answer: According to the reviewer's suggestion, we reassessed the viremia and viral loads in murine tissues throughout our study by focus forming assay. Please see

Fig 5D-H, Fig 6D-H, Fig 7F and I, Fig EV4E-H, and Fig EV5A-F.

#2F. *Line 323-324. The authors indicate that experiments were performed to assess flavivirus transmission. This is not an accurate description of the experiments displayed in Fig 5.*

#2F Answer: We have rephrased the expression and replaced *transmission* with *infection* in the revised manuscript (Line 355, Page 13).

#2G. *ADDITIONAL SUGGESTIONS-The authors could consider the use of TLR1 and/or TLR4 deficient animals (which can be treated with an IFNAR1 blocking Ab) to confirm a role for these receptors in AaNRP driven inflammation and enhancement of ZIKV infection in vivo. This would enhance the study and support the author's model regarding mechanism of action.*

#2G Answer: Thank you for the reviewer's suggestion. In this study, we found that AaNRP recognized TLR1/4 on the surface of cutaneous macrophages, thus inducing the expression of these chemoattractants by activating the TLR1/4-MyD88-NF-κB signaling axis. Pharmacological inhibition of TLR1 and TLR4 by a mixture of TLR1/4 antagonists (Anti-TLR-MIX), including CU-CPT22 against TLR-1 and TLR4-IN-C34 against TLR-4, abolished the AaNRP-mediated activation of NF-κB signaling (Fig 4D) and AaNRP-mediated stimulation of CXCL1/2/3 expression (Fig 4E). Moreover, knockdown of *TLR1* and *TLR4* by siRNAs (Fig 4F) consistently abolished the AaNRP-mediated activation of NF-κB signaling (Fig 4G) and the subsequent production of CXCL1/2/3 (Fig 4H). In future investigations, we plan to exploit TLR1- and/or TLR4-deficient animals to further validate a role for TLR receptors in AaNRP-driven inflammation and enhancement of ZIKV infection.

Responses to Referee #3:

#3A. *Line 125 and Figure 1C. The authors screened the panel of proteins for neutrophil recruitment activity by inoculating these proteins subcutaneously into mouse footpads. However, the mosquito proboscis only probes the dermis in humans and cannot extend into the subcutaneous layer. Could there be differences in how antigen presenting cells resident in the dermis or even keratinocytes respond to these proteins and hence influence downstream neutrophil recruitment? The experimental approach could thus have underestimated the number of salivary proteins that could play a role in augmenting ZIKV and DENV infection.*

#3A Answer: We recognize the reviewer's concern. Subcutaneous injection of mosquito salivary proteins and/or mosquito-borne viruses in mouse footpads has been a well-established method to simulate mosquito salivation/bite-mediated infection (**Conway et al., 2016, *PLoS neglected tropical diseases*, 10: e0004941; Jin et al., 2018, *Nature immunology*, 19: 342-353; Pinggen et al., 2016, *Immunity*, 44: 1455-1469; Sun et al., 2020, *Nature communications*, 11: 260; Uraki et al., 2019, *Nature microbiology*, 4: 948-955**). Recently, two noted clinical trials of

mosquito salivary protein vaccines also used a subcutaneous route to inject salivary proteins (*Friedman-Klabanoff et al., 2022, EBioMedicine, 86: 104375; Manning et al., 2020, Lancet, 395: 1998-2007*). Martin-Martin et al. injected SGE via footpad injection, and they described the footpad injection as subcutaneous/intradermal injection (*Martin-Martin et al., 2022, Cell reports, 39: 110648*). We performed the footpad injection at a very superficial layer, which is anatomically located in the dermis layer or the transition region between the dermis and hypodermis, where the mosquito mouthparts can completely reach (*Demeure et al., 2005, J Immunol, 174: 3932-3940*). Thus, our footpad injection is actually intradermal or intradermal/subcutaneous, which could simulate natural mosquito bites. We have corrected all the descriptions of footpad injection as intradermal in the revised manuscript to avoid confusion.

#3B. *What is the normal function of AaNRP? Obviously, neutrophil recruitment is not required for successful completion of bloodmeal for mosquitoes to lay their eggs.*

#3B Answer: Thank you for the reviewer's question. AaNRP, an *A. aegypti* salivary protein encoded by *AAEL007923*, has not been investigated by any previous functional study. We named this protein according to its capability to cause the recruitment of neutrophils to mosquito bite sites. Intriguingly, AaNRP was specifically expressed in the salivary glands of female *A. aegypti* (Fig EV1F) rather than in those of male *A. aegypti* (Fig EV1G). Moreover, the expression of AaNRP was induced following blood feeding (Fig EV1H), suggesting a role of AaNRP in the process of blood feeding and/or host-mosquito interactions. Thus, we plan to further investigate the extended functions of AaNRP in future investigations.

#3C. *For many of the experiments, AaNRP was compared against PBS as a negative control. PBS is not the ideal control. A more appropriate negative control would be another protein, preferably a protein that is also excreted in Ae aegypti saliva. Given the many negative findings in Figure 1C, a negative control could have been chosen from that list of proteins. Such a control would have supported the specificity of AaNRP in mediating the effects shown in the rest of the manuscript.*

#3C Answer: Thank you for the reviewer's suggestion. Reviewer 2 raised a similar concern for the negative controls (**Question #2A**). According to the suggestions, we included PBS and a heat-inactivated AaNRP (hiAaNRP) as negative controls, and a noninflammatory control (NIC) salivary protein encoded by *AAEL009524* acted as an unrelated control. Thus, 100 ng of each protein or an equal volume of PBS was injected into mouse footpads. The injected skin was later collected for a flow cytometry analysis. Skin injected with PBS, hiAaNRP or the *AAEL009524*-encoded protein (NIC) showed similar levels of neutrophils (Fig 1D) or monocyte-lineage myeloid cells (Fig 1E), while AaNRP-injected skin manifested much higher levels of neutrophils (Fig 1D) and monocyte-lineage myeloid cells (Fig 1E), indicating the specific role of AaNRP in inducing myeloid cell influx.

#3D. Lines 138-140 and Fig S1G. Increased expression of AaNRP after a bloodmeal does not support the notion proposed in this study, that AaNRP is necessary to augment infection. AaNRP should be measured before and not after a bloodmeal. This could be done by quantifying AaNRP expression from when the mosquito hatches to when it is ready to become gravid through a bloodmeal.

#3D Answer: We agree with the reviewer's suggestion. These data (Fig EV1H) showed that AaNRP was upregulated in the mosquito salivary glands after a blood meal. Due to the infection-enhancing effect of AaNRP, blood-fed mosquitoes with upregulated AaNRP might thus be more effective in transmitting viruses to vertebrate hosts. Based on the editor's suggestion, we have now adjusted the interpretation to indicate that AaNRP might have additional functions post bloodmeal. We have added this to the revised manuscript (Line 146-148, Page 6).

#3E. Lines 200-201. The approach of using scRNAseq to identify mediators of neutrophil recruitment completely misses a major contributor of acute inflammation - pre-formed mast cell mediators. These pre-formed chemicals are stored in granules that, upon mast cell activation, would be released through degranulation, several of which are known to play roles in neutrophil recruitment. Post-mosquito bite scRNAseq would thus miss many factors that facilitate neutrophil recruitment.

#3E Answer: We recognize this careful concern. The preformed chemicals stored in granules may be released from activated mast cells, thus leading to the recruitment of neutrophils (**Demeure et al., 2005, J Immunol, 174: 3932-3940**). scRNA-Seq was used to detect the regulation of gene transcripts post mosquito bites, thereby missing the regulation of pre-formed factors mediated by mosquito bites. We therefore assessed the activation of cutaneous mast cells at 4 hours post mosquito bite (hpb). The percentage of activated mast cells, defined as CD63+CD203c+ (**Ebo et al., 2021, The Journal of allergy and clinical immunology, 147: 1143-1153**), showed no significant difference between resting skin and mosquito bitten skin (Fig EV3C), indicating that the mosquito bite does not induce the activation of cutaneous mast cells at the early time point after biting (Fig EV3C). In this study, we found that skin resident macrophages are activated by mosquito bites to release neutrophil chemoattractants. Antibody-mediated depletion of skin resident macrophages significantly offset the recruitment of neutrophils toward the mosquito bite site. These findings indicate that skin-resident macrophages play a pivotal role in neutrophil recruitment induced by mosquito bites.

#3F. Lines 255-256, Figures 2E and F. The data in Figure 2E is incongruent with 2F, the latter showing that anti-F4/80 mAb treatment inhibited neutrophil infiltration compared to isotype IgG control treatment but not to baseline resting levels. As Figure 2E only showed one point per animal, it would be important to know how the data was obtained. How many sections were examined with IHC per animal? How many fields on each slide were viewed and hence sampled to calculate % neutrophil infiltration. Without this information, Figure 2E could be due to biased sampling.

#3F Answer: We recognize the reviewer's concern. Indeed, these figures represent

different experiments, with Fig 2E displaying flow cytometry data and Fig 2F presenting IHC images. In light of this, we have now included a more representative IHC panel in Fig 2F in the revised manuscript. We had two footpad sections for each animal and three random scopes were sampled from each section, and the most representative image was selected to delegate the mouse (this is now clearly stated in Fig 2F legend).

#3G. *Line 230 and 235. There is no evidence that AaNRP is "the primary" salivary protein involved in neutrophil recruitment. Neither is there any evidence that it was the "major" factor in upregulation of neutrophil chemoattractants. The authors did not assess the other salivary proteins beyond the first screen which was done through subcutaneous and not intradermal inoculation. Please rephrase.*

#3G Answer: Thank you for the reviewer's suggestion. We have now rephrased the expression properly to define the function of AaNRP in the revised manuscript (Line 246 and 251, Page 9). We have clarified the subcutaneous/intradermal injection details in the **#3A Answer**.

#3H. *Lines 237-240. Does silencing of AaNRP have off-target effects on mosquito salivary production/secretion?*

#3H Answer: We appreciate this concern and have performed a forced salivation assay to check whether AaNRP silencing affects mosquito salivation. In this experiment, the AaNRP dsRNA was intrathoracically inoculated into mosquitoes. The mosquitoes inoculated with GFP dsRNA served as negative controls. Three days post injection with dsRNA, the mosquitoes were intrathoracically injected with ZIKV. At 8 days postinjection with ZIKV, saliva from each mosquito was collected and quantified. The results showed that AaNRP silencing by dsRNA had no influence on mosquito salivary production (Appendix Fig S4C).

#3I. *Section under lines 320-321 and Figure 5. Inclusion of a control constituting T6167923 treatment followed by ZIKV infection without AaNRP would have been helpful. Blocking MyD88 signaling could impact ZIKV and DENV infection and thus confound the data reported in this section.*

#3I Answer: We recognize this concern and have included a PBS + ZIKV + T6167923 group in Fig 5B-I in the revised manuscript. Consistent with previous results, the reproduced experiments showed that AaNRP significantly enhanced myeloid cell influx (Fig 5B) and ZIKV infection (Fig 5C-I), while the MyD88 inhibitor T6167923 efficiently attenuated these effects by AaNRP. Notably, in the absence of AaNRP, T6167923 alone had no influence on myeloid cell influx or ZIKV infection (Fig. 5B-5I). Indeed, this is understandable because mice were only i.p. injected with T6167923 once (one hour prior to s.c. injection with PBS/ZIKV/AaNRP) throughout the experiments, such a single dose of T6167923 may only specifically affect early surged inflammatory responses induced by AaNRP and may have little or no influence on ZIKV infection per se.

#3J. *Lines 353-354 and Figure 6C. RT-qPCR alone to measure viral load is not sufficient. There is no information on how AaNRP silencing would affect ZIKV lifecycle and maturation. Besides measuring viral RNA, the amount of infectious ZIKV particle in the mosquito saliva should also be shown.*

#3J Answer: According to the reviewer's suggestion, we reassessed the ZIKV load in mosquito saliva by a focus forming assay. Saliva from each mosquito was collected at 8 days post infection with ZIKV, and the infectious viral titers were quantified by focus forming assay. The results showed that AaNRP silencing did not influence ZIKV loads in mosquito saliva. This data has been added to Fig 6D in the revised manuscript.

#3K. *Lines 437-456. Resveratrol has pleiotropic effects, some of which, such as increased risk of DNA damage, are harmful. For this compound to be used prophylactically to reduce the level of viremia and hence risk of severe disease, persons living in endemic countries will have to take this drug over long periods. How would the prophylactic benefits weigh against the risk associated with long-term usage of this compound? A more balanced discussion would be helpful.*

#3K Answer: We recognize the reviewer's concern. Some previous studies have suggested that resveratrol can increase DNA damage in pathogenic bacteria (**Lee and Lee, 2017, Biochemical and biophysical research communications, 489: 228-234**) and tumor cells (**Qian et al., 2022, Journal of oncology, 2022: 9672773**). Although both of these effects are beneficial for host animals and resveratrol-increased DNA damage is usually observed in tumor cells rather than in normal cells (**Hadi et al., 2010, Pharmaceutical research, 27: 979-988**), resveratrol still has the potential to induce DNA damage in some types of normal cells under specific conditions (**Hadi et al., 2010, Pharmaceutical research, 27: 979-988**). Nonetheless, there are also many investigations demonstrating that resveratrol can prominently reduce DNA damage and other oxidative injuries induced by various physical, chemical or biological factors in multiple tissues (such as skin, liver and brain) (**Afaq et al., 2003, Toxicology and applied pharmacology, 186: 28-37; Jin et al., 2023, Cell death & disease, 13: 847; Pal and Sarkar, 2014, Environmental toxicology and pharmacology, 38: 684-699; Zemheri-Navruz et al., 2023, Environmental science and pollution research international, 30: 6414-6423**) and different cell types (such as macrophages, lymphocytes and epithelial cells) (**Aydin et al., 2013, Human & experimental toxicology, 32: 1048-1057; Dobrzynska and Gajowik, 2022, Radiation research, 197: 149-156; Moon et al., 2006, Archives of pharmacal research, 29: 213-217**) (**Dobrzynska and Gajowik, 2022, Radiation research, 197: 149-156; Quincozes-Santos et al., 2007, Neurotoxicology, 28: 886-891; Zhang et al., 2020, International journal of molecular medicine, 45: 1673-1684**). Resveratrol is a polyphenolic nutraceutical that exhibits pleiotropic activities in human subjects. The efficacy, safety, and pharmacokinetics of resveratrol have been documented in over 244 clinical trials, with an additional 27 clinical trials currently ongoing. Resveratrol is reported to potentially improve the therapeutic outcome in patients suffering from various human diseases (**Singh et al., 2019, Medicinal research reviews, 39: 1851-1891**). Resveratrol is reported to be safe at

doses up to 5 g/day when used either alone or as a combination therapy (**Breuss et al., 2019, International journal of molecular sciences, 20; Singh et al., 2019, Medicinal research reviews, 39: 1851-1891**). Nonetheless, we should be aware of the potential side effects of resveratrol. Long-term epidemiological surveillance is needed to assess the safety and prophylactic benefits to prevent flavivirus transmission. We have added this discussion to the revised manuscript (Line 508-529, Page19).

#3L. *Lines 458-460. The notion that lower viremia will reduce human-mosquito-human transmission is incorrect (Duong et al, PNAS 2015; 112:14688). Please rephrase.*

#3L Answer: Thank you for this correction. The relevant statements have been removed in the revised manuscript.

Dear Gong,

Thank you for submitting a revised version of your manuscript. Your study has now been seen by two of the original referees, who find that their previous concerns have been addressed and now recommend acceptance of the manuscript.

There now remain only a few editorial points that need addressing before I can extend formal acceptance of the manuscript:

1. Email to the contributing author Xuechun Feng did not reach the addressee - please double check the provided address (fengxc@szbl.ac.cn).
2. CRediT has replaced the traditional author contributions section because it offers a systematic, machine-readable author contributions format that allows for more effective research assessment. Please remove the Authors Contributions from the manuscript and use the free text boxes beneath each contributing author's name in our online submission system to add specific details on the author's contribution. More information is available in our guide to authors.
3. Please rename "Conflict of interest" section into "Disclosure and competing interests statement" (further info: <https://www.embopress.org/page/journal/14602075/authorguide#conflictsofinterest>).
4. In the Data Availability section, please add resolvable links to the datasets. More information about the format of this section can be found here: <https://www.embopress.org/page/journal/14602075/authorguide#dataavailability>.
5. Our data editors have flagged the following issues in figure legends that need correcting:
 - The statistical test part of the legend of figure 6i is incorrectly labelled as 6h, please correct.
6. In our standard image check, we noticed that there are undeclared splice sites in the figures 1A and 1B, including the source data for figure 1 B. Please indicate the splice sites with vertical separating lines and submit the full source data for figure 1B.
7. The following blots appear to be vertically flipped in comparison to the source data: Figure 4F (actin), 4G (p-p65). On the other hand, in Figure 7B, it appears that the IkappaB and p-IkappaB blots should be flipped vertically (higher molecular weight direction up).
8. Papers published in The EMBO Journal are accompanied online by a 'Synopsis' to enhance discoverability of the manuscript. It consists of A) a short (1-2 sentences) summary of the findings and their significance, B) 3-4 bullet points highlighting key results and C) a synopsis image that is 550x300-600 pixels large (width x height, jpeg or png format). You can either show a model or key data in the synopsis image. Please note that the image size is rather small and that text needs to be readable at the final size. Please send us this information together with the revised manuscript.

With best wishes,

Ieva

Ieva Gailite, PhD
Senior Scientific Editor
The EMBO Journal
Meyershofstrasse 1
D-69117 Heidelberg
Tel: +4962218891309
i.gailite@embojournal.org

We realize that it is difficult to revise to a specific deadline. In the interest of protecting the conceptual advance provided by the work, we recommend a revision within 3 months (25th Apr 2024). Please discuss the revision progress ahead of this time with the editor if you require more time to complete the revisions.

Referee #1:

The authors have comprehensively answered all of this reviewer's queries and suggested changes. The new experiments undertaken are well done and precisely rebut the concerns that had been raised. The findings of this report are novel and will be of interest to a wide variety of researchers in several fields, not limited to vector-borne disease, Immunology and dermatology. Congratulations to the authors, and all those involved, in putting together such an interesting and well written manuscript. I look forward to seeing it in print!

Referee #2:

The authors thoroughly and rigorously addressed the prior concerns and the study is substantially improved. I do not have additional specific concerns.

The authors addressed the remaining editorial issues.

Dear Gong,

Thank you for addressing the final editorial issues. I am now pleased to inform you that your manuscript has been accepted for publication.

I will look into the synopsis text in the next couple of days and let you know if any edits to the journal style are needed.

Finally, we would like to promote your manuscript among the Chinese readership. Therefore, we would like to invite you to prepare a short summary of the manuscript in Chinese (1500-2000 Chinese characters), which we will promote on the WeChat platform 'BioArt' with 440,000 followers.

If you are interested in this opportunity, we recommend covering the article very close to its online publication date. Thus, ideally we would very much appreciate if you could send us a draft within the next 7 working days. Please let us know whether or not you would be interested in contributing such a short summary in Chinese.

I have included below some general guidelines on how to prepare a summary and a link to recent examples for your reference. Please let me know if you have any questions about this.

If you have any questions, please do not hesitate to contact the Editorial Office. Thank you for this contribution to The EMBO Journal and congratulations on a great paper!

Best wishes,

Ieva

Ieva Gailite, PhD
Senior Scientific Editor
The EMBO Journal
Meyerohofstrasse 1
D-69117 Heidelberg
Tel: +4962218891309
i.gailite@embojournal.org

General WeChat Summary Guidelines

1. These summary articles are meant to be targeting general audience so please limit the use of specialized technical terms, acronyms and jargon.
2. A summary usually starts with brief background information of the reported work, which is followed by explaining the findings in some detail, and ends with a short review of the conclusions as well as the implications of the work and future directions for the research.

3. The summary should at least contain one graphical item, such as a scheme or a figure from the paper.
4. Please provide ONE SINGLE document containing all text and graphical materials, ideally as a Word.docx or .doc file. Please DO NOT provide the document as a .pdf file.
5. Please DO NOT publicly release the document before the paper is officially published online.

Summary Examples

EMBO J | 罗招庆/欧阳松应揭示谷酰胺脱氨酶MvcA的去泛素化功能

EMBO J | 王松灵院士团队揭示组织内应力调控大型哺乳动物乳恒牙替换的新机制